# MOMENTUM AND ERROR FEEDBACK FOR CLIPPING WITH FAST RATES AND DIFFERENTIAL PRIVACY

## ABSTRACT

Achieving both strong Differential Privacy (DP) and efficient optimization is critical for Federated Learning (FL), where client data must remain confidential without compromising model performance. However, existing methods typically sacrifice one for the other: they either provide robust DP guarantees at the cost of assuming bounded gradients/data heterogeneity, or they achieve strong optimization rates without any privacy protection. In this paper, we bridge this gap by introducing `Clip21-SGD2M`, a novel method that integrates gradient clipping, heavy-ball momentum, and error feedback to deliver state-of-the-art optimization and strong privacy guarantees. Specifically, we establish optimal convergence rates for non-convex smooth distributed problems, even in the challenging setting of heterogeneous client data, without requiring restrictive boundedness assumptions. Additionally, we demonstrate that `Clip21-SGD2M` achieves competitive (local) DP guarantees, comparable to the best-known results. Numerical experiments on non-convex logistic regression and neural network training confirm the superior optimization performance of our approach across a wide range of DP noise levels, underscoring its practical value in real-world FL applications.

## 1 INTRODUCTION

Federated Learning (FL) (Konečný et al., 2016; McMahan et al., 2017a) is a modern training paradigm where multiple (possibly heterogeneous) clients aim to collaboratively train a shared model without exposing their private data. This paradigm brings a host of design challenges, including communication efficiency, partial participation of clients, data heterogeneity, security, and privacy (Kairouz et al., 2021; Wang et al., 2021), which have spurred the development of numerous optimization methods for FL. Yet despite this progress, it remains difficult to design FL algorithms that achieve both fast optimization convergence and strong differential privacy (DP) guarantees (Dwork et al., 2014) due to the conflicting nature of these objectives. Indeed, most of the results in the field of DP are obtained by injecting noise (e.g. Gaussian noise) into the method's update (Abadi et al., 2016; Chen et al., 2020) to protect the client's data and prevent data reconstruction. This inevitably reduces update accuracy and slows convergence. Furthermore, to control sensitivity and ensure DP, updates must be bounded—typically by applying *gradient clipping* (Pascanu et al., 2013)—before noise injection.

In FL, *data heterogeneity* is ubiquitous and critically affects algorithmic behavior. Indeed, naïve distributed Clipped Gradient Descent (`Clip-GD`) can fail to converge under *heterogeneous* client data—even without any DP-noise (Khirirat et al., 2023). To tackle this issue, Khirirat et al. (2023) embeds the `EF21` mechanism—originally proposed by Richtárik et al. (2021) to enhance standard Error Feedback (Seide et al., 2014) for contractive compressors—into `Clip-GD`, resulting in a method known as `Clip21-GD`. They prove that, unlike `Clip-GD`, `Clip21-GD` attains an $\mathcal{O}(1/T)$ rate on smooth non-convex objectives for arbitrary heterogeneous data on clients. However, their guarantees rely on full-batch gradients and break down in the presence of DP noise. This leads us to the natural question:

*Is it possible to design a method that achieves both fast convergence and strong DP guarantees while accommodating arbitrary data heterogeneity?*

**Our contribution.** We answer this affirmatively by introducing `Clip21-SGD2M`, a novel algorithm that integrates gradient clipping, error-feedback, and Heavy-Ball momentum (Polyak, 1964). For smooth non-convex distributed objectives under arbitrary data heterogeneity, we prove that `Clip21-SGD2M` $(i)$ attains the optimal $\mathcal{O}(1/T)$ in the full-batch regime, $(ii)$ achieves the optimal high-probability convergence rate $\widetilde{\mathcal{O}}(1/\sqrt{nT})$ when using sub-Gaussian stochastic gradients, and $(iii)$ achieves competitive local DP-error when DP-noise is added to the clients' updates. We further show that `Clip21-SGD` can fail to converge with stochastic gradients, underscoring the critical role of our momentum extension. Our experiments on logistic regression and neural networks highlight the robustness of `Clip21-SGD2M` across clipping thresholds and its competitive privacy-utility trade-off compared to several baselines at fixed DP budgets.

## 1.1 PROBLEM FORMULATION AND ASSUMPTIONS

We consider the optimization problem of the form

$$\min_{x \in \mathbb{R}^d} \left[ f(x) := \tfrac{1}{n} \sum_{i=1}^n f_i(x) \right], \tag{1}$$

where $x$ are the model parameters, $f_i$ is the loss associated with the local dataset $\mathcal{D}_i$ of worker $i \in [n]$, and $f$ is the overall average loss across all $n$ clients.

We work under two standard assumptions. First, we assume smoothness and a finite optimum (Carmon et al., 2020; Danilova et al., 2022).

**Assumption 1.1.** Each individual loss function $f_i$ is $L$-smooth, i.e., for any $x, y \in \mathbb{R}^d$ and $i \in [n]$ we have

$$\|\nabla f_i(x) - \nabla f_i(y)\| \le L\|x - y\|. \tag{2}$$

Moreover, we assume that $f^* := \inf_{x \in \mathbb{R}^d} f(x) > -\infty$.

Our analysis can be straightforwardly generalized to allow each $f_i$ to have its own smoothness constant $L_i$. Second, since full gradients are often impractical, we model stochastic gradients with sub-Gaussian noise.

**Assumption 1.2.** Each worker $i$ has access to a $\sigma$-sub-Gaussian unbiased estimator $\nabla f_i(x, \xi)$ of a local gradient $\nabla f_i(x)$, i.e., for some[1] $\sigma \ge 0$ and any $x \in \mathbb{R}^d$ and $\forall i \in [n]$ we have

$$\mathbb{E}\left[\nabla f_i(x, \xi)\right] = \nabla f_i(x), \mathbb{E}\left[\exp\left(\|\theta_i\|^2/\sigma^2\right)\right] \le \exp(1), \tag{3}$$

where $\xi$ denotes the source of the stochasticity and $\theta_i := \nabla f_i(x, \xi) - \nabla f_i(x)$.

Although this assumption is stronger than bounded variance, it is standard for the high-probability[2] analysis of `SGD`-type methods with polylogarithmic dependence on the confidence level (Nemirovski et al., 2009; Ghadimi & Lan, 2012). Equivalently, the second part of (3) implies the tail bound $\Pr\left(\|\theta_i^t\| \ge b\right) \le 2\exp\left(-b^2/(2\sigma^2)\right)$ (up to constant factors in $\sigma^2$) (Vershynin, 2018). Our results can be extended to heavier sub-Weibull tails (Madden et al., 2024)—still with only polylogarithmic dependence on the confidence level—at the cost of worse logarithmic factors in the final rates (Madden et al., 2024).

Finally, we introduce two key definitions. The first one is the clipping operator, a nonlinear map from $\mathbb{R}^d$ to $\mathbb{R}^d$ parameterized by the clipping threshold/level $\tau > 0$ and defined as

$$\text{clip}_\tau(x) := \begin{cases} \frac{\tau}{\|x\|}x, & \text{if } \|x\| > \tau, \\ x, & \text{if } \|x\| \le \tau. \end{cases} \tag{4}$$

Second, we recall the standard definition of $(\varepsilon, \delta)$-Differential Privacy, which introduces plausible deniability into the output of a learning algorithm.

**Definition 1.3** ($(\varepsilon, \delta)$-Differential Privacy (Dwork et al., 2014))**.** A randomized method $\mathcal{M} : \mathcal{D} \to \mathcal{R}$ satisfies $(\varepsilon, \delta)$-Differential Privacy ($(\varepsilon, \delta)$-DP) if for any adjacent datasets $D, D' \in \mathcal{D}$ (e.g., if $D$ and $D'$ differ in 1 sample) and for any $S \subseteq \mathcal{R}$

$$\Pr\left(\mathcal{M}(D) \in S\right) \le e^\varepsilon \Pr\left(\mathcal{M}(D') \in S\right) + \delta. \tag{5}$$

---

[1]For simplicity, we define $0/0 := 0$. Then, (3) with $\sigma = 0$ implies $\nabla f_i(x, \xi) = \nabla f_i(x)$ almost surely.

[2]We elaborate on the reasons why we focus on high-probability analysis in Section 3.2.

In this definition, the smaller $\varepsilon, \delta$ are, the more private the method is. Intuitively, if inequality (5) holds with small values of $\varepsilon$ and $\delta$, it becomes difficult to infer the specific data point that differs between two similar datasets based solely on the output of $\mathcal{M}$.

## 1.2 RELATED WORK

**Differential Privacy.** The standard recipe for differential privacy in federated learning is to first clip each client's update to a fixed $\ell_2$-norm bound and then add Gaussian noise—either to each individual update or to their aggregated average—so as to mask the influence of any single participant (McMahan et al., 2017b). There are two prevailing privacy models. In the *central model*, a trusted server gathers updates from clients and injects noise only when forming the global update; this protects client data from external observers but still requires trusting the server. In the *local model*, each client clips and perturbs its own update before transmission, thus safeguarding privacy even against the server and other clients (Kasiviswanathan et al., 2011; Allouah et al., 2024). While local privacy offers stronger protection, it typically degrades learning accuracy, since heavier noise is needed to obscure individual updates (Chan et al., 2012; Duchi et al., 2018). This trade-off can be mitigated by using secure shuffling, which randomly permutes client updates before aggregation (Erlingsson et al., 2019; Balle et al., 2019), or a secure aggregator (Bonawitz et al., 2017), which sums updates before sending them to the server. These methods anonymize updates and enhance privacy while maintaining reasonable learning performance, even without a fully trusted server. Finally, (Chaudhuri et al., 2022; Hegazy et al., 2024) show that when DP is required, one can also achieve compression of updates for free.

In this work, we adopt the local DP model by injecting Gaussian noise into each client's update. However, the average noise can also be viewed as noise added to the average update. Therefore, `Clip21-SGD2M` is compatible with all the aforementioned techniques and can also be applied to the central DP model with a smaller amount of noise. However, it is worth mentioning that our analysis is not directly compatible with the privacy amplification by sub-sampling (Balle et al., 2018; Li et al., 2012; Dong et al., 2025; Bonawitz et al., 2017), which is another important tool for achieving improved DP guarantees.

**Error Feedback.** Error Feedback (EF) (Seide et al., 2014) is widely used to incorporate communication compression into distributed and federated learning, but its convergence theory for smooth non-convex objectives has remained limited. Existing analyses either focus on the single-node setting or impose stringent conditions—such as bounded gradient/compression error, or under data heterogeneity (gradient dissimilarity)—to prove convergence (Stich et al., 2018; Stich & Karimireddy, 2019; Karimireddy et al., 2019; Koloskova et al., 2019; Beznosikov et al., 2023; Tang et al., 2019; Xie et al., 2020; Sahu et al., 2021). Moreover, the known EF convergence rates degrade in the presence of client heterogeneity, and this dependence is not merely an artifact of the proofs—it shows up empirically in solving strongly convex problems (Gorbunov et al., 2020b). To overcome these drawbacks, Richtárik et al. (2021) introduced EF21, a variant whose convergence guarantees no longer rely on heterogeneity bounds; however, `EF21-SGD` still requires increasingly large batch sizes to reach any fixed accuracy (Fatkhullin et al., 2021). Fortunately, this drawback is not fundamental: recent work demonstrates that adding Heavy-Ball momentum removes the large-batch requirement (Fatkhullin et al., 2024), and momentum likewise enhances EF's performance in decentralized setting (Yau & Wai, 2022; Huang et al., 2023; Islamov et al., 2024a).

**Distributed methods with clipping.** In the single-node setting, `Clip-SGD` has been rigorously studied under a range of assumptions (Zhang et al., 2020b;c;a; Gorbunov et al., 2020a; Cutkosky & Mehta, 2021; Sadiev et al., 2023; Liu et al., 2023). These analyses extend to multi-client training when clipping is applied to the aggregate (e.g., the averaged update), although mini-batching requires a refined analysis when the noise is heavy-tailed (Kornilov et al., 2024). However, ensuring DP requires clipping each client's communicated update before aggregation; in this regime `Clip-SGD` can fail to converge *even with deterministic gradients* (Chen et al., 2020; Khirirat et al., 2023). To recover convergence, prior work imposes additional restrictive *heterogeneity bounds*. For instance, Liu et al. (2022) prove convergence of a clipped `FedAvg/Local-SGD` variant under *homogeneous* clients with gradients symmetric around their mean, and Wei et al. (2020) analyze clipped `Local-SGD` assuming *bounded heterogeneity*. Other approaches assume *bounded gradients* (thereby implicitly bounding heterogeneity): Zhang et al. (2022) study `FedAvg` with clipping of

model differences (see also the empirical study in (Geyer et al., 2017)); Noble et al. (2022) propose and analyze DP-SCAFFOLD; Li & Chi (2023) develop PORTER (a clipped BEER) under bounded-gradient/heterogeneity assumptions; Allouah et al. (2023) give convex lower bounds and new upper bounds for distributed SGD with momentum and clipped stochastic gradients; and Allouah et al. (2024) study clipped Gossip-SGD (DECOR). While these methods come with formal DP guarantees, none prove convergence *without some bounded heterogeneity condition*. Moreover, several works require the clipping threshold to *exceed the norm of the communicated vector* (Zhang et al., 2022; Noble et al., 2022; Allouah et al., 2023; 2024), rely on symmetric gradient noise (Liu et al., 2022), or assume full-gradient computation at clients (Wei et al., 2020). In this work, we remove these limitations: Clip21-SGD2M achieves fast optimization and strong (local-)DP guarantees under arbitrary data heterogeneity.

**Challenges of Coupling Error Feedback and Clipping.** Various prior works have combined error feedback with clipping. In particular, Khirirat et al. (2023) introduced Clip21-GD by embedding the EF21 mechanism into the gradient-clipping operator, while Gorbunov et al. (2024) developed algorithms that clip the difference between stochastic gradients and learnable shifts – an idea originally proposed by Mishchenko et al. (2019) to address data heterogeneity under unbiased communication compression. Viewing *clipping as a contractive compressor*, as suggested by Khirirat et al. (2023), highlights a key limitation: standard contractive compressors admit a uniform contraction factor across all inputs, whereas the contractive behavior of clipping is inherently input-dependent. To address this limitation, Khirirat et al. (2023) analyzed Clip21-GD only in a full-batch, noise-free regime and *without a valid DP guarantee*.[3] More recently, Shulgin et al. (2025a;b) partially closed this DP gap by replacing clipping with a smoothed normalization operator. However, their guarantees still depend on *full-batch gradients* and *careful initialization*. Thus, it remains an open problem whether error feedback and clipping can be combined in a way that avoids such restrictive theoretical assumptions.

## 2 NON-CONVERGENCE OF CLIP-SGD AND CLIP21-SGD

We start with a discussion of the key limitation of Clip-SGD (Algortihm 1) and Clip21-SGD (Alg. 2) – their potential non-convergence.

| **Algorithm 1** Clip-SGD (Abadi et al., 2016) | **Algorithm 2** Clip21-SGD (Khirirat et al., 2023) |
|---|---|
| **Require:** $x^0 \in \mathbb{R}^d$, stepsize $\gamma > 0$, clipping parameter $\tau > 0$ | **Require:** $x^0, g^0 \in \mathbb{R}^d$, stepsize $\gamma > 0$, clipping parameter $\tau > 0$ |
| 1: | 1: Initialize $g_i^0 = g^0$ for all $i \in [n]$ |
| 2: **for** $t = 0, \ldots, T-1$ **do** | 2: **for** $t = 0, \ldots, T-1$ **do** |
| 3: | 3: $\quad x^{t+1} = x^t - \gamma g^t$ |
| 4: $\quad$ **for** $i = 1, \ldots, n$ in parallel **do** | 4: $\quad$ **for** $i = 1, \ldots, n$ in parallel **do** |
| 5: | 5: $\quad\quad c_i^{t+1} = \text{clip}_\tau(\nabla f_i(x^{t+1}, \xi_i^{t+1}) - g_i^t)$ |
| 6: $\quad\quad g_i^t = \text{clip}_\tau(\nabla f_i(x^t, \xi_i^t))$ | 6: $\quad\quad g_i^{t+1} = g_i^t + c_i^{t+1}$ |
| 7: $\quad$ **end for** | 7: $\quad$ **end for** |
| 8: $\quad g^t = \frac{1}{n} \sum_{i=1}^n g_i^t$ | 8: $\quad g^{t+1} = g^t + \frac{1}{n} \sum_{i=1}^n c_i^{t+1}$ |
| 9: $\quad x^{t+1} = x^t - \gamma g^t$ | 9: |
| 10: **end for** | 10: **end for** |

We start by restating the example from (Chen et al., 2020) illustrating the potential non-convergence of Clip-SGD even when full gradients are computed on clients (Clip-GD).

**Example 2.1** (Non-Convergence of Clip-GD (Chen et al., 2020)). *Let $n = 2$, $d = 1$, and $f_1(x) = \frac{1}{2}(x-3)^2$, $f_2(x) = \frac{1}{2}(x+3)^2$ in problem (1) having a unique solution $x^* = 0$. Consider Clip-GD with $\tau = 1$ applied to this problem. If for some $t_0$ we have $x^{t_0} \in [-2, 2]$ in Clip-GD, then $g^t = 0$ and $x^t = x^{t_0}$ for any $t \geq t_0$, which can be seen via direct calculations. In particular, for any $x^0 \in [-2, 2]$, the method does not move away from $x^0$.*

---

[3]The DP guarantee in Khirirat et al. (2023) relies on the condition that for some $C > 1$ and $\nu, \sigma_\omega \geq 0$, one has $\min\{\nu^2, \sigma_\omega^2\} \geq C \max\{\nu^2, \sigma_\omega^2\}$. This holds if and only if $\nu = \sigma_\omega = 0$, implying that no DP noise is added, since $\sigma_\omega^2$ denotes the variance of the DP noise.

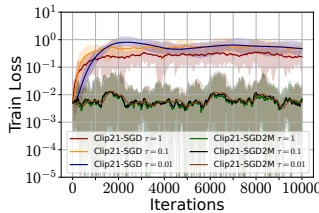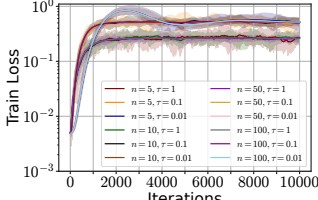

Figure 1: **Left:** behavior of stochastic `Clip21-SGD` and `Clip21-SGD2M` without DP noise (see Alg. 3) initialized at $x^0 = (0, -0.07)^\top$, with stepsize $\gamma = 1/\sqrt{T}$ where $T = 10^4$, i.e., close to the solution and small stepsize. We observe that `Clip21-SGD` escapes the good neighborhood of the solution for the problem from Theorem 2.2 with $n = 1, L = 2, \sigma = 5$, and varying $\tau \in \{1, 0.1, 0.01\}$. In contrast, `Clip21-SGD2M` remains stable around the solution. **Right:** convergence of `Clip21-SGD` does not improve with the increase of $n$ for the same problem.

To address `Clip-GD`'s non-convergence, Khirirat et al. (2023) introduce `Clip21-GD`, which applies clipping not to raw gradients but to their "shifted" differences: $\nabla f_i(x^{t+1}) - g_i^t$, where $g_i^t$ tracks the previous gradient. In the deterministic setting, this guarantees that after enough iterations, every client's difference falls below the threshold $\tau$ in norm, so clipping effectively turns off and the algorithm converges.

However, even if we replace the exact shift $g_i^t$ with the stochastic gradient itself, i.e., we use

$$x^{t+1} = x^t - \gamma g^t, g^t = \frac{1}{n} \sum_{i=1}^n g_i^t,$$
$$g_i^{t+1} = \nabla f_i(x^{t+1}) + \text{clip}_\tau(\nabla f_i(x^{t+1}, \xi_i^{t+1}) - \nabla f_i(x^{t+1})), \quad (6)$$

this "idealized" stochastic version of `Clip21-SGD` can diverge. The following theorem demonstrates non-convergence on a simple quadratic under sub-Gaussian noise.

**Theorem 2.2.** *Let $L, \sigma > 0, 0 < \gamma \leq 1/L, n = 1$. There exists a convex, $L$-smooth problem, clipping parameter $\tau < 3\sigma\sqrt{3}/10$, and an unbiased stochastic gradient satisfying Assumption 1.2 such that the method (6) is run with a stepsize $\gamma$ and clipping parameter $\tau$, then for all $x^0 \in \{(0, x_{(2)}^0) \in \mathbb{R}^2 \mid x_{(2)}^0 < 0\}$ we have*

$$\mathbb{E}\left[\|\nabla f(x^T)\|^2\right] \geq \frac{1}{2} \min\left\{\|\nabla f(x^0)\|^2, \frac{\tau^2}{45}\right\}. \quad (7)$$

*Moreover, fix $0 < \varepsilon < L/\sqrt{2}$ and $x^0 = (0, -1)^\top$. Let the sub-Gaussian variance of stochastic gradients is bounded by $\sigma^2/B$ where $B$ is a batch size. If $B < 27\sigma^2/(60\varepsilon^2)$ and $\tau \geq \varepsilon/(3\sqrt{10})$, then we have $\mathbb{E}\left[\|\nabla f(x^T)\|^2\right] > \varepsilon^2$ for all $T > 0$.*

We also illustrate the above result with simple numerical experiments reported in Figure 1. The left figure shows that `Clip21-SGD` diverges from the initial function sub-optimality level while the right one demonstrates non-improvement with the number of workers $n$ — one of the desired properties of algorithms for FL. We note that analogous reasoning applies to $\alpha$-`NormEC-SGD` (Shulgin et al., 2025a): While it enjoys similar convergence guarantees in the full-batch setting, it can fail to converge once stochastic gradient noise is used.

## 3 CLIP21-SGD2M: NEW METHOD AND THEORETICAL RESULTS

We now introduce `Clip21-SGD2M` (Alg. 3) for private distributed training and outline its key components. First, we employ client momentum with parameter $\beta$, which averages out stochastic gradient noise by exploiting momentum's variance–reduction effect (Ma & Yarats, 2018; Cutkosky & Orabona, 2019). This removes the need for the full-batch updates assumed in prior work. A central challenge in combining client-side momentum with DP, however, is that DP noise accumulates in the momentum vector; to mitigate this, we incorporate a server-side momentum that damps and smooths the noisy aggregated update. While similar double-momentum schemes have appeared in the optimization literature (Fatkhullin et al., 2024; Xu & Huang, 2022; Wang et al., 2023), to the best of our knowledge, this is the first application in a DP setting analyzed under a standard smoothness assumption. Finally, we adopt EF21-style error feedback on the client side to correct clipping-induced client drift. Since clipping acts as a contractive compressor but with input-dependent contractivity, standard EF analyses fail to apply. To overcome this, we first develop an induction-based analysis in

---

**Algorithm 3** `Clip21-SGD2M`

---

**Require:** $x^0, g^0, v^0 \in \mathbb{R}^d$ (by default $g^0 = v^0 = 0$), momentum parameters $\beta, \hat{\beta} \in (0,1]$, stepsize $\gamma > 0$, clipping parameter $\tau > 0$, DP-variance parameter $\sigma_\omega^2 \geq 0$

1: Set $g_i^0 = g^0$ and $v_i^0 = v^0$ for all $i \in [n]$
2: **for** $t = 0, \ldots, T-1$ **do**
3:     $x^{t+1} = x^t - \gamma g^t$
4:     **for** $i = 1, \ldots, n$ **do**
5:         $v_i^{t+1} = (1-\beta)v_i^t + \beta \nabla f_i(x^{t+1}, \xi_i^{t+1})$
6:         $\omega_i^{t+1} \sim \mathcal{N}(0, \sigma_\omega^2 \mathbf{I})$            only for DP version
7:         $c_i^{t+1} = \text{clip}_\tau(v_i^{t+1} - g_i^t) + \omega_i^{t+1}$
8:         $g_i^{t+1} = g_i^t + \hat{\beta}\,\text{clip}_\tau(v_i^{t+1} - g_i^t)$
9:     **end for**
10:     $g^{t+1} = g^t + \frac{\hat{\beta}}{n}\sum_{i=1}^n c_i^{t+1}$
11: **end for**

---

the deterministic regime by explicitly bounding the magnitude of the clipping input, and then extend the result to the stochastic setting using a high-probability argument that guarantees steady progress despite DP noise.

## 3.1 ANALYSIS IN THE DETERMINISTIC CASE

The next result derives a convergence rate for `Clip21-SGD2M` when $\nabla f_i(x^{t+1}, \xi_i^{t+1}) \equiv \nabla f_i(x^t)$ almost surely, i.e., Assumption 1.2 holds with $\sigma = 0$.

**Theorem 3.1** (Simplified). *Let Assumptions 1.1 and 1.2 with $\sigma = 0$ hold. Let $B :=$ $\max_i \|\nabla f_i(x^0)\| > 3\tau$ and $\Delta \geq f(x^0) - f^*$. Then, for any constant $\hat{\beta} \in (0,1]$, there exists a stepsize $\gamma \leq \min\{1/12L, \tau/12BL\}$ and momentum parameter $\beta = 4L\gamma$ such that the iterates of* `Clip21-SGD2M` *(Algorithm 3) converge with the rate*

$$\frac{1}{T}\sum_{t=0}^{T-1}\|\nabla f(x^t)\|^2 \leq \mathcal{O}\left(\frac{L\Delta(1+B/\tau)}{T}\right). \tag{8}$$

*Moreover, after at most $\frac{2B}{\hat{\beta}\tau}$ iterations, the clipping will eventually be turned off for all workers.*

*Proof sketch* The proof of Theorem 3.1 (and all subsequent theorems) relies on a carefully constructed Lyapunov function:

$$\Phi^t := \delta^t + \frac{2\gamma}{\hat{\beta}\eta n}\sum_{i=1}^n\|g_i^t - v_i^t\|^2 + \frac{8\gamma\beta}{\hat{\beta}^2\eta^2 n}\sum_{i=1}^n\|v_i^t - \nabla f_i(x^t)\|^2 + \frac{2\gamma}{\beta}\|v^t - \nabla f(x^t)\|^2, \tag{9}$$

where $\delta^t := f(x^t) - f^*$. The coefficients are calibrated so that all terms contribute on a comparable scale to $\Phi^t$. Once we establish a descent of $\Phi^t$, it follows that both the learning shift variables $\{g_i^t\}_{i=1}^n$ and the momentum buffers $\{v_i^t\}_{i=1}^n$ track the true gradients $\{\nabla f_i(x^t)\}_{i=1}^n$, thereby justifying their role in the method. The only new constant introduced is $\eta$, which captures the key technical difficulty in the proof. Through an induction argument, and with a careful choice of $\eta \sim \tau$, we establish a uniform gap bound $\|v_i^{t+1} - g_i^t\| \leq \tau/\eta$. This result allows us to regard clipping as a contractive operation on the increments $v_i^{t+1} - g_i^t$, thereby enabling a standard error-feedback analysis. The full proof is provided in Appendix E.

This theorem guarantees an $\mathcal{O}(1/T)$ convergence rate, which is known to be optimal for smooth nonconvex first-order methods (Carmon et al., 2020; 2021). Notably, like `Clip21-SGD`, `Clip21-SGD2M` also turns off clipping after finitely many iterations—once $\|v_i^{t+1} - g_i^t\| \leq \tau$. Crucially, our result holds without any bounded-heterogeneity or bounded-gradient assumptions. By contrast, even under such restrictive conditions, many prior nonconvex analyses (Liu et al., 2022; Zhang et al., 2022; Li & Chi, 2023; Allouah et al., 2024) fail to achieve an $\mathcal{O}(1/T)$ rate in the noise-free setting.

## 3.2 ANALYSIS IN THE STOCHASTIC CASE WITHOUT DP-NOISE

Next, we turn to the stochastic setting where each worker has access to local gradient estimators satisfying Assumption 1.2. First, we consider the case without DP noise, i.e., non-private training.

**Theorem 3.2** (Simplified). *Let Assumptions 1.1 and 1.2 hold and $\alpha \in (0,1)$. Let $\widetilde{B} := \max_i \|\nabla f_i(x^0)\| > 3\tau$ and $\Delta \geq \Phi^0$. Then, for any constant $\hat{\beta} \in (0,1]$, there exists a stepsize $\gamma$ and momentum parameter $\beta$ such that the iterates of `Clip21-SGD2M` (Algorithm 3) with probability at least $1 - \alpha$ are such that $\frac{1}{T}\sum_{t=0}^{T-1}\|\nabla f(x^t)\|^2$ is bounded by*

$$\widetilde{\mathcal{O}}\left(\frac{L\Delta(1+\widetilde{B}/\tau)}{T} + \frac{\sigma(\sqrt{L\Delta}+\widetilde{B}+\sigma)}{\sqrt{T}n}\right) \tag{10}$$

*where $\widetilde{\mathcal{O}}$ hides constant and polylogarithmic factors, and higher order terms that decrease in $T$.*

*Proof sketch.* The proof follows the same overall structure as Theorem 3.1, but with the key complication that the increments $v_i^{t+1} - g_i^t$ are now random and can, in principle, grow without bound under Assumption 1.2. To handle this, we switch to a high-probability argument: by inductively showing that, with a large probability, each $v_i^{t+1} - g_i^t$ stays below a fixed threshold, we recover a contractive property of the clipping operator on these random vectors. The remainder of the proof then mirrors the deterministic case, augmented by careful martingale-difference concentration bounds; see Appendix H for full details. This result demonstrates that `Clip21-SGD2M` achieves an optimal $\mathcal{O}(1/\sqrt{nT})$ (Arjevani et al., 2023) rate in the stochastic setting. In contrast to the previous works establishing similar rates (Liu et al., 2022; Noble et al., 2022; Allouah et al., 2024), our result does not rely on the boundedness of the gradients or data heterogeneity. Moreover, when $\sigma = 0$ (no stochastic noise), the rate from (10) becomes $\mathcal{O}(1/T)$, recovering the one given by Theorem 3.1.

### 3.3 ANALYSIS IN THE STOCHASTIC CASE WITH DP-NOISE

Finally, we provide the convergence result for `Clip21-SGD2M` with DP-noise.

**Theorem 3.3.** *Let Assumptions 1.1 and 1.2 hold and $\alpha \in (0,1)$. Let $\Delta \geq \Phi^0$. Then, there exists a stepsize $\gamma$ and momentum parameters $\beta, \hat{\beta}$ such that the iterates of `Clip21-SGD2M` (Algorithm 3) with the DP-noise variance $\sigma_\omega^2$ with probability at least $1 - \alpha$ are such that $\frac{1}{T}\sum_{t=0}^{T-1}\|\nabla f(x^t)\|^2$ is bounded by*

$$\widetilde{\mathcal{O}}\left(\left(\frac{L\Delta\sigma d\sigma_\omega^2\widetilde{B}^2}{(nT)^{3/2}\tau^2}\left(\sqrt{L\Delta}+\widetilde{B}+\sigma\right)\right)^{1/3} + \left(\frac{\sqrt{L\Delta d}\sigma_\omega}{\tau\sqrt{nT}} + \frac{\sqrt{L\Delta}d^{1/3}\sigma_\omega^{2/3}}{\tau^{2/3}(Tn)^{1/3}}\right)\left(\sqrt{L\Delta}+\widetilde{B}+\sigma\right)\right), \tag{11}$$

*where $\widetilde{\mathcal{O}}$ hides constant and polylogarithmic factors, and higher order terms decreasing in $T$.*

In the special case of local Differential Privacy, the noise level has to be chosen in a specific way. In this setting, we obtain the following privacy-utility trade-off.

**Corollary 3.4.** *Let Assumptions 1.1 and 1.2 hold and $\alpha \in (0,1)$. Let $\Delta \geq \Phi^0$ and $\sigma_\omega$ be chosen as $\sigma_\omega = \Theta\left(\frac{\tau}{\varepsilon}\sqrt{T\log\left(\frac{T}{\delta}\right)\log\left(\frac{1}{\delta}\right)}\right)$ for some $\varepsilon, \delta \in (0,1)$. Then there exists a stepsize $\gamma$ and momentum parameters $\beta, \hat{\beta}$ such that the iterates of `Clip21-SGD2M` (Algorithm 3) with probability at least $1 - \alpha$ satisfy local $(\varepsilon, \delta)$-DP and*

$$\frac{1}{T}\sum_{t=0}^{T-1}\|\nabla f(x^t)\|^2 \leq \widetilde{\mathcal{O}}\left(\sqrt{L\Delta}\left(\frac{\sqrt{d}}{\sqrt{n}\varepsilon} + \left(\frac{\sqrt{d}}{\sqrt{n}\varepsilon}\right)^{2/3}\right)\left(\sqrt{L\Delta}+\widetilde{B}+\sigma\right)\right), \tag{12}$$

*where $\widetilde{\mathcal{O}}$ hides constant and polylogarithmic factors, and terms decreasing in $T$.*

The proof of the above result is provided in Appendix G. Disregarding dependencies on polylogarithmic factors, $L\Delta$, $\widetilde{B}$, and $\sigma$, the derived utility bound simplifies to $\widetilde{\mathcal{O}}\left(\sqrt{d}/(\sqrt{n}\varepsilon) + \left(\sqrt{d}/(\sqrt{n}\varepsilon)\right)^{2/3}\right)$. When $\sqrt{d}/\sqrt{n}\varepsilon > 1$— which is common in modern models where $d$ is at least hundreds of millions and far exceeds the number of clients $n$ (Charles et al., 2024; Chua et al., 2024)—the first term in (12) dominates, yielding a rate that matches the best-known non-convex utility bounds (Allouah et al., 2023). However, when $\sqrt{d}/(\sqrt{n}\varepsilon) < 1$, our bound is less favorable. The tightness of this bound under the general assumptions considered in this work remains an open question.

A key limitation of our DP guarantee is its incompatibility with privacy amplification by subsampling. This arises from the client-side computation of vectors $v_i^{t+1}$ and $g_i^{t+1}$, which accumulate private information over multiple iterations. These components are essential for our method to handle data heterogeneity (through $g_i^{t+1}$) and to reduce stochastic noise (through $v_i^{t+1}$). In contrast,

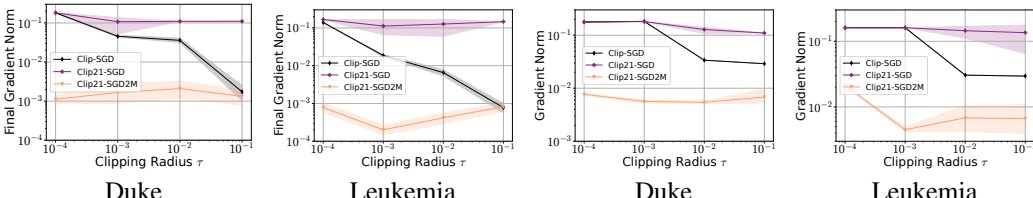

| Duke | Leukemia | Duke | Leukemia |

Figure 2: Comparison of `Clip-SGD`, `Clip21-SGD`, and `Clip21-SGD2M` on logistic regression with non-convex regularization for various clipping radii $\tau$ with mini-batch (**two left**) and Gaussian-added (**two right**) stochastic gradients. The final gradient norm is averaged over the last 100 iterations. The gradient norm dynamics are reported in Figure I.1.

many existing methods benefit from this amplification, as illustrated by `Clip-SGD` (Abadi et al., 2016), which achieves a smaller DP-noise parameter $\sigma_\omega = \Theta\left((q\tau/\varepsilon)\sqrt{T\log(1/\delta)}\right)$, where $q$ is the sampling probability for each individual data point. However, these methods typically rely on restrictive assumptions such as bounded data heterogeneity, as discussed in Section 1.2. Achieving both privacy amplification by sub-sampling and provable convergence without such limiting assumptions remains an open challenge. Despite these limitations, our experimental results indicate that `Clip21-SGD2M` achieves a privacy-utility trade-off comparable to `Clip21-SGD`.

## 4 EXPERIMENTS

In this section, we provide an empirical evaluation of the proposed algorithm against baselines such as `Clip21-SGD` (Khrirat et al., 2023), $\alpha$-`NormEC-SGD` (Shulgin et al., 2025a), and `Clip-SGD`, where the latter is considered as the method of choice in private training.

First, we test the convergence of `Clip-SGD`, `Clip21-SGD`, and the proposed `Clip21-SGD2M` algorithms with stochastic gradients for various clipping radii $\tau$ on several workloads. These results demonstrate the significance of using the momentum technique to achieve better performance.

**Non-convex Logistic Regression.** In this experiment, we assess each algorithm using only stochastic gradients—either by adding Gaussian noise to the full local gradient $\nabla f_i(x)$ or by sampling mini-batches—without any additional DP noise. We focus on logistic regression with a non-convex regularize, $f_i(x) = \frac{1}{m}\sum_{j=1}^{m}\log(1 + \exp(-b_{ij}a_{ij}^\top x)) + \lambda\sum_{l=1}^{d}\frac{x_l^2}{1+x_l^2}$, on the Duke and Leukemia datasets (Chang & Lin, 2011), a setup used in prior work (Khrirat et al., 2023; Li & Chi, 2023). We fix $\hat{\beta}$ (no DP noise), and full tuning details appear in Appendix I.1. Figure 2 plots the average gradient norm over the final 100 iterations, aggregated across three runs, for a range of clipping radii $\tau$ `Clip21-SGD2M` consistently matches or outperforms the other methods—especially at small $\tau$—demonstrating its robustness to the choice of clipping threshold and aligning with our theoretical guarantees. Furthermore, the convergence curves in Figure I.1 show that `Clip21-SGD2M` reaches optimality faster than its competitors.

**Training Resnet20 and VGG16.** We next evaluate our methods on training ResNet-20 (He et al., 2016) and VGG-16 (Simonyan & Zisserman, 2014) models on CIFAR-10 (Krizhevsky et al., 2009)[4]. Results, averaged over three random seeds, appear in Figure 3 (global clipping across all weights) and Figure I.2 (layer-wise clipping). As before, we set $\hat{\beta} = 1$ for `Clip21-SGD2M` due to the absence of DP noise. The detailed experiment description is provided in Appendix I.2.1.

We report both test accuracy and training loss at the end of training. `Clip-SGD`'s performance degrades steadily as the clipping radius $\tau$ shrinks, whereas both `Clip21-SGD` and `Clip21-SGD2M` remain much more stable. In particular, for small $\tau$, `Clip21-SGD2M` outperforms `Clip21-SGD`, achieving lower training loss and higher test accuracy—empirical findings that further validate our theoretical predictions. Full training curves are given in Figures I.3–I.4 for VGG-16 and Figures I.5–I.6 for ResNet-20.

**Adding Gaussian Noise for DP.** In our second experimental suite, we evaluate Gaussian-DP variants of the optimizers on MLP and CNN architectures using the MNIST dataset (Deng, 2012).

---

[4]Our implementation is based on the open-source code of (Horváth & Richtárik, 2020) with minor adjustments.

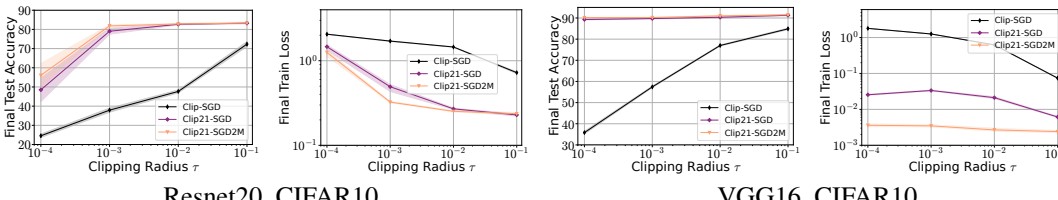

Resnet20, CIFAR10 — VGG16, CIFAR10

Figure 3: Comparison of `Clip-SGD`, `Clip21-SGD`, and `Clip21-SGD2M` when training Resnet20 (**two left**) and VGG16 (**two right**) models on CIFAR10 dataset where the clipping is applied globally. The train loss and test accuracy dynamics are reported in Figure I.3 and Figure I.5.

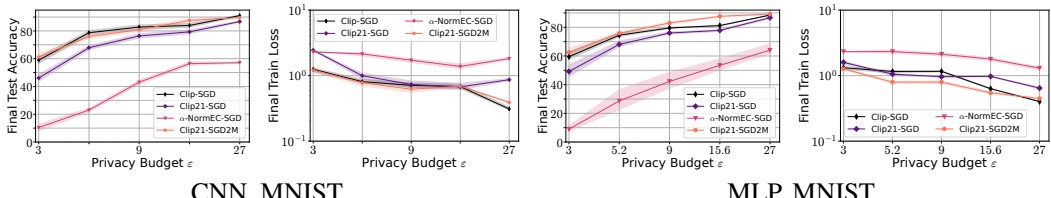

CNN, MNIST — MLP, MNIST

Figure 4: Comparison of `Clip-SGD`, `Clip21-SGD`, and `Clip21-SGD2M` when training CNN (**two left**) and MLP (**two right**) models on MNIST dataset, varying the privacy budget $\varepsilon$ where the clipping is applied globally. The training loss and test accuracy dynamics are presented in Figures I.7 to I.10.

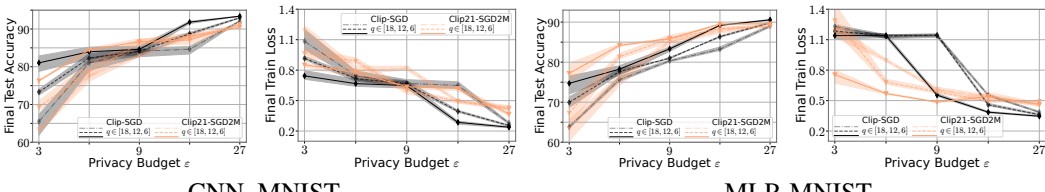

CNN, MNIST — MLP, MNIST

Figure 5: Comparison of `Clip-SGD` and `Clip21-SGD2M` when training CNN (**two left**) and MLP (**two right**) models on MNIST dataset, varying the privacy budget $\varepsilon$ and number of sampled clients $|S_t|$, where the clipping is applied globally.

We compare `Clip-SGD`, `Clip21-SGD`, $\alpha$-`NormEC`, and `Clip21-SGD2M` across privacy budgets $\varepsilon \in \{3, 5.2, 9, 15.6, 27\}$ (with $\delta = 10^{-3}$). The data are split into $n = 25$ equal shards, and each method is run for $T = 150$ epochs with batch size 64 and 3 random seeds. Full experimental details are reported in Appendix I.2.2. As shown in Figure 4, `Clip21-SGD2M` achieves competitive performance: it slightly outperforms `Clip-SGD` on the MLP and matches it on the CNN, further corroborating our theoretical results. We report the training dynamics in Figures I.7 to I.10. To remain consistent with our analysis (where we assume $\sigma$-sub-Gaussian gradient noise), we do not consider amplification by client sub-sampling in the experiments.

**Partial Client Participation.** Although our current theory does not cover partial client participation, our experiments in Figure 5 indicate that `Clip21-SGD2M` benefits from privacy amplification via client sub-sampling. In this variant, the server updates $g^t$ (line 10) using only $\{c_i^{t+1}\}_{i \in S_t}$ from the sampled set $S_t$ (see Appendix A for more details). We train CNN and MLP models on MNIST dataset following the previous setup, varying the number of sampled clients $|S_t| \in \{6, 12, 18\}$ with $n = 24$. We observe that the performance of `Clip21-SGD2M` is competitive with that of `Clip-SGD`.

## 5 CONCLUSION AND FUTURE WORK

In this work, we introduced `Clip21-SGD2M`, a method achieving optimal convergence rates and strong privacy-utility trade-offs without assuming bounded gradients or data heterogeneity. Several promising extensions remain open, including: $(i)$ improving the DP neighborhood and enabling privacy amplification by sub-ampling (see Section 3.3); $(ii)$ generalizing the analysis to handle heavy-tailed noise; $(iii)$ developing `AdaGrad`/`Adam`-type variants for improved deep learning performance (Streeter & McMahan, 2010; Duchi et al., 2011; Kingma & Ba, 2014); and $(iv)$ extending the analysis to settings with generalized smoothness (Zhang et al., 2020b).

## REPRODUCIBILITY STATEMENT

All experiments utilize publicly available datasets, cited accordingly. We provide the implementation of our algorithms in the supplementary, while the training details are listed in the appendix.

## ETHICS STATEMENT

This paper presents work whose goal is to advance the field of Machine Learning. There are many potential societal consequences of our work, none of which we feel must be specifically highlighted here.

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

# Appendix

C­ONTENTS

## A  E­XTENSION TO P­ARTIAL P­ARTICIPATION S­ETTING

In this section, we provide a more detailed discussion of the extension of `Clip21-SGD2M` when the server samples only a subset $S_t$ of clients at each communication round.

The algorithm design in this case is outlined in Alg. 4. There are two main changes in the algorithm design.

1. Only clients sampled in $S_t$ execute steps in lines 6–9; unsampled clients remain idle.

2. The server uses the updates $\{c_i^{t+1}\}_{i \in S_t}$ from the sampled clients only.

This variation of `Clip21-SGD2M` benefits from amplification by sub-sampling similar to `Clip-SGD`.

---

**Algorithm 4** `Clip21-SGD2M` with partial participation

---

**Require:** $x^0, g^0, v^0 \in \mathbb{R}^d$ (by default $g^0 = v^0 = 0$), momentum parameters $\beta, \hat{\beta} \in (0,1]$, stepsize $\gamma > 0$, clipping parameter $\tau > 0$, number of sampled clients $s$, DP-variance parameter $\sigma_\omega^2 \geq 0$

1: Set $g_i^0 = g^0$ and $v_i^0 = v^0$ for all $i \in [n]$
2: **for** $t = 0, \dots, T-1$ **do**
3:   $x^{t+1} = x^t - \gamma g^t$
4:   sample $S_t \subseteq [n]$ such that $|S_t| = s$
5:   **for** $i \in S_t$ **do**
6:     $v_i^{t+1} = (1-\beta)v_i^t + \beta \nabla f_i(x^{t+1}, \xi_i^{t+1})$
7:     $\omega_i^{t+1} \sim \mathcal{N}(0, \sigma_\omega^2 \mathbf{I})$         only for DP version
8:     $c_i^{t+1} = \mathrm{clip}_\tau(v_i^{t+1} - g_i^t) + \omega_i^{t+1}$
9:     $g_i^{t+1} = g_i^t + \hat{\beta}\,\mathrm{clip}_\tau(v_i^{t+1} - g_i^t)$
10:   **end for**
11:   **for** $i \notin S_t$ **do**
12:     $v_i^{t+1} = v_i^t$
13:     $g_i^{t+1} = g_i^t$
14:   **end for**
15:   $g^{t+1} = g^t + \frac{\hat{\beta}}{s} \sum_{i \in S_t}^n c_i^{t+1}$
16: **end for**

---

# B  NOTATION

For brevity, in all proofs, we use the following notation

$$\delta^t := f(x^t) - f^*, \quad \widetilde{V}^t := \frac{1}{n}\sum_{i=1}^n \|g_i^t - v_i^t\|^2,$$

$$\widetilde{P}^t := \frac{1}{n}\sum_{i=1}^n \|v_i^t - \nabla f_i(x^t)\|^2, \quad P^t := \|v^t - \nabla f(x^t)\|^2,$$

$$R^t := \|x^{t+1} - x^t\|^2.$$

We additionally denote $\eta_i^t := \frac{\tau}{\|v_i^t - g_i^{t-1}\|}$ and $\eta := \frac{\tau}{B}$ where $B$ is defined in each section (it is different in deterministic and stochastic settings). Besides, we define $\mathcal{I}_t := \{i \in [n] \mid \|v_i^t - g_i^{t-1}\| > \tau\}$.

We denote $\theta_i^t := \nabla f_i(x^t, \xi_i^t) - \nabla f_i(x^t)$. From Assumption 1.2, we have that $\theta_i^t$ is zero-centered $\sigma$-sub-Gaussian random vector conditioned at $x^t$, namely

$$\mathbb{E}\left[\theta_i^t \mid x^t\right] = 0, \quad \mathbb{E}\left[\exp\left(\frac{\|\theta_i^t\|^2}{\sigma^2}\right) \mid x^t\right] \leq \exp(1), \tag{13}$$

which is equivalent to

$$\Pr(\|\theta_i^t\| > b) \leq 2\exp\left(-\frac{b^2}{2\sigma^2}\right) \quad \forall b > 0 \tag{14}$$

up to the numerical factor in $\sigma$ (Vershynin, 2018). Moreover, we define an average of $\theta_i^t$ as $\theta^t := \frac{1}{n}\sum_{i=1}^n \theta_i^t$, an average of $\omega_i^t$ as $\Omega^t = \frac{1}{n}\sum_{l=1}^t\sum_{i=1}^n \omega_i^l$, and an average of $g_i^t$ as $\overline{g}^t = \frac{1}{n}\sum_{i=1}^n g_i^t$. Thus, we have the following relation between $g^t$ and $\overline{g}^t$:

$$g^t = \overline{g}^t + \hat{\beta}\Omega^t. \tag{15}$$

Indeed, it is true at iteration $0$ by the initialization. Let us assume that it holds at iteration $t$, then we have

$$g^{t+1} = g^t + \frac{\hat{\beta}}{n}\sum_{i=1}^n(\mathrm{clip}_\tau(v_i^{t+1} - g_i^t) + \omega_i^{t+1}) = \overline{g}^t + \hat{\beta}\Omega^t + \frac{\hat{\beta}}{n}\sum_{i=1}^n(\mathrm{clip}_\tau(v_i^{t+1} - g_i^t) + \omega_i^{t+1}) = \overline{g}^{t+1} + \hat{\beta}\Omega^{t+1},$$

i.e., it holds at iteration $t+1$ as well.

## C  USEFUL LEMMAS

**Lemma C.1** (Lemma C.3 in (Gorbunov et al., 2019)). *Let $\{\xi_k\}_{k=1}^N$ be the sequence of random vectors with values in $\mathbb{R}^n$ such that*

$$\mathbb{E}\left[\xi_k \mid \xi_{k-1}, \dots, \xi_1\right] = 0 \text{ almost surely, } \forall k \in \{1, \dots, N\},$$

*and set $S_N := \sum_{k=1}^N \xi_k$. Assume that the sequence $\{\xi_k\}_{k=1}^N$ are sub-Gaussian, i.e.*

$$\mathbb{E}\left[\exp\left(\|\xi_k\|^2/\sigma_k^2 \mid \xi_{k-1}, \dots, \xi_1\right)\right] \leq \exp(1) \text{ almost surely, } \forall k \in \{1, \dots, N\},$$

*where $\sigma_2, \dots, \sigma_N$ are some positive numbers. Then for all $\gamma \geq 0$*

$$\Pr\left(\|S_N\| \geq (\sqrt{2} + 2\gamma)\sqrt{\sum_{k=1}^N \sigma_k^2}\right) \leq \exp(-\gamma^2/3). \tag{16}$$

**Lemma C.2.** *Let $f$ be $L$-smooth, $\delta^t = f(x^t) - f^*$, $\{x^t\}$ be generated by Algorithm 3, and the stepsize $\gamma \leq \frac{1}{2L}$. Then*

$$\delta^{t+1} \leq \delta^t - \frac{\gamma}{2}\|\nabla f(x^t)\|^2 - \frac{1}{4\gamma}\|x^t - x^{t+1}\|^2 + 2\gamma\|\nabla f(x^t) - v^t\|^2$$

$$+ \frac{2\gamma}{n}\sum_{i=1}^n \|g_i^t - v_i^t\|^2 + \gamma\hat{\beta}^2\|\Omega^t\|^2. \tag{17}$$

*Proof.* Using $L$-smoothness of $f$ we have

$$f(x^{t+1}) \overset{(i)}{\leq} f(x^t) + \langle \nabla f(x^t), x^{t+1} - x^t \rangle + \frac{L}{2}\|x^{t+1} - x^t\|^2$$

$$\overset{(ii)}{=} f(x^t) - \gamma\langle \nabla f(x^t), g^t \rangle + \frac{L\gamma^2}{2}\|g^t\|^2$$

$$\overset{(iii)}{=} f(x^t) - \frac{\gamma}{2}\left(\|\nabla f(x^t)\|^2 + \|g^t\|^2 - \|\nabla f(x^t) - g^t\|^2\right) + \frac{L\gamma^2}{2}\|g^t\|^2$$

$$= f(x^t) - \frac{\gamma}{2}\|\nabla f(x^t)\|^2 - \frac{\gamma}{2}\|g^t\|^2(1 - L\gamma) + \frac{\gamma}{2}\|\nabla f(x^t) - g^t\|^2$$

$$\overset{(iv)}{\leq} f(x^t) - \frac{\gamma}{2}\|\nabla f(x^t)\|^2 - \frac{\gamma}{4}\|g^t\|^2 + \frac{\gamma}{2}\|\nabla f(x^t) - g^t\|^2. \tag{18}$$

where $(i)$ follows from smoothness; $(ii)$ from the update rule $(iii)$ from $\|a - b\|^2 = \|a\|^2 + \|b\|^2 - 2\langle a, b \rangle$; $(iv)$ from the stepsize restriction $\gamma \leq \frac{1}{2L}$. Using (15) we continue as follows

$$f(x^{t+1}) \leq f(x^t) - \frac{\gamma}{2}\|\nabla f(x^t)\|^2 - \frac{\gamma}{4}\|g^t\|^2 + \gamma\|\nabla f(x^t) - \overline{g}^t\|^2 + \gamma\hat{\beta}^2\|\Omega^t\|^2$$

$$\overset{(i)}{\leq} f(x^t) - \frac{\gamma}{2}\|\nabla f(x^t)\|^2 - \frac{\gamma}{4}\|g^t\|^2 + 2\gamma\|\nabla f(x^t) - v^t\|^2 + 2\gamma\|\overline{g}^t - v^t\|^2 + \gamma\hat{\beta}^2\|\Omega^t\|^2$$

$$\overset{(ii)}{\leq} f(x^t) - \frac{\gamma}{2}\|\nabla f(x^t)\|^2 - \frac{\gamma}{4}\|g^t\|^2 + 2\gamma\|\nabla f(x^t) - v^t\|^2 + \frac{2\gamma}{n}\sum_{i=1}^n \|g_i^t - v_i^t\|^2 + \gamma\hat{\beta}^2\|\Omega^t\|^2, \tag{19}$$

where $(i$-$ii)$ follow from Young's inequality. It remains to subtract $f^*$ from both sides. It remains to replace $g^t$ by $\frac{1}{\gamma}(x^t - x^{t+1})$

$\square$

**Lemma C.3** (Lemma 4.1 in (Khirirat et al., 2023)). *The clipping operator satisfies for any $x \in \mathbb{R}^d$*

$$\|\text{clip}_\tau(x) - x\| \leq \max\{\|x\| - \tau, 0\}. \tag{20}$$

**Lemma C.4** (Property of smooth functions). *Let $\phi: \mathbb{R}^d \to \mathbb{R}$ be $L$-smooth and lower bounded by $\phi^* \in \mathbb{R}$, i.e. $\phi(x) \geq \phi^*$ for any $x \in \mathbb{R}^d$. Then we have*

$$\|\nabla\phi(x)\|^2 \leq 2L(\phi(x) - \phi^*). \tag{21}$$

*Proof.* It is a standard property of smooth functions. We refer to Theorem 4.23 of (Orabona, 2019).

$\square$

## D  PROOF OF THEOREM 2.2 (NON-CONVERGENCE OF CLIP21-SGD)

*Proof.* **The case** $n = 1$. Let us consider the problem $f(x) = \frac{L}{2}\|x\|^2$. Let vectors $\{z_j\}_{j=1}^3$ be defined as

$$z_1 = \begin{pmatrix} 3 \\ 0 \end{pmatrix} \sqrt{\frac{3\sigma^2}{100}}, \quad z_2 = \begin{pmatrix} 0 \\ 4 \end{pmatrix} \sqrt{\frac{3\sigma^2}{100}}, \quad z_1 = \begin{pmatrix} -3 \\ -4 \end{pmatrix} \sqrt{\frac{3\sigma^2}{100}}.$$

Note that we have

$$\|z_1\|^2 = \frac{27\sigma^2}{100}, \quad \|z_2\|^2 = \frac{24\sigma^2}{50}, \quad \|z_3\|^2 = \frac{3\sigma^2}{4},$$

meaning that $\tau < \|z_i\|$ for all $i \in [3]$. We define the stochastic gradient as $\nabla f(x^t, \xi^t) = \nabla f(x^t) + \xi^t = Lx^t + \xi^t$ where $\xi^t$ is picked uniformly at random from $\{z_1, z_2, z_3\}$. Simple calculations verify that Assumption 1.2 holds for such noise. Next, the update rule of the method (6) in the case $n = 1$ is

$$x^{t+1} = x^t - \gamma g^t = x^t - \gamma(\nabla f(x^t) + \text{clip}_\tau(\nabla f(x^t, \xi^t) - \nabla f(x^t))) = x^t - L\gamma x^t - \gamma\, \text{clip}_\tau(\xi^t).$$

Since $\tau < \|z_i\|$ for any $i \in \{1, 2, 3\}$ clipping is always active and we have

$$\begin{aligned}
\mathbb{E}\left[\text{clip}_\tau(\xi^t)\right] &= \frac{1}{3}\text{clip}_\tau(z_1) + \frac{1}{3}\text{clip}_\tau(z_2) + \frac{1}{3}\text{clip}_\tau(z_3) \\
&= \frac{1}{3}\frac{\tau}{\|z_1\|}z_1 + \frac{1}{3}\frac{\tau}{\|z_2\|}z_2 + \frac{1}{3}\frac{\tau}{\|z_3\|}z_3 \\
&= \frac{1}{3}\frac{\tau}{\frac{3\sqrt{3}\sigma}{10}}\frac{\sigma\sqrt{3}}{10}\begin{pmatrix} 3 \\ 0 \end{pmatrix} + \frac{1}{3}\frac{\tau}{\frac{4\sqrt{3}\sigma}{10}}\frac{\sigma\sqrt{3}}{10}\begin{pmatrix} 0 \\ 4 \end{pmatrix} + \frac{1}{3}\frac{\tau}{\frac{5\sqrt{3}\sigma}{10}}\frac{\sigma\sqrt{3}}{10}\begin{pmatrix} -3 \\ -4 \end{pmatrix} \\
&= \frac{\tau}{9}\begin{pmatrix} 3 \\ 0 \end{pmatrix} + \frac{\tau}{12}\begin{pmatrix} 0 \\ 4 \end{pmatrix} + \frac{\tau}{15}\begin{pmatrix} -3 \\ -4 \end{pmatrix} \\
&= \underbrace{\frac{\tau}{15}\begin{pmatrix} 2 \\ 1 \end{pmatrix}}_{:=h}.
\end{aligned}$$

Thus, we obtain

$$\begin{aligned}
\mathbb{E}\left[x^T\right] &= (1 - L\gamma)\mathbb{E}\left[x^{T-1}\right] - \gamma\mathbb{E}\left[\text{clip}_\tau(\xi^t)\right] \\
&= (1 - L\gamma)\mathbb{E}\left[x^{T-1}\right] - \gamma h \\
&= (1 - L\gamma)^T x^0 - \gamma h \sum_{t=0}^{T-1}(1 - L\gamma)^{T-1-t} \\
&= (1 - L\gamma)^T \begin{pmatrix} 0 \\ x_{(2)}^0 \end{pmatrix} - \frac{\tau\gamma}{15}\begin{pmatrix} 2 \\ 1 \end{pmatrix}\frac{1 - (1 - L\gamma)^T}{1 - (1 - L\gamma)} \\
&= (1 - L\gamma)^T \begin{pmatrix} 0 \\ x_{(2)}^0 \end{pmatrix} - \frac{\tau}{15L}\begin{pmatrix} 2 \\ 1 \end{pmatrix}(1 - (1 - L\gamma)^T).
\end{aligned}$$

Therefore, since $x_{(2)}^0 < 0$ we have

$$\begin{aligned}
\mathbb{E}\left[\|\nabla f(x^T)\|^2\right] &= \mathbb{E}\left[\|Lx^T\|^2\right] \\
&= \left\|\mathbb{E}\left[Lx^T\right]\right\|^2 + \mathbb{E}\left[\|Lx^T - \mathbb{E}\left[Lx^T\right]\|^2\right] \\
&\geq \left\|\mathbb{E}\left[Lx^T\right]\right\|^2 \\
&= \frac{4\tau^2}{165}\left(1 - (1 - L\gamma)^T\right)^2 + L^2\left((1 - L\gamma)^T x_{(2)}^0 - \frac{\tau}{15L}\left(1 - (1 - L\gamma)^T\right)\right)^2 \\
&\geq \frac{4\tau^2}{165}\left(1 - (1 - L\gamma)^T\right)^2 + (1 - L\gamma)^{2T}\|Lx^0\|^2 + \frac{\tau^2}{165}(1 - (1 - L\gamma)^T)^2 \\
&= \frac{\tau^2}{45}\left(1 - (1 - L\gamma)^T\right)^2 + (1 - L\gamma)^{2T}\|\nabla f(x^0)\|^2.
\end{aligned}$$

Note that the function $a(1-x)^2 + x^2 b \geq \frac{ab}{a+b}$. Applying this result for $a = \frac{\tau^2}{45}, b = \|\nabla f(x^0)\|^2$, and $x = (1 - L\gamma)^T$ we get

$$\mathbb{E}\left[\|\nabla f(x^T)\|^2\right] \geq \frac{\frac{\tau^2}{45}\|\nabla f(x^0)\|^2}{\frac{\tau^2}{45} + \|\nabla f(x^0)\|^2} \geq \frac{1}{2} \min\left\{\|\nabla f(x^0)\|^2, \frac{\tau^2}{45}\right\}.$$

**The case $n > 1$.** If $n > 1$ then we can consider a similar example where each client is quadratic $\frac{L}{2}\|x\|^2$ and the stochastic gradient is constructed as $\nabla f_i(x^t, \xi_i^t) = \nabla f_i(x^t) + \xi_i^t = Lx^t + \xi_i^t$ where $\xi_i^t$ is sampled uniformly at random from vectors $\{z_1, z_2, z_3\}$ such that

$$z_1 = \binom{3}{0}\sqrt{\frac{3\sigma^2}{100B}}, \quad z_2 = \binom{0}{4}\sqrt{\frac{3\sigma^2}{100B}}, \quad z_1 = \binom{-3}{-4}\sqrt{\frac{3\sigma^2}{100B}}.$$

Then, Assumption 1.2 is satisfied with $\sigma^2/B$. Therefore, if $x_{(2)}^0 = -1$, $\varepsilon < \frac{L}{\sqrt{2}}$, and $\tau \geq \frac{\varepsilon}{3\sqrt{10}}$, this implies that $B \leq \frac{243\sigma^2}{5\varepsilon^2} < \frac{27\sigma^2}{50\tau^2}$, and

$$\mathbb{E}\left[\|\nabla f(x^T)\|^2\right] \geq \frac{1}{2} \min\left\{\|\nabla f(x^0)\|^2, \frac{\tau^2}{45}\right\} \geq \varepsilon^2.$$

$\square$

# E    PROOF OF THEOREM 3.1 (CONVERGENCE OF Clip21-SGD2M IN FULL-BATCH SETTING)

As we mention in the main part of the paper, the proofs are induction-based: by induction, we show that several quantities remain bounded throughout the work of the method. That is, in Lemmas E.1-E.7, we establish several useful bounds and recurrences. These lemmas allow us to use the contraction-like property (Lemma C.3) of the clipping operator and finish the proof of Theorem 3.1 applying similar techniques used in the analysis of EF21.

**Lemma E.1.** *Let each $f_i$ be $L$-smooth. Then, the iterates generated by* Clip21-SGD2M *with* $\nabla f_i(x^{t+1}, \xi_i^{t+1}) = \nabla f_i(x^{t+1})$ *(full gradients) and $\sigma_\omega = 0$ (no DP-noise) satisfy the following inequality*

$$\|v_i^{t+1} - g_i^t\| \leq (1 - \hat{\beta})\|v_i^t - g_i^{t-1}\| + \hat{\beta}\max\{0, \|v_i^t - g_i^{t-1}\| - \tau\} + L\gamma\beta\|g^t\| \qquad (22)$$
$$+ \beta\|\nabla f_i(x^t) - v_i^t\|.$$

*Proof.* We have

$$\|v_i^{t+1} - g_i^t\| \overset{(i)}{=} \|(1 - \beta)v_i^t + \beta\nabla f_i(x^{t+1}) - g_i^t\|$$

$$\overset{(ii)}{\leq} \|v_i^t - g_i^t\| + \beta\|\nabla f_i(x^{t+1}) - v_i^t\|$$

$$\overset{(iii)}{=} \|v_i^t - g_i^{t-1} - \hat{\beta}\operatorname{clip}_\tau(v_i^t - g_i^{t-1})\| + \beta\|\nabla f_i(x^{t+1}) - \nabla f_i(x^t)\| + \beta\|\nabla f_i(x^t) - v_i^t\|$$

$$\overset{(iv)}{\leq} (1 - \hat{\beta})\|v_i^t - g_i^{t-1}\| + \hat{\beta}\|v_i^t - g_i^{t-1} - \operatorname{clip}_\tau(v_i^t - g_i^{t-1})\| + L\gamma\beta\|g^t\| + \beta\|\nabla f_i(x^t) - v_i^t\|$$

$$\overset{(v)}{\leq} (1 - \hat{\beta})\|v_i^t - g_i^{t-1}\| + \hat{\beta}\max\{0, \|v_i^t - g_i^{t-1}\| - \tau\} + L\gamma\beta\|g^t\| + \beta\|\nabla f_i(x^t) - v_i^t\|.$$

where $(i)$ follows from the update rule of $v_i^t$ in deterministic case, $(ii)$ from triangle inequality, $(iii)$ from the update rule of $g_i^t$, $(iv)$ from triangle inequality, update rule of $x^t$, and $L$-smoothness, $(v)$ properties of clipping from Lemma C.3. $\square$

**Lemma E.2.** *Let each $f_i$ be $L$-smooth, $\Delta \geq \Phi^0$, and $B > \tau$. Assume that the following inequalities hold for the iterates generated by* Clip21-SGD2M *with* $\nabla f_i(x^{t+1}, \xi_i^{t+1}) = \nabla f_i(x^{t+1})$ *(full gradients) and $\sigma_\omega = 0$ (no DP-noise)*

1. $\|g^{t-1}\| \leq \sqrt{64L\Delta} + 3(B - \tau);$

2. $\|\nabla f_i(x^{t-1}) - v_i^{t-1}\| \le \sqrt{4L\Delta} + \frac{3}{2}(B - \tau)$;

3. $\|v_i^t - g_i^{t-1}\| \le B \ \forall i \in [n]$;

4. $\gamma \le \frac{1}{12L}$;

5. $\hat{\beta}, \beta \in [0, 1]$;

6. $\Phi^t \le \Delta$.

*Then we have*

$$\|g^t\| \le \sqrt{64L\Delta} + 3(B - \tau). \tag{23}$$

*Proof.* We have

$$\|g^t\|$$

$$\overset{(i)}{=} \left\| g^{t-1} + \frac{\hat{\beta}}{n} \sum_{i=1}^n \mathrm{clip}_\tau(v_i^t - g_i^{t-1}) \right\|$$

$$= \left\| g^{t-1} + \hat{\beta}(v^t - g^{t-1}) + \frac{\hat{\beta}}{n} \sum_{i=1}^n \left( \mathrm{clip}_\tau(v_i^t - g_i^{t-1}) - (v_i^t - g_i^{t-1}) \right) \right\|$$

$$= \left\| (1 - \hat{\beta})g^{t-1} + \hat{\beta}\nabla f(x^t) + \hat{\beta}(v^t - \nabla f(x^t)) + \frac{\hat{\beta}}{n} \sum_{i=1}^n \left( \mathrm{clip}_\tau(v_i^t - g_i^{t-1}) - (v_i^t - g_i^{t-1}) \right) \right\|$$

$$\overset{(ii)}{\le} (1 - \hat{\beta})\|g^{t-1}\| + \hat{\beta}\|\nabla f(x^t)\| + \frac{\hat{\beta}}{n} \sum_{i=1}^n \|v_i^t - \nabla f_i(x^t)\| + \frac{\hat{\beta}}{n} \sum_{i=1}^n \max\left\{ 0, \|v_i^t - g_i^{t-1}\| - \tau \right\},$$

where $(i)$ follows from the update rule $g_i^t$, $(ii)$ from triangle inequality and clipping properties from Lemma C.3. We continue the derivation of the bound for $\|g^t\|$ as follows

$$\|g^t\| \overset{(i)}{\le} (1 - \hat{\beta})\|g^{t-1}\| + \hat{\beta}\|\nabla f(x^{t-1})\| + \hat{\beta}\|\nabla f(x^t) - \nabla f(x^{t-1})\|$$

$$+ \frac{\hat{\beta}}{n} \sum_{i=1}^n \|(1 - \beta)v_i^{t-1} + \beta\nabla f_i(x^t) - \nabla f_i(x^t)\| + \hat{\beta}(B - \tau)$$

$$\overset{(ii)}{\le} (1 - \hat{\beta})\|g^{t-1}\| + \hat{\beta}\sqrt{2L(f(x^t) - f^*)} + L\gamma\hat{\beta}\|g^{t-1}\| + \frac{\hat{\beta}}{n}(1 - \beta) \sum_{i=1}^n \|\nabla f_i(x^t) - v_i^{t-1}\|$$

$$+ \hat{\beta}(B - \tau)$$

$$\overset{(iii)}{\le} (1 - \hat{\beta} + L\gamma\hat{\beta})\|g^{t-1}\| + \hat{\beta}\sqrt{2L\Phi^t} + \frac{\hat{\beta}}{n}(1 - \beta) \sum_{i=1}^n \|\nabla f_i(x^t) - \nabla f_i(x^{t-1})\|$$

$$+ \frac{\hat{\beta}}{n}(1 - \beta) \sum_{i=1}^n \|\nabla f_i(x^{t-1}) - v_i^{t-1}\| + \hat{\beta}(B - \tau)$$

$$\overset{(iv)}{\le} (1 - \hat{\beta} + L\gamma\hat{\beta}(2 - \beta))\|g^{t-1}\| + \hat{\beta}\sqrt{2L\Delta} + \hat{\beta}(1 - \beta)(\sqrt{4L\Delta} + \frac{3}{2}(B - \tau)) + \hat{\beta}(B - \tau)$$

$$\overset{(v)}{\le} (1 - \hat{\beta} + L\gamma\hat{\beta}(2 - \beta))(\sqrt{64L\Delta} + 3(B - \tau)) + \hat{\beta}\sqrt{2L\Delta} + \hat{\beta}(1 - \beta)(\sqrt{4L\Delta} + \frac{3}{2}(B - \tau))$$

$$+ \hat{\beta}(B - \tau),$$

where $(i)$ follows from triangle inequality and update of $v_i^t$, $(ii)$ from $L$-smoothness and update rule of $x^t$, $(iii)$ from the definition of $\Phi^t$ and triangle inequality, $(iv)$ from the assumptions 2 and 6, $(v)$ from the assumption 1. The above is satisfied if we have simultaneously

$$8(1 - \hat{\beta} + 2L\gamma\hat{\beta}) + \sqrt{2}\hat{\beta} + 2\hat{\beta} \le 8$$

$$3(1 - \hat{\beta} + 2L\gamma\hat{\beta}) + \frac{3}{2}\hat{\beta} + \hat{\beta} \le 3.$$

Both inequalities hold when $L\gamma \leq \frac{1}{12}$. $\qquad\square$

**Lemma E.3.** *Let each $f_i$ be $L$-smooth, $\Delta \geq \Phi^0$, and $B > \tau$. Assume that the following inequalities hold for the iterates generated by* `Clip21-SGD2M` *with $\nabla f_i(x^{t+1}, \xi_i^{t+1}) = \nabla f_i(x^{t+1})$ (full gradients) and $\sigma_\omega = 0$ (no DP-noise)*

    *1. $4L\gamma \leq \beta$ and $\gamma \leq \frac{1}{4L}$;*

    *2. $\|\nabla f_i(x^{t-1}) - v_i^{t-1}\| \leq \sqrt{4L\Delta} + \frac{3}{2}(B - \tau)$;*

    *3. $\|g^{t-1}\| \leq \sqrt{64L\Delta} + 3(B - \tau)$.*

*Then we have*

$$\|\nabla f_i(x^t) - v_i^t\| \leq \sqrt{4L\Delta} + \frac{3}{2}(B - \tau) \quad \forall i \in [n]. \tag{24}$$

*Proof.* We have

$$
\begin{aligned}
\|\nabla f_i(x^t) - v_i^t\| &\overset{(i)}{=} \|\nabla f_i(x^t) - (1-\beta)v_i^{t-1} - \beta\nabla f_i(x^t)\| \\
&= (1-\beta)\|\nabla f_i(x^t) - v_i^{t-1}\| \\
&\overset{(ii)}{\leq} (1-\beta)L\gamma\|g^{t-1}\| + (1-\beta)\|\nabla f_i(x^{t-1}) - v_i^{t-1}\| \\
&\overset{(iii)}{\leq} L\gamma\left(\sqrt{64L\Delta} + 3(B - \tau)\right) + (1-\beta)\left(\sqrt{4L\Delta} + \frac{3}{2}(B - \tau)\right) \\
&= (8L\gamma + 2(1-\beta))\sqrt{L\Delta} + \left(3L\gamma + \frac{3(1-\beta)}{2}\right)(B - \tau),
\end{aligned}
$$

where $(i)$ follows from the update rule of $v_i^t$, $(ii)$ from triangle inequality, smoothness, and update of $x^t$, $(iii)$ from conditions 2-3 in the statement of the lemma. We need to satisfy

$$
\begin{aligned}
8L\gamma + 2(1-\beta) &\leq 2 \Leftrightarrow 4L\gamma \leq \beta. \\
3L\gamma + \frac{3}{2}(1-\beta) &\leq \frac{3}{2} \Leftrightarrow 2L\gamma \leq \beta.
\end{aligned}
$$

Since $4L\gamma \leq \beta$, both inequalities are satisfied. $\qquad\square$

**Lemma E.4.** *Let each $f_i$ be $L$-smooth, $\Delta \geq \Phi^0$, $B > \tau$, and $i \in \mathcal{I}_t := \{i \in [n] \mid \|v_i^t - g_i^{t-1}\| > \tau\}$. Assume that the following inequalities hold for the iterates generated by* `Clip21-SGD2M` *with $\nabla f_i(x^{t+1}, \xi_i^{t+1}) = \nabla f_i(x^{t+1})$ (full gradients) and $\sigma_\omega = 0$ (no DP-noise)*

    *1. $4L\gamma \leq \beta$;*

    *2. $L\gamma \leq \frac{1}{12}$;*

    *3. $\frac{8}{3}\beta\sqrt{L\Delta} \leq \frac{\hat{\beta}\tau}{4}$;*

    *4. $\frac{7}{4}\beta(B - \tau) \leq \frac{\hat{\beta}\tau}{4}$;*

    *5. $\|g^t\| \leq \sqrt{64L\Delta} + 3(B - \tau)$;*

    *6. $\|\nabla f_i(x^t) - v_i^t\| \leq \sqrt{4L\Delta} + \frac{3}{2}(B - \tau)$.*

*Then*

$$\|v_i^{t+1} - g_i^t\| \leq \|v_i^t - g_i^{t-1}\| - \frac{\hat{\beta}\tau}{2}. \tag{25}$$

*Proof.* Since $i \in \mathcal{I}_t$, then $\|v_i^t - g_i^{t-1}\| > \tau$, thus from Lemma E.1 we have

$$\|v_i^{t+1} - g_i^t\| \leq (1-\hat{\beta})\|v_i^t - g_i^{t-1}\| + \hat{\beta}(\|v_i^t - g_i^{t-1}\| - \tau) + \beta L\gamma\|g^t\| + \beta\|\nabla f_i(x^t) - v_i^t\|$$

$$\overset{(i)}{\leq} \|v_i^t - g_i^{t-1}\| - \hat{\beta}\tau + \beta L\gamma\left(\sqrt{64L\Delta} + 3(B-\tau)\right) + \beta\left(\sqrt{4L\Delta} + \frac{3}{2}(B-\tau)\right)$$

$$= \|v_i^t - g_i^{t-1}\| - \hat{\beta}\tau + (8\beta L\gamma + 2\beta)\sqrt{L\Delta} + (3\beta L\gamma + 3\beta/2)(B-\tau),$$

where $(i)$ follows from assumptions 5-6 of the statement of the lemma. Since $L\gamma \leq \frac{1}{12}$, we have

$$\|v_i^{t+1} - g_i^t\| \leq \|v_i^t - g_i^{t-1}\| - \hat{\beta}\tau + \frac{8}{3}\beta\sqrt{L\Delta} + \frac{7}{4}\beta(B-\tau).$$

Due to assumptions 2-3 of the lemma, we have

$$\|v_i^{t+1} - g_i^t\| \leq \|v_i^t - g_i^{t-1}\| - \frac{\hat{\beta}\tau}{2},$$

which concludes the proof. $\qquad\square$

**Lemma E.5.** *Let each $f_i$ be $L$-smooth. Then, for the iterates generated by* `Clip21-SGD2M` *with $\nabla f_i(x^{t+1}, \xi_i^{t+1}) = \nabla f_i(x^{t+1})$ (full gradients) and $\sigma_\omega = 0$ (no DP-noise) the quantity $\widetilde{P}^t := \frac{1}{n}\sum_{i=1}^n \|v_i^t - \nabla f_i(x^t)\|^2$ decreases as*

$$\widetilde{P}^{t+1} \leq (1-\beta)\widetilde{P}^t + \frac{3L^2}{\beta}R^t. \tag{26}$$

*Proof.* We have

$$\|v_i^{t+1} - \nabla f_i(x^{t+1})\|^2 \overset{(i)}{=} \|(1-\beta)v_i^t + \beta\nabla f_i(x^{t+1}) - \nabla f_i(x^{t+1})\|^2$$

$$= (1-\beta)^2\|\nabla f_i(x^{t+1}) - v_i^t\|^2$$

$$\overset{(ii)}{\leq} (1-\beta)^2(1+\beta/2)\|v_i^t - \nabla f_i(x^t)\|^2$$

$$+ (1-\beta)^2(1+2/\beta)\|\nabla f_i(x^t) - \nabla f_i(x^{t+1})\|^2$$

$$\overset{(iii)}{\leq} (1-\beta)\|v_i^t - \nabla f_i(x^t)\|^2 + \frac{3L^2}{\beta}\|x^t - x^{t+1}\|^2,$$

where $(i)$ follows from the update rule of $v_i^t$, $(ii)$ – from the inequality $\|a + b\|^2 \leq (1 + \beta/2)\|a\|^2 + (1+2/\beta)\|b\|^2$ that holds for any $a, b \in \mathbb{R}^d$ and $\beta > 0$, and $(iii)$ – from $(1-\beta)(1+\beta/2) \leq 1$, which holds for any $\beta \in [0, 1]$, and smoothness. Averaging the inequalities above across $i \in [n]$, we get the statement of the lemma. $\qquad\square$

Similarly, we can get the recursion for $P^t := \|v^t - \nabla f(x^t)\|^2$.

**Lemma E.6.** *Let each $f_i$ be $L$-smooth. Then, for the iterates generated by* `Clip21-SGD2M` *with $\nabla f_i(x^{t+1}, \xi_i^{t+1}) = \nabla f_i(x^{t+1})$ (full gradients) and $\sigma_\omega = 0$ (no DP-noise) the quantity $P^t := \|v^t - \nabla f(x^t)\|^2$ decreases as*

$$P^{t+1} \leq (1-\beta)P^t + \frac{3L^2}{\beta}R^t. \tag{27}$$

Next, we establish the recursion for $\widetilde{V}^t := \frac{1}{n}\sum_{i=1}^n \|g_i^t - v_i^t\|^2$.

**Lemma E.7.** *Let each $f_i$ be $L$-smooth. Consider* `Clip21-SGD2M` *with $\nabla f_i(x^{t+1}, \xi_i^{t+1}) = \nabla f_i(x^{t+1})$ (full gradients) and $\sigma_\omega = 0$ (no DP-noise). Let $\|v_i^t - g_i^{t-1}\| \leq B$, for all $i \in [n]$ and some $B \geq \tau$, and $\hat{\beta} \leq \frac{1}{2\eta}$. Then*

$$\|g_i^t - v_i^t\|^2 \leq (1 - \hat{\beta}\eta)\|g_i^{t-1} - v_i^{t-1}\|^2 + \frac{4\beta^2}{\hat{\beta}\eta}\|v_i^{t-1} - \nabla f_i(x^{t-1})\|^2 + \frac{4L^2\beta^2}{\hat{\beta}}R^{t-1}.$$

*and, in particular,*

$$\widetilde{V}^t \le (1-\eta)\widetilde{V}^{t-1} + \frac{4\beta^2}{\hat{\beta}\eta}\widetilde{P}^{t-1} + \frac{4\beta^2 L^2}{\hat{\beta}\eta}R^{t-1},$$

*where* $\eta := \frac{\tau}{B}$, $R^t := \|x^{t+1} - x^t\|^2$, *and* $\widetilde{V}^t := \frac{1}{n}\sum_{i=1}^{n}\|g_i^t - v_i^t\|^2$.

*Proof.* Since $\|v_i^t - g_i^{t-1}\| \le B$, for $\eta_i^t := \frac{\tau}{\|v_i^t - g_i^{t-1}\|}$ we have $\eta_i^t \ge \eta$. This implies

$$\|g_i^t - v_i^t\|^2 \overset{(i)}{=} \|g_i^{t-1} + \hat{\beta}\operatorname{clip}_\tau(v_i^t - g_i^{t-1}) - v_i^t\|^2$$
$$= \|\hat{\beta}(g_i^{t-1} - v_i^t + \operatorname{clip}_\tau(v_i^t - g_i^{t-1})) + (1-\hat{\beta})(g_i^{t-1} - v_i^t)\|^2$$
$$\overset{(ii)}{\le} (1-\eta)^2\hat{\beta}\|g_i^{t-1} - v_i^t\|^2 + (1-\hat{\beta})\|g_i^{t-1} - v_i^t\|^2,$$

where $(i)$ follows from the update rule of $g_i^t$ and $(ii)$ from the convexity of $\|\cdot\|^2$ and the fact that $\|v_i^t - g_i^{t-1}\| \le B$. We continue the derivations as follows

$$\|g_i^t - v_i^t\|^2 = (1 - \hat{\beta} + \hat{\beta}(1 - 2\eta + \eta^2))\|g_i^{t-1} - v_i^t\|^2$$
$$= (1 - \hat{\beta}\eta(2-\eta))\|g_i^{t-1} - v_i^t\|^2.$$

Let $\rho = 2\hat{\beta}\eta$ (note that $\eta \le 1$). Then we have

$$\|g_i^t - v_i^t\|^2 \le (1-\rho)\|g_i^{t-1} - v_i^t\|^2$$
$$\overset{(i)}{=} (1-\rho)\|g_i^{t-1} - (1-\beta)v_i^{t-1} - \beta\nabla f_i(x^t)\|^2$$
$$\overset{(ii)}{\le} (1-\rho)(1+\rho/2)\|g_i^{t-1} - v_i^{t-1}\|^2 + (1-\rho)(1+2/\rho)\beta^2\|v_i^{t-1} - \nabla f_i(x^t)\|^2$$
$$\overset{(iii)}{\le} (1-\rho/2)\|g_i^{t-1} - v_i^{t-1}\|^2 + \frac{4\beta^2}{\rho}\|v_i^{t-1} - \nabla f_i(x^{t-1})\|^2 + \frac{4L^2\beta^2}{\rho}R^{t-1},$$

where $(i)$ follows from the update rule of $g_i^t$, $(ii)$ from the inequality $\|a+b\|^2 \le (1+r/2)\|a\|^2 + (1+2/r)\|b\|^2$, which holds for any positive $r$ (i.e., for $r = \rho$ for some $\rho > 0$) and $a, b \in \mathbb{R}^d$, $(iii)$ from the fact that $\rho \le 1$ by assumption, the inequality $\|a+b\|^2 \le 2\|a\|^2 + 2\|b\|^2$, which holds for any $a, b \in \mathbb{R}^d$, and smoothness. Finally, since $2\hat{\beta}\eta \le 1$, we ensure that $\rho \le 1$, and derive the final bound

$$\|g_i^t - v_i^t\|^2 \le (1-\hat{\beta}\eta)\|g_i^{t-1} - v_i^{t-1}\|^2 + \frac{4\beta^2}{\hat{\beta}\eta}\|v_i^{t-1} - \nabla f_i(x^{t-1})\|^2 + \frac{4L^2\beta^2}{\hat{\beta}}R^{t-1}.$$

$\square$

**Theorem E.8** (Full statement of Theorem 3.1). *Let Assumption 1.1 hold. Let* $B := \max\{3\tau, \max_i\|\nabla f_i(x^0)\|\}$ *and* $\Phi^0$ *defined in* (9) *satisfies* $\Delta \ge \Phi^0$ *for some* $\Delta > 0$. *Assume the following inequalities hold*

1. **stepsize restrictions:** $\gamma \le \frac{1}{12L}, 4L\gamma = \beta$, *and*
$$\frac{5}{8} - \frac{32\beta^2 L^2}{\hat{\beta}^2\eta^2}\gamma^2 - \frac{96L^2}{\hat{\beta}^2\eta^2}\gamma^2 \ge 0;$$

2. **momentum restrictions:** $\frac{8}{3}\beta\sqrt{L\Delta} \le \frac{\hat{\beta}\tau}{4}, \frac{7}{4}\beta(B-\tau) \le \frac{\hat{\beta}\tau}{4}, \hat{\beta} \le \frac{1}{2\eta}$[5].

*Then, the Lyapunov function from* (9) *for* `Clip21-SGD2M` *with* $\nabla f_i(x^{t+1}, \xi_i^{t+1}) = \nabla f_i(x^{t+1})$ *(full gradients) and* $\sigma_\omega = 0$ *(no DP-noise) decreases as*

$$\Phi^{t+1} \le \Phi^t - \frac{\gamma}{2}\|\nabla f(x^t)\|^2,$$

---

[5]Note that $\eta = \frac{\tau}{B} \le \frac{1}{3}$ by the choice of $B$, therefore $\hat{\beta} \le \frac{1}{2\eta}$ does not impose any additional assumption on $\hat{\beta}$ and it can be chosen from $[0, 1]$.

*and we have*

$$\frac{1}{T}\sum_{t=0}^{T-1}\|\nabla f(x^t)\|^2 \le \frac{2\Delta}{\gamma T} = \mathcal{O}\left(\frac{1}{T}\right). \tag{28}$$

*Moreover, after at most $\frac{2B}{\hat{\beta}\tau}$ iterations, the clipping operator will be turned off for all workers.*

*Proof.* For convenience, we define

$$\nabla f_i(x^{-1}) = v_i^{-1} = g_i^{-1} = 0, \quad \Phi^{-1} = +\infty.$$

Then, we will derive the result by induction, i.e., using the induction w.r.t. $t$, we will show that

1. the Lyapunov function decreases as $\Phi^t \le \Phi^{t-1} - \frac{\gamma}{2}\|\nabla f(x^{t-1})\|^2$;

2. $\|g^t\| \le \sqrt{64L\Delta} + 3(B - \tau)$;

3. $\|v_i^t - \nabla f_i(x^t)\| \le \sqrt{4L\Delta} + \frac{3}{2}(B - \tau)$;

4. $\|v_i^t - g_i^{t-1}\| \le \max\left\{0, B - \frac{t\hat{\beta}\tau}{2}\right\}$.

First, we prove that the base of induction holds.

**Base of induction.**

1. $\|v_i^0 - g_i^{-1}\| = \|v_i^0\| = \beta\|\nabla f_i(x^0)\| \le \frac{1}{2}B \le B$ holds;

2. $g^0 = \frac{1}{n}\sum_{i=1}^n (g_i^{-1} + \hat{\beta}\,\text{clip}_\tau(v_i^0 - g_i^{-1}) = \frac{\hat{\beta}}{n}\sum_{i=1}^n \text{clip}_\tau(\beta\nabla f_i(x^0))$. Therefore, we have

$$\|g^0\| \le \left\|\frac{\hat{\beta}}{n}\sum_{i=1}^n \beta\nabla f_i(x^0) + (\text{clip}_\tau(\beta\nabla f_i(x^0)) - \beta\nabla f_i(x^0))\right\|$$

$$\le \hat{\beta}\beta\|\nabla f(x^0)\| + \frac{\hat{\beta}}{n}\sum_{i=1}^n \max\left\{0, \beta\|\nabla f_i(x^0)\| - \tau\right\}$$

$$\le \hat{\beta}\beta\sqrt{2L(f(x^0) - f^*)} + \hat{\beta}(B - \tau)$$

$$\le \sqrt{64L\Delta} + 3(B - \tau).$$

3. We have

$$\|v_i^0 - \nabla f_i(x^0)\| = \|\beta\nabla f_i(x^0) - \nabla f_i(x^0)\|$$

$$\le (1 - \beta)B$$

$$\le \sqrt{4L\Delta} + \frac{3}{2}(B - \tau)$$

4. $\Phi^0 \le \Phi^{-1} - \frac{\gamma}{2}\|\nabla f(x^{-1})\|^2 = \Phi^{-1}$ holds.

**Transition of induction.** Assume that for $K$ we have that for all $t \in \{0, 1, \dots, K\}$

1. $\Phi^t \le \Phi^{t-1} - \frac{\gamma}{2}\|\nabla f(x^{t-1})\|^2$ (implying $\Phi^t \le \Delta$);

2. $\|g^t\| \le \sqrt{64L\Delta} + 3(B - \tau)$;

3. $\|v_i^t - \nabla f_i(x^t)\| \le \sqrt{4L\Delta} + \frac{3}{2}(B - \tau)$;

4. $\|v_i^t - g_i^{t-1}\| \le \max\left\{\hat{\beta}\tau, B - \frac{t\hat{\beta}\tau}{2}\right\}$.

We proceed via analyzing two possible situations for $\mathcal{I}_{K+1} := \{i \in [n] \mid \|v_i^{K+1} - g_i^K\| > \tau\}$: either $|\mathcal{I}_{K+1}| > 0$ (there are workers with turned on gradient clipping) or $|\mathcal{I}_{K+1}| = 0$ (for all workers the clipping is turned off).

CASE $|\mathcal{I}_{K+1}| > 0$. Since all requirements of Lemma E.4 are satisfied at iteration $K$ we get for all $i \in \mathcal{I}_{K+1}$

$$\|v_i^{K+1} - g_i^K\| \leq \|v_i^K - g_i^{K-1}\| - \frac{\hat{\beta}\tau}{2} \overset{(i)}{\leq} \max\left\{\tau, B - \frac{K\hat{\beta}\tau}{2}\right\} - \frac{\hat{\beta}\tau}{2} \leq \max\left\{\tau, B - \frac{(K+1)\hat{\beta}\tau}{2}\right\},$$

where (i) follows from the condition 4 of the induction assumption. Similarly due to the assumption of induction, from Lemma E.2 we get that

$$\|g^{K+1}\| \leq \sqrt{64L\Delta} + 3(B - \tau),$$

and from Lemma E.3

$$\|\nabla f_i(x^{K+1}) - v_i^{K+1}\| \leq \sqrt{4L\Delta} + \frac{3}{2}(B - \tau).$$

This means that conditions 2-4 in the assumption of the induction are also verified for step $K + 1$. The remaining part is the descent of the Lyapunov function. For estimating $\widetilde{V}^{K+1} := \frac{1}{n}\sum_{i=1}^{n}\|g_i^{K+1} - v_i^{K+1}\|^2$ we have Lemma E.7 since $\|v_i^{K+1} - g_i^K\| \leq B - \frac{\tau}{2}$

$$\widetilde{V}^{K+1} \leq (1 - \hat{\beta}\eta)\widetilde{V}^K + \frac{4\beta^2}{\hat{\beta}\eta}\widetilde{P}^K + \frac{4\beta^2 L^2}{\hat{\beta}\eta}R^K.$$

Combining this result with the claims of Lemmas C.2, E.5 and E.6 we get

$$\Phi^{K+1} = \delta^{K+1} + \frac{2\gamma}{\hat{\beta}\eta}\widetilde{V}^{K+1} + \frac{8\gamma\beta}{\hat{\beta}^2\eta^2}\widetilde{P}^{K+1} + \frac{2\gamma}{\beta}P^{K+1}$$

$$\leq \delta^K - \frac{\gamma}{2}\|\nabla f(x^K)\|^2 - \frac{1}{4\gamma}R^K + 2\gamma\widetilde{V}^K + 2\gamma P^K$$

$$+ \frac{2\gamma}{\hat{\beta}\eta}\left((1 - \hat{\beta}\eta)\widetilde{V}^K + \frac{4\beta^2}{\hat{\beta}\eta}\widetilde{P}^K + \frac{4\beta^2 L^2}{\hat{\beta}\eta}R^K\right)$$

$$+ \frac{8\gamma\beta}{\hat{\beta}^2\eta^2}\left((1 - \beta)\widetilde{P}^K + \frac{3L^2}{\beta}R^K\right)$$

$$+ \frac{2\gamma}{\beta}\left((1 - \beta)P^K + \frac{3L^2}{\beta}R^K\right)$$

$$= \delta^K - \frac{\gamma}{2}\|\nabla f(x^K)\|^2 + \frac{2\gamma}{\hat{\beta}\eta}\widetilde{V}^K\left(1 - \hat{\beta}\eta + \hat{\beta}\eta\right) + \frac{8\gamma\beta}{\hat{\beta}^2\eta^2}\widetilde{P}^K(1 - \beta + \beta)$$

$$+ \frac{2\gamma}{\beta}P^K(1 - \beta + \beta) - \frac{1}{4\gamma}\left(1 - \frac{32\beta^2 L^2}{\hat{\beta}^2\eta^2}\gamma^2 - \frac{96L^2}{\hat{\beta}^2\eta^2}\gamma^2 - \frac{24L^2}{\beta^2}\gamma^2\right)R^K$$

$$= \Phi^K - \frac{\gamma}{2}\|\nabla f(x^K)\|^2 - \frac{1}{4\gamma}\left(1 - \frac{32\beta^2 L^2}{\hat{\beta}^2\eta^2}\gamma^2 - \frac{96L^2}{\hat{\beta}^2\eta^2}\gamma^2 - \frac{24L^2}{\beta^2}\gamma^2\right)R^K.$$

Since we choose $\beta^2 = 64L^2\gamma^2$, then $-\frac{1}{\beta^2} = -\frac{1}{64L^2\gamma^2}$ and $-\frac{24L^2}{\beta^2}\gamma^2 = -\frac{24L^2}{64^2 L^2\gamma^2}\gamma^2 \geq -\frac{3}{8}$ Therefore,

$$1 - \frac{32\beta^2 L^2}{\eta^2}\gamma^2 - \frac{96L^2}{\hat{\beta}^2\eta^2}\gamma^2 - \frac{24L^2}{\beta^2}\gamma^2 \geq \frac{5}{8} - \frac{32\beta^2 L^2}{\hat{\beta}^2\eta^2}\gamma^2 - \frac{96L^2}{\hat{\beta}^2\eta^2}\gamma^2 \geq 0,$$

by the choice of $\gamma$. Thus, we get

$$\Phi^{K+1} \leq \Phi^K - \frac{\gamma}{2}\|\nabla f(x^K)\|^2.$$

In particular, this implies $\Phi^{K+1} \leq \Phi^K \leq \Delta$.

CASE $|\mathcal{I}_{K+1}| = 0$. In this case, $\eta_i^{K+1} = 1$ for all $i \in [n]$, i.e., $\mathrm{clip}_\tau(v_i^{K+1} - g_i^K) = v_i^{K+1} - g_i^K$ that leads to $g_i^{K+1} = v_i^{K+1}$. Thus, $\widetilde{V}^{K+1} = 0$. Moreover, $|\mathcal{I}_{K+1}| = 0$ implies that condition 4 from the induction assumption holds for $t = K + 1$ and using this and induction assumption we get $\|g^{K+1}\| \leq \sqrt{64L\Delta} + 3(B - \tau)$ from Lemma E.2 and $\|\nabla f_i(x^{K+1}) - v_i^{K+1}\| \leq \sqrt{4L\Delta} + \frac{3}{2}(B - \tau)$ from Lemma E.3. Next, taking into account that $\widetilde{V}^{K+1} = 0$, we can perform similar steps as before for $\Phi^{K+1}$ and get less restrictive inequality

$$\Phi^{K+1} \leq \Phi^K - \frac{\gamma}{2}\|\nabla f(x^K)\|^2 - \frac{1}{4\gamma}\left(1 - \frac{96L^2}{\hat{\beta}^2\eta^2}\gamma^2 - \frac{24L^2}{\beta^2}\gamma^2\right)R^K.$$

Again, $1 - \frac{96L^2}{\hat{\beta}^2\eta^2}\gamma^2 - \frac{24L^2}{\beta^2}\gamma^2 \geq \frac{5}{8} - \frac{96L^2}{\hat{\beta}^2\eta^2}\gamma^2 \geq 0$ which is satisfied by the choice of $\gamma$.

We conclude that in both cases the Lyapunov function decreases as $\Phi^{K+1} \leq \Phi^K - \frac{\gamma}{2}\|\nabla f(x^K)\|^2$, and consequently, $\Phi^{K+1} \leq \Delta$. This finalizes the induction step. Therefore, we can guarantee that for all iterations $t \in \{0, 1, \ldots, T-1\}$ we have

$$\Phi^{t+1} \leq \Phi^t - \frac{\gamma}{2}\|\nabla f(x^t)\|^2 \Rightarrow \frac{1}{T}\sum_{t=0}^{T-1}\|\nabla f(x^t)\|^2 \leq \frac{2\Delta}{\gamma T}.$$

Moreover, the proof shows that the clipping operator will be eventually turned off after at most $\frac{2B}{\hat{\beta}\tau}$ iterations since $\|v_i^t - g_i^{t-1}\| \leq \max\left\{\tau, B - \frac{t\hat{\beta}\tau}{2}\right\}$. □

# F    PROOF OF THEOREM 3.3 (CONVERGENCE OF Clip21-SGD2M IN THE STOCHASTIC SETTING WITH DP NOISE)

We define constants $a$, $b$, and $c$, which will be used later in the proofs, as follows:

$$a := \left( \sqrt{2} + 2\sqrt{3 \log \frac{6(T+1)}{\alpha}} \right) \sqrt{d} \sigma_\omega \sqrt{\frac{T}{n}},$$

$$b^2 := 2\sigma^2 \log \left( \frac{12(T+1)n}{\alpha} \right), \qquad (29)$$

$$c^2 := \left( \sqrt{2} + 2\sqrt{3 \log \frac{6(T+1)}{\alpha}} \right)^2 \sigma^2,$$

where $T$ is the number of iterations, $n$ is the number of workers, $d$ is the dimension of the problem, $\sigma$ is from Assumption 1.2, $\alpha \in (0,1)$ is a constant, and $\sigma_\omega$ is the variance of DP noise.

**Lemma F.1.** *Let each $f_i$ be L-smooth. Then, for the iterates of Clip21-SGD2M we have the following inequality with probability* 1

$$\|v_i^{t+1} - g_i^t\| \le (1 - \hat{\beta})\|v_i^t - g_i^{t-1}\| + \hat{\beta} \max \left\{0, \|v_i^t - g_i^{t-1}\| - \tau\right\} + \beta L \gamma \|g^t\|$$
$$+ \beta \|\nabla f_i(x^t) - v_i^t\| + \beta \|\theta_i^{t+1}\|, \qquad (30)$$

*where $\theta_i^t := \nabla f_i(x^t, \xi_i^t) - \nabla f_i(x^t)$.*

*Proof.* We have

$$\|v_i^{t+1} - g_i^t\| \overset{(i)}{=} \|(1 - \beta)v_i^t + \beta \nabla f_i(x^{t+1}, \xi_i^{t+1}) - g_i^t\|$$

$$\overset{(ii)}{\le} \|v_i^t - g_i^t\| + \beta \|\nabla f_i(x^{t+1}, \xi_i^{t+1}) - v_i^t\|$$

$$\overset{(iii)}{=} \|v_i^t - \hat{\beta} \operatorname{clip}_\tau(v_i^t - g_i^{t-1}) - g_i^{t-1}\| + \beta \|\nabla f_i(x^{t+1}, \xi_i^{t+1}) - v_i^t\|$$

$$\overset{(iv)}{\le} (1 - \hat{\beta})\|v_i^t - g_i^{t-1}\| + \hat{\beta} \max \left\{0, \|v_i^t - g_i^{t-1}\| - \tau\right\} + \beta \|\nabla f_i(x^{t+1}, \xi_i^{t+1}) - \nabla f_i(x^{t+1})\|$$
$$+ \beta \|\nabla f_i(x^{t+1}) - \nabla f_i(x^t)\| + \beta \|\nabla f_i(x^t) - v_i^t\|$$

$$\overset{(v)}{\le} (1 - \hat{\beta})\|v_i^t - g_i^{t-1}\| + \hat{\beta} \max \left\{0, \|v_i^t - g_i^{t-1}\| - \tau\right\} + \beta L \|x^{t+1} - x^t\|$$
$$+ \beta \|\nabla f_i(x^t) - v_i^t\| + \beta \|\theta_i^{t+1}\|$$

$$\overset{(vi)}{=} (1 - \hat{\beta})\|v_i^t - g_i^{t-1}\| + \hat{\beta} \max \left\{0, \|v_i^t - g_i^{t-1}\| - \tau\right\} + \beta L \gamma \|g^t\|$$
$$+ \beta \|\nabla f_i(x^t) - v_i^t\| + \beta \|\theta_i^{t+1}\|,$$

where $(i)$ follows from the update rule of $v_i^t$, $(ii)$ from triangle inequality, $(iii)$ from the update rule of $g_i^t$, $(iv)$ from the properties of the clipping operator from Lemma C.3 and triangle inequality, $(v)$ from smoothness, $(vi)$ from the update rule of $x^t$. □

**Lemma F.2.** *Let each $f_i$ be L-smooth, $\Delta \ge \Phi^0$. Assume that the following inequalities hold for the iterates generated by Clip21-SGD2M*

1. $g^0 = \frac{1}{n} \sum_{i=1}^n g_i^0$;

2. $\|g^{t-1}\| \le \sqrt{64L\Delta} + 3(B - \tau) + 3b + 3\hat{\beta}a$;

3. $\|\overline{g}^{t-1}\| \le \sqrt{64L\Delta} + 3(B - \tau) + 3b$;

4. $\|\nabla f_i(x^{t-1}) - v_i^{t-1}\| \le \sqrt{4L\Delta} + \frac{3}{2}(B - \tau) + \frac{3}{2}b + \hat{\beta}a$ *for all $i \in [n]$*;

5. $\|v_i^t - g_i^{t-1}\| \le B$ *for all $i \in [n]$*;

6. $\gamma \le \frac{1}{12L}$;

7. $\|\theta_i^t\| \leq b$ *for all* $i \in [n]$;

8. $\left\| \frac{1}{n} \sum_{l=1}^{t} \sum_{i=1}^{n} \omega_i^l \right\| \leq a$;

9. $\beta, \hat{\beta} \in [0, 1]$;

10. $\Phi^{t-1} \leq 2\Delta$.

*Then we have*

$$\|g^t\| \leq \sqrt{64L\Delta} + 3(B - \tau) + 3b + 3\hat{\beta}a. \tag{31}$$

*Proof.* We start as follows

$$
\|g^t\| \overset{(i)}{=} \left\| g^{t-1} + \frac{\hat{\beta}}{n} \sum_{i=1}^{n} \text{clip}_\tau(v_i^t - g_i^{t-1}) + \frac{\hat{\beta}}{n} \sum_{i=1}^{n} \omega_i^t \right\|
$$

$$
= \left\| g^{t-1} + \frac{\hat{\beta}}{n} \sum_{i=1}^{n} \left[ \nabla f_i(x^t) + (v_i^t - \nabla f_i(x^t)) + \text{clip}_\tau(v_i^t - g_i^{t-1}) - (v_i^t - g_i^{t-1}) \right] \right.
$$

$$
\left. - \overline{g}^{t-1} + (1 - \hat{\beta})\overline{g}^{t-1} + \frac{\hat{\beta}}{n} \sum_{i=1}^{n} \omega_i^t \right\|
$$

$$
\overset{(ii)}{\leq} \left\| g^{t-1} - \overline{g}^{t-1} + \frac{\hat{\beta}}{n} \sum_{i=1}^{n} \omega_i^t \right\| + \hat{\beta}\|\nabla f(x^t)\| + \frac{\hat{\beta}}{n} \sum_{i=1}^{n} \|\text{clip}_\tau(v_i^t - g_i^{t-1}) - v_i^t + g_i^{t-1}\|
$$

$$
+ (1 - \hat{\beta})\|\overline{g}^{t-1}\| + \frac{\hat{\beta}}{n} \sum_{i=1}^{n} \|v_i^t - \nabla f_i(x^t)\|
$$

$$
\overset{(iii)}{\leq} \left\| \overline{g}^{t-1} + \hat{\beta}\Omega^{t-1} - \overline{g}^{t-1} + \frac{\hat{\beta}}{n} \sum_{i=1}^{n} \omega_i^t \right\| + \hat{\beta}\|\nabla f(x^{t-1})\| + \hat{\beta}\|\nabla f(x^t) - \nabla f(x^{t-1})\|
$$

$$
+ \frac{\hat{\beta}}{n} \sum_{i=1}^{n} \|\text{clip}_\tau(v_i^t - g_i^{t-1}) - v_i^t + g_i^{t-1}\| + (1 - \hat{\beta})\|\overline{g}^{t-1}\|
$$

$$
+ \frac{\hat{\beta}}{n} \sum_{i=1}^{n} \|(1 - \beta)v_i^{t-1} + \beta\nabla f_i(x^t, \xi_i^t) - \nabla f_i(x^t)\|,
$$

where $(i)$ follows from the update rule of $g^t$, $(ii)$ – from the triangle inequality, $(iii)$ – from the update rule of $v_i^t$, equality (15), and triangle inequality. Using the definition of $\Omega^t$, we continue as

follows

$$\|g^t\| \overset{(iv)}{\leq} \hat{\beta}\|\Omega^t\| + \hat{\beta}\|\nabla f(x^{t-1})\| + \hat{\beta}L\gamma\|g^{t-1}\| + \frac{\hat{\beta}}{n}\sum_{i=1}^n \max\{0, \|v_i^t - g_i^{t-1}\| - \tau\} + (1-\hat{\beta})\|\overline{g}^{t-1}\|$$

$$+ \frac{\hat{\beta}}{n}\sum_{i=1}^n \|(1-\beta)v_i^{t-1} + \beta\nabla f_i(x^t, \xi_i^t) - \nabla f_i(x^t)\|$$

$$\overset{(v)}{\leq} \hat{\beta}\sqrt{2L(f(x^{t-1}) - f^*)} + \hat{\beta}L\gamma\|g^{t-1}\| + (1-\hat{\beta})\|\overline{g}^{t-1}\| + \hat{\beta}(B - \tau) + \hat{\beta}\|\Omega^t\|$$

$$+ \frac{\hat{\beta}}{n}\sum_{i=1}^n \left((1-\beta)\|v_i^{t-1} - \nabla f_i(x^t)\| + \beta\|\nabla f_i(x^t, \xi_i^t) - \nabla f_i(x^t)\|\right)$$

$$\overset{(vi)}{\leq} \hat{\beta}\sqrt{2L(f(x^{t-1}) - f^*)} + \hat{\beta}L\gamma\|g^{t-1}\| + (1-\hat{\beta})\|\overline{g}^{t-1}\| + \hat{\beta}(B - \tau) + \hat{\beta}\|\Omega^t\|$$

$$+ \frac{\hat{\beta}\beta}{n}\sum_{i=1}^n \|\theta_i^t\| + \frac{\hat{\beta}}{n}(1-\beta)\sum_{i=1}^n \left(\|v_i^{t-1} - \nabla f_i(x^{t-1})\| + \|\nabla f_i(x^t) - \nabla f_i(x^{t-1})\|\right)$$

$$\overset{(vii)}{\leq} \hat{\beta}\sqrt{2L(f(x^{t-1}) - f^*)} + \hat{\beta}L\gamma(2-\beta)\|g^{t-1}\| + (1-\hat{\beta})\|\overline{g}^{t-1}\| + \hat{\beta}(B - \tau) + \hat{\beta}\|\Omega^t\|$$

$$+ \frac{\hat{\beta}\beta}{n}\sum_{i=1}^n \|\theta_i^t\| + \frac{\hat{\beta}}{n}(1-\beta)\sum_{i=1}^n \|v_i^{t-1} - \nabla f_i(x^{t-1})\|.$$

$(iv)$ – from the properties of the clipping operator from Lemma C.3, $L$-smoothness and update rule of $x^t$, $(v)$ – from $L$-smoothness and triangle inequality, $(vi)$ – from triangle inequality, $(vii)$ – from $L$-smoothness. Now we use the assumptions 2-5, 7-8, and 10 to bound the terms

$$\|g^t\| \leq \hat{\beta}\sqrt{4L\Delta} + 2L\gamma\hat{\beta}\left(\sqrt{64L\Delta} + 3(B - \tau) + 3b + 3\hat{\beta}a\right) + (1-\hat{\beta})\left(\sqrt{64L\Delta} + 3(B - \tau) + 3b\right)$$

$$+ \hat{\beta}(B - \tau) + \hat{\beta}a + \hat{\beta}\beta b + \hat{\beta}(1-\beta)\left(\sqrt{4L\Delta} + \frac{3}{2}(B - \tau) + \frac{3}{2}b + \hat{\beta}a\right).$$

Regrouping the terms we obtain

$$\|g^t\| \leq \sqrt{L\Delta}[2\hat{\beta} + 16L\gamma\hat{\beta} + 8(1-\hat{\beta}) + 2\hat{\beta}(1-\beta)] + b[6L\gamma\hat{\beta} + 3(1-\hat{\beta}) + \hat{\beta}\beta + \sfrac{3}{2}\hat{\beta}(1-\beta)]$$

$$+ (B - \tau)[6L\gamma\hat{\beta} + 3(1-\hat{\beta}) + \hat{\beta} + \sfrac{3}{2}\hat{\beta}(1-\beta)] + a[6L\gamma\hat{\beta}^2 + \hat{\beta} + \hat{\beta}^2(1-\beta)].$$

For the first coefficient, we have

$$2\hat{\beta} + 16L\gamma\hat{\beta} + 8(1-\hat{\beta}) + 2\hat{\beta}(1-\beta) \leq 8 \Leftarrow 4\hat{\beta} + 16L\gamma\hat{\beta} \leq 8\hat{\beta} \Leftarrow 4L\gamma \leq 1,$$

where the last inequality is satisfied by the choice of the stepsize $L\gamma \leq \frac{1}{12}$. For the second coefficient, we have

$$6L\gamma\hat{\beta} + 3(1-\hat{\beta}) + \hat{\beta}\beta + \frac{3}{2}\hat{\beta}(1-\beta) \leq 3 \Leftarrow 6L\gamma\hat{\beta} + \hat{\beta}\beta + \frac{3}{2}\hat{\beta}(1-\beta) \leq 3\hat{\beta}$$

$$\Leftarrow 6L\gamma + 1 + \frac{3}{2}(1-\beta) \leq 3,$$

where the last inequality is satisfied by the choice of the stepsize $6L\gamma \leq \frac{1}{2}$ and momentum parameter $\beta \leq 1$. For the third coefficient, we have

$$6L\gamma\hat{\beta} + 3(1-\hat{\beta}) + \hat{\beta} + \frac{3}{2}\hat{\beta}(1-\beta) \leq 3 \Leftarrow 6L\gamma\hat{\beta} + \hat{\beta} + \frac{3}{2}\hat{\beta}(1-\beta) \leq 3\hat{\beta} \Leftarrow 6L\gamma + 1 + \frac{3}{2} \leq 3,$$

where the last inequality is satisfied by the choice of the stepsize $6L\gamma \leq \frac{1}{2}$. For the fourth coefficient, we have

$$6L\gamma\hat{\beta}^2 + \hat{\beta} + \hat{\beta}^2(1-\beta) \leq 3\hat{\beta} \Leftarrow 6L\gamma\hat{\beta}^2 + \hat{\beta}^2 \leq 2\hat{\beta} \Leftarrow 6L\gamma\hat{\beta} + \hat{\beta} \leq 2,$$

where the last inequality is satisfied by the choice of the stepsize $6L\gamma \leq \frac{1}{2}$ and momentum parameter $\hat{\beta} \leq 1$. Thus, the statement of the lemma holds. $\qquad\square$

**Lemma F.3.** *Let each $f_i$ be $L$-smooth, $\Delta \geq \Phi^0$, $B > \tau$. Assume that the following inequalities hold for the iterates generated by* `Clip21-SGD2M`

    *1.* $\gamma \leq \frac{1}{12L}$;

    *2.* $6L\gamma \leq \beta$;

    *3.* $\|\nabla f_i(x^{t-1}) - v_i^{t-1}\| \leq \sqrt{4L\Delta} + \frac{3}{2}(B - \tau) + \frac{3}{2}b + \hat{\beta}a$ *for all $i \in [n]$;*

    *4.* $\|\theta_i^t\| \leq b$ *for all $i \in [n]$;*

    *5.* $\|g^{t-1}\| \leq \sqrt{64L\Delta} + 3(B - \tau) + 3b + 3\hat{\beta}a$;

    *6.* $\|\overline{g}^{t-1}\| \leq \sqrt{64L\Delta} + 3(B - \tau) + 3b$.

*Then we have*

$$\|\nabla f_i(x^t) - v_i^t\| \leq \sqrt{4L\Delta} + \frac{3}{2}(B - \tau) + \frac{3}{2}b + \hat{\beta}a. \tag{32}$$

*Proof.* We have

$$\begin{aligned}
\|\nabla f_i(x^t) - v_i^t\| &\overset{(i)}{=} \|\nabla f_i(x^t) - (1-\beta)v_i^{t-1} - \beta\nabla f_i(x^t, \xi_i^t)\| \\
&\overset{(ii)}{\leq} (1-\beta)\|\nabla f_i(x^t) - v_i^{t-1}\| + \beta\|\nabla f_i(x^t) - \nabla f_i(x^t, \xi_i^t)\| \\
&\overset{(iii)}{\leq} (1-\beta)L\gamma\|g^{t-1}\| + (1-\beta)\|\nabla f_i(x^{t-1}) - v_i^{t-1}\| + \beta\|\theta_i^t\| \\
&\overset{(iv)}{\leq} (1-\beta)L\gamma\left(\sqrt{64L\Delta} + 3(B - \tau) + 3b + 3\hat{\beta}a\right) \\
&\quad + (1-\beta)\left(\sqrt{4L\Delta} + \frac{3}{2}(B - \tau) + \frac{3}{2}b + \hat{\beta}a\right) + \beta b \\
&= (8L\gamma + 2(1-\beta))\sqrt{L\Delta} + (3L\gamma + {}^{3(1-\beta)}\!/\!{}_2)(B - \tau) \\
&\quad + (3L\gamma(1-\beta) + {}^3\!/\!{}_2(1-\beta) + \beta)b + (3L\gamma\hat{\beta} + (1-\beta)\hat{\beta})a,
\end{aligned}$$

where $(i)$ follows from the update rule of $v_i^t$, $(ii)$ from the triangle inequality, $(iii)$ from triangle inequality, smoothness, and the update rule of $x^t$, $(iv)$ from assumptions 2-4 of the lemma. We notice that

$$\begin{aligned}
8L\gamma + 2(1-\beta) &\leq 2 \Leftarrow 4L\gamma \leq \beta, \\
3L\gamma + \frac{3}{2}(1-\beta) &\leq \frac{3}{2} \Leftarrow 2L\gamma \leq \beta, \\
3L\gamma + \frac{3}{2}(1-\beta) + \beta &\leq \frac{3}{2} \Leftarrow 6L\gamma \leq \beta, \\
3L\gamma\hat{\beta} + (1-\beta)\hat{\beta} &\leq \hat{\beta} \Leftarrow 3L\gamma \leq \beta,
\end{aligned}$$

where the last inequalities in each line are satisfied for $\beta$, satisfying the conditions of the lemma. $\square$

**Lemma F.4.** *Let each $f_i$ be $L$-smooth, $\Delta \geq \Phi^0$, $B > \tau$. Assume that the following inequalities hold for the iterates generated by* `Clip21-SGD2M`

    *1.* $\gamma \leq \frac{1}{12L}$;

    *2.* $\hat{\beta} \leq \min\{\frac{\sqrt{L\Delta}}{a}, 1\}$;

    *3.* $\|v_i^t - g_i^{t-1}\| \leq B$ *for all $i \in [n]$;*

    *4.* $\|g^{t-1}\| \leq \sqrt{64L\Delta} + 3(B - \tau) + 3b + \hat{\beta}a$;

    *5.* $\|\overline{g}^{t-1}\| \leq \sqrt{64L\Delta} + 3(B - \tau) + 3b$;

6. $\|\nabla f_i(x^{t-1}) - v_i^{t-1}\| \le \sqrt{4L\Delta} + \frac{3}{2}(B - \tau) + \frac{3}{2}b + \hat{\beta}a$ *for all* $i \in [n]$;

7. $\Phi^{t-1} \le 2\Delta$;

8. $\|\theta_i^t\| \le b$ *for all* $i \in [n]$.

*Then we have*

$$\|\overline{g}^t\| \le \sqrt{64L\Delta} + 3(B - \tau) + 3b.$$

*Proof.* We have

$$\|\overline{g}^t\| \overset{(i)}{=} \left\| \overline{g}^{t-1} + \frac{\hat{\beta}}{n} \sum_{i=1}^{n} \mathrm{clip}_\tau(v_i^t - g_i^{t-1}) \right\|$$

$$= \left\| \hat{\beta}\nabla f(x^t) + \hat{\beta}(v^t - \nabla f(x^t)) + (1 - \hat{\beta})\overline{g}^{t-1} + \frac{\hat{\beta}}{n} \sum_{i=1}^{n} [\mathrm{clip}_\tau(v_i^t - g_i^{t-1}) - (v_i^t - g_i^{t-1})] \right\|$$

$$\overset{(ii)}{\le} \hat{\beta}\|\nabla f(x^t)\| + \frac{\hat{\beta}}{n} \sum_{i=1}^{n} \|v_i^t - \nabla f_i(x^t)\| + (1 - \hat{\beta})\|\overline{g}^{t-1}\|$$

$$+ \frac{\hat{\beta}}{n} \sum_{i=1}^{n} \|\mathrm{clip}_\tau(v_i^t - g_i^{t-1}) - (v_i^t - g_i^{t-1})\|$$

$$\overset{(iii)}{\le} \hat{\beta}\|\nabla f(x^{t-1})\| + \hat{\beta}L\gamma\|g^{t-1}\| + \frac{\hat{\beta}}{n} \sum_{i=1}^{n} \|(1 - \beta)v_i^{t-1} + \beta\nabla f_i(x^t, \xi_i^t) - \nabla f_i(x^t)\|$$

$$+ (1 - \hat{\beta})\|\overline{g}^{t-1}\| + \frac{\hat{\beta}}{n} \sum_{i=1}^{n} \max\{0, \|v_i^t - g_i^{t-1}\| - \tau\}$$

$$\overset{(iv)}{\le} \hat{\beta}\sqrt{2L(f(x^{t-1}) - f^*)} + \hat{\beta}L\gamma\|g^{t-1}\| + (1 - \hat{\beta})\|\overline{g}^{t-1}\| + \hat{\beta}(B - \tau)$$

$$+ \frac{\hat{\beta}}{n} \sum_{i=1}^{n} \left( (1 - \beta)[\|v_i^{t-1} - \nabla f_i(x^{t-1})\| + \|\nabla f_i(x^{t-1}) - \nabla f_i(x^t)\|] + \beta\|\nabla f_i(x^t) - \nabla f_i(x^t, \xi_i^t)\| \right),$$

where $(i)$ follows from the update rule of each $g_i^t$, $(ii)$ – from the triangle inequality, $(iii)$ – from the update of $v_i^t$ and properties of clipping from Lemma C.3, $(iv)$ – from $L$-smoothness, assumption 3 of the lemma, and triangle inequality. Now we use assumptions 4-7 to derive

$$\|\overline{g}^t\| \le \hat{\beta}\sqrt{4L\Delta} + \hat{\beta}L\gamma(2 - \beta)\left(\sqrt{64L\Delta} + 3(B - \tau) + 3b + \hat{\beta}a\right) + \hat{\beta}(B - \tau)$$

$$+ (1 - \hat{\beta})\left(\sqrt{64L\Delta} + 3(B - \tau) + 3b\right) + \hat{\beta}(1 - \beta)\left(\sqrt{4L\Delta} + \frac{3}{2}(B - \tau) + \frac{3}{2}b + \hat{\beta}a\right) + \hat{\beta}\beta b$$

$$= \sqrt{L\Delta}\left(2\hat{\beta} + 8L\gamma(2 - \beta)\hat{\beta} + 8(1 - \hat{\beta}) + 2\hat{\beta}(1 - \beta)\right) + a(L\gamma\hat{\beta}^2(2 - \beta) + \hat{\beta}^2)$$

$$+ (B - \tau)\left(3L\gamma\hat{\beta}(2 - \beta) + \hat{\beta} + 3(1 - \hat{\beta}) + \frac{3}{2}\hat{\beta}(1 - \beta)\right)$$

$$+ b(3L\gamma\hat{\beta}(2 - \beta) + 3(1 - \hat{\beta}) + {}^3\!/{}_2\hat{\beta}(1 - \beta)).$$

For the second term, we have

$$2L\gamma\hat{\beta}^2a + \hat{\beta}^2a \le 2L\gamma\hat{\beta}\sqrt{L\Delta} + \hat{\beta}\sqrt{L\Delta} = (2L\gamma\hat{\beta} + \hat{\beta})\sqrt{L\Delta},$$

where we use $\hat{\beta} \le \frac{\sqrt{L\Delta}}{a}$. Therefore, the second term should be added to the first term. Thus, we have for the term with $\sqrt{L\Delta}$

$$2L\gamma\hat{\beta} + \hat{\beta} + 2\hat{\beta} + 8L\gamma\hat{\beta}(2 - \beta) + 8(1 - \hat{\beta}) + 2\hat{\beta}(1 - \beta) \le 8$$

$$\Leftarrow 2L\gamma + 1 + 2 + 8L\gamma(2 - \beta) + 2(1 - \beta) \le 8$$

$$\Leftarrow 18L\gamma \le 3,$$

where the last inequality is satisfied by the choice of the stepsize $L\gamma \leq \frac{1}{12}$. For the third coefficient, we have

$$3L\gamma\hat{\beta}(2-\beta) + \hat{\beta} + 3(1-\hat{\beta}) + \frac{3}{2}\hat{\beta}(1-\beta) \leq 3 \Leftarrow 3L\gamma(2-\beta) + 1 + \frac{3}{2}(1-\beta) \leq 3 \Leftarrow 6L\gamma \leq \frac{1}{2},$$

where the last inequality is satisfied by the choice of the stepsize $L\gamma \leq \frac{1}{12}$. For the fourth coefficient, we have the same derivations as for the third one. This implies that

$$\|\bar{g}^t\| \leq 8\sqrt{L\Delta} + 3(B-\tau) + 3b,$$

which concludes the proof.

$\square$

**Lemma F.5.** *Let each $f_i$ be $L$-smooth, $\Delta \geq \Phi^0$, $B > \tau$, and $i \in \mathcal{I}_t := \{i \in [n] \mid \|v_i^t - g_i^{t-1}\| > \tau\}$. Assume that the following inequalities hold for the iterates generated by* `Clip21-SGD2M`

> *1.* $12L\gamma \leq 1$;
>
> *2.* $6L\gamma \leq \beta$;
>
> *3.* $\beta \leq \min\{\frac{3\hat{\beta}\tau}{64\sqrt{L\Delta}}, 1\}$;
>
> *4.* $\beta \leq \min\{\frac{\hat{\beta}\tau}{14(B-\tau)}, 1\}$;
>
> *5.* $\beta \leq \min\{\frac{\hat{\beta}\tau}{22b}, 1\}$;
>
> *6.* $\hat{\beta} \leq \min\{\frac{\sqrt{L\Delta}}{a}, 1\}$;
>
> *7.* $\|g^t\| \leq \sqrt{64L\Delta} + 3(B-\tau) + 3b + 3a$;
>
> *8.* $\|\theta_i^{t+1}\| \leq b$;
>
> *9.* $\|\nabla f_i(x^t) - v_i^t\| \leq \sqrt{4L\Delta} + \frac{3}{2}(B-\tau) + \frac{3}{2}b + \hat{\beta}a$.

*Then*

$$\|v_i^{t+1} - g_i^t\| \leq \|v_i^t - g_i^{t-1}\| - \frac{\hat{\beta}\tau}{2}. \tag{33}$$

*Proof.* Since $i \in \mathcal{I}_t$, then $\|v_i^t - g_i^{t-1}\| > \tau$ and from Lemma F.1 we have

$$\|v_i^{t+1} - g_i^t\| \leq (1-\hat{\beta})\|v_i^t - g_i^{t-1}\| + \hat{\beta}\|v_i^t - g_i^{t-1}\| - \hat{\beta}\tau + \beta L\gamma\|g^t\| + \beta\|\nabla f_i(x^t) - v_i^t\| + \beta\|\theta_i^{t+1}\|$$

$$\overset{(i)}{\leq} \|v_i^t - g_i^{t-1}\| - \hat{\beta}\tau + \beta L\gamma\left(\sqrt{64L\Delta} + 3(B-\tau) + 3b + 3\hat{\beta}a\right)$$

$$+ \beta\left(\sqrt{4L\Delta} + \frac{3}{2}(B-\tau) + \frac{3}{2}b + \hat{\beta}a\right) + \beta b$$

$$= \|v_i^t - g_i^{t-1}\| - \hat{\beta}\tau + (8\beta L\gamma + 2\beta)\sqrt{L\Delta} + (3L\gamma\beta + 3\beta/2)(B-\tau)$$

$$+ (3L\gamma\beta + 3\beta/2 + \beta)b + (3L\gamma\beta + \beta)\hat{\beta}a,$$

where $(i)$ follows from assumptions 6-8 of the lemma. Since $12L\gamma \leq 1$ we have

$$(8\beta L\gamma + 2\beta)\sqrt{L\Delta} \leq (2\beta/3 + 2\beta)\sqrt{L\Delta} = \frac{8}{3}\beta\sqrt{L\Delta} \leq \frac{\hat{\beta}\tau}{8},$$

where we used $\beta \leq \frac{3\hat{\beta}\tau}{64\sqrt{L\Delta}}$. Since $12L\gamma \leq 1$ we have

$$\left(3L\gamma\beta + \frac{3\beta}{2}\right)(B-\tau) \leq (\beta/4 + \frac{3\beta}{2})(B-\tau) = \frac{7}{4}\beta(B-\tau) \leq \frac{\hat{\beta}\tau}{8},$$

where we used $\beta \le \frac{\hat{\beta}\tau}{14(B-\tau)}$. Since $12L\gamma \le 1$ we have

$$(3L\gamma\beta + {}^{5\beta}\!/\!{}_2)b \le ({}^{\beta}\!/\!{}_4 + {}^{5\beta}\!/\!{}_2)\,b = \frac{11}{4}\beta b \le \frac{\hat{\beta}\tau}{8},$$

where we used $\beta \le \frac{\hat{\beta}\tau}{22b}$. Since $12L\gamma \le 1$ and $\hat{\beta} \le \frac{\sqrt{L\Delta}}{a}$ we have

$$\left(3L\gamma\beta + \beta\right)\hat{\beta}a \le ({}^{\beta}\!/\!{}_4 + \beta)\sqrt{L\Delta} = \frac{5}{4}\beta\sqrt{L\Delta} \le \frac{\hat{\beta}\tau}{8},$$

where we used $\beta \le \frac{\hat{\beta}\tau}{22b}$. Thus we have

$$\|v_i^{t+1} - g_i^t\| \le \|v_i^t - g_i^{t-1}\| - \hat{\beta}\tau + 4 \cdot \frac{\hat{\beta}\tau}{8} = \|v_i^t - g_i^{t-1}\| - \frac{\hat{\beta}\tau}{2},$$

which concludes the proof. $\qquad\square$

**Lemma F.6.** *Let* $\|\theta_i^{t+1}\| \le b$ *for all* $i \in [n]$. *Let each* $f_i$ *be* $L$-*smooth. Then, for the iterates generated by* `Clip21-SGD2M` *the quantity* $\widetilde{P}^t := \frac{1}{n}\sum_{i=1}^n \|v_i^t - \nabla f_i(x^t)\|^2$ *decreases as*

$$\widetilde{P}^{t+1} \le (1-\beta)\widetilde{P}^t + \frac{3L^2}{\beta}R^t + \beta^2 b^2 + \frac{2}{n}\beta(1-\beta)\sum_{i=1}^n \langle v_i^t - \nabla f_i(x^{t+1}), \theta_i^{t+1}\rangle, \qquad (34)$$

*where* $R^t := \|x^{t+1} - x^t\|$ *and* $\theta_i^t := \nabla f_i(x^t, \xi_i^t) - \nabla f_i(x^t)$.

*Proof.* We have

$$
\begin{aligned}
\|v_i^{t+1} - \nabla f_i(x^{t+1})\|^2 &\overset{(i)}{=} \|(1-\beta)v_i^t + \beta\nabla f_i(x^{t+1}, \xi_i^{t+1}) - \nabla f_i(x^{t+1})\|^2 \\
&= \|(1-\beta)(v_i^t - \nabla f_i(x^{t+1})) + \beta(\nabla f_i(x^{t+1}, \xi_i^{t+1}) - \nabla f_i(x^{t+1}))\|^2 \\
&= (1-\beta)^2\|v_i^t - \nabla f_i(x^{t+1})\|^2 + \beta^2\|\theta_i^{t+1}\|^2 \\
&\quad + 2\beta(1-\beta)\langle v_i^t - \nabla f_i(x^{t+1}), \theta_i^{t+1}\rangle \\
&\overset{(ii)}{\le} (1-\beta)^2(1+{}^{\beta}\!/\!{}_2)\|v_i^t - \nabla f_i(x^t)\|^2 \\
&\quad + (1-\beta)^2(1+{}^2\!/\!{}_\beta)\|\nabla f_i(x^t) - \nabla f_i(x^{t+1})\|^2 + \beta^2 b^2 \\
&\quad + 2\beta(1-\beta)\langle v_i^t - \nabla f_i(x^{t+1}), \theta_i^{t+1}\rangle \\
&\overset{(iii)}{\le} (1-\beta)\|v_i^t - \nabla f_i(x^t)\|^2 + \frac{3L^2}{\beta}\|x^t - x^{t+1}\|^2 + \beta^2 b^2 \\
&\quad + 2\beta(1-\beta)\langle v_i^t - \nabla f_i(x^{t+1}), \theta_i^{t+1}\rangle,
\end{aligned}
$$

where $(i)$ follows from the update rule of $v_i^t$, $(ii)$ from $\|x + y\|^2 \le (1+r)\|x\|^2 + (1+r^{-1})\|y\|^2$ for any $x, y \in \mathbb{R}^d$ and $r > 0$, $(iii)$ from the smoothness and inequalities $(1-\beta)^2(1+{}^{\beta}\!/\!{}_2) \le (1-\beta)$ and $(1-\beta)^2(1+{}^2\!/\!{}_\beta) \le {}^3\!/\!{}_\beta$. Averaging the inequalities above across all $i \in [n]$, we get the lemma's statement. $\qquad\square$

Similarly, we can get the recursion for $P^t := \|v^t - \nabla f(x^t)\|^2$.

**Lemma F.7.** *Let* $\|\theta^{t+1}\| \le \frac{c}{\sqrt{n}}$ *for all* $i \in [n]$. *Let each* $f_i$ *be* $L$-*smooth. Then, for the iterates generated by* `Clip21-SGD2M` *the quantity* $P^t := \|v^t - \nabla f(x^t)\|^2$ *decreases as*

$$P^{t+1} \le (1-\beta)P^t + \frac{3L^2}{\beta}R^t + \beta^2\frac{c^2}{n} + 2\beta(1-\beta)\langle v^t - \nabla f(x^{t+1}), \theta^{t+1}\rangle,$$

*where* $R^t := \|x^{t+1} - x^t\|$ *and* $\theta^t := \frac{1}{n}\sum_{i=1}^n \theta_i^t = \frac{1}{n}\sum_{i=1}^n (\nabla f_i(x^t, \xi^t) - \nabla f_i(x^t))$.

*Proof.* For shortness, we denote $\nabla f(x^t, \xi^t) := \frac{1}{n}\sum_{i=1}^n \nabla f_i(x^t, \xi_i^t)$ and $\theta^t := \frac{1}{n}\sum_{i=1}^n(\nabla f_i(x^t, \xi^t) - \nabla f_i(x^t))$. Then, we have

$$\|v^{t+1} - \nabla f(x^{t+1})\|^2 \overset{(i)}{=} \|(1-\beta)v^t + \beta\nabla f(x^{t+1}, \xi^{t+1}) - \nabla f(x^{t+1})\|^2$$

$$= \|(1-\beta)(v^t - \nabla f(x^{t+1})) + \beta(\nabla f(x^{t+1}, \xi^{t+1}) - \nabla f(x^{t+1}))\|^2$$

$$= (1-\beta)^2\|v^t - \nabla f(x^{t+1})\|^2 + \beta^2\|\theta^{t+1}\|^2$$

$$\quad + 2\beta(1-\beta)\langle v^t - \nabla f(x^{t+1}), \theta^{t+1}\rangle$$

$$\overset{(ii)}{\leq} (1-\beta)^2(1 + \beta/2)\|v^t - \nabla f(x^t)\|^2$$

$$\quad + (1-\beta)^2(1 + 2/\beta)\|\nabla f(x^t) - \nabla f(x^{t+1})\|^2 + \beta^2\frac{c^2}{n}$$

$$\quad + 2\beta(1-\beta)\langle v^t - \nabla f(x^{t+1}), \theta_i^{t+1}\rangle$$

$$\overset{(iii)}{\leq} (1-\beta)\|v^t - \nabla f(x^t)\|^2 + \frac{3L^2}{\beta}\|x^t - x^{t+1}\|^2 + \beta^2\frac{c^2}{n}$$

$$\quad + 2\beta(1-\beta)\langle v^t - \nabla f(x^{t+1}), \theta^{t+1}\rangle,$$

where $(i)$ follows from the update rule of $v_i^t$, $(ii)$ from $\|x+y\|^2 \leq (1+r)\|x\|^2 + (1+r^{-1})\|y\|^2$ for any $x, y \in \mathbb{R}^d$ and $r > 0$, $(iii)$ from the smoothness and inequalities $(1-\beta)^2(1 + \beta/2) \leq (1-\beta)$ and $(1-\beta)^2(1 + 2/\beta) \leq 3/\beta$. $\square$

Next, we establish the recursion for $\widetilde{V}^t := \frac{1}{n}\sum_{i=1}^n \|g_i^t - v_i^t\|^2$.

**Lemma F.8.** *Let $\|\theta_i^t\| \leq b$ for all $i \in [n]$, each $f_i$ be $L$-smooth, and $\|v_i^t - g_i^{t-1}\| \leq B$ for all $i \in [n]$ and some $B > \tau$, and $\hat{\beta} \leq \frac{1}{2\eta}$[6]. Then, for the iterates generated by* `Clip21-SGD2M` *we have*

$$\|g_i^t - v_i^t\|^2 \leq (1-\hat{\beta}\eta)\|g_i^{t-1} - v_i^{t-1}\|^2 + \frac{4\beta^2}{\hat{\beta}\eta}\|v_i^{t-1} - \nabla f_i(x^{t-1})\|^2 + \frac{4\beta^2 L^2}{\hat{\beta}\eta}R^{t-1} + \beta^2 b^2 \quad (35)$$

$$\quad + 2(1-\hat{\beta}\eta)^2\beta\langle(g_i^{t-1} - v_i^{t-1}) + \beta(v_i^{t-1} - \nabla f_i(x^{t-1})), \theta_i^t\rangle$$

$$\quad + 2(1-\hat{\beta}\eta)^2\beta\langle\beta(\nabla f_i(x^{t-1}) - \nabla f_i(x^t)), \theta_i^t\rangle,$$

*where $R^t := \|x^{t+1} - x^t\|^2$ and $\eta := \frac{\tau}{B}$. Moreover, averaging the inequalities across all $i \in [n]$, we get*

$$\widetilde{V}^t \leq (1-\hat{\beta}\eta)\widetilde{V}^{t-1} + \frac{4\beta^2}{\hat{\beta}\eta}\widetilde{P}^{t-1} + \frac{4\beta^2 L^2}{\hat{\beta}\eta}R^{t-1} + \beta^2 b^2 \quad (36)$$

$$\quad + \frac{2}{n}(1-\hat{\beta}\eta)^2\beta\sum_{i=1}^n\langle(g_i^{t-1} - v_i^{t-1}) + \beta(v_i^{t-1} - \nabla f_i(x^{t-1})) + \beta(\nabla f_i(x^{t-1}) - \nabla f_i(x^t)), \theta_i^t\rangle,$$

*where $\widetilde{V}^t := \frac{1}{n}\sum_{i=1}^n\|g_i^t - v_i^t\|^2$ and $\widetilde{P}^t := \frac{1}{n}\sum_{i=1}^n\|v_i^t - \nabla f_i(x^t)\|^2$.*

*Proof.* Since $\|v_i^t - g_i^{t-1}\| \leq B$ and $B > \tau$, we have $\eta_i^t := \frac{\tau}{\|v_i^t - g_i^{t-1}\|} \geq \frac{\tau}{B} =: \eta \in (0,1)$. Thus, we have

$$\|g_i^t - v_i^t\|^2 \overset{(i)}{=} \|g_i^{t-1} + \hat{\beta}\,\text{clip}_\tau(v_i^t - g_i^{t-1}) - v_i^t\|^2$$

$$= \|\hat{\beta}(\text{clip}_\tau(v_i^t - g_i^{t-1}) - (v_i^t - g_i^{t-1})) + (1-\hat{\beta})(g_i^{t-1} - v_i^t)\|^2$$

$$\overset{(ii)}{\leq} (1-\hat{\beta})\|g_i^{t-1} - v_i^t\|^2 + \hat{\beta}\|\text{clip}_\tau(v_i^t - g_i^{t-1}) - (v_i^t - g_i^{t-1})\|^2$$

$$\overset{(iii)}{\leq} (1-\hat{\beta})\|g_i^{t-1} - v_i^t\|^2 + \hat{\beta}(1-\eta)^2\|g_i^{t-1} - v_i^t\|^2$$

$$= (1-\hat{\beta}\eta(2-\eta))\|g_i^{t-1} - v_i^t\|^2,$$

---

[6]Since $\eta \in (0,1)$, then this restriction is not necessary because the momentum parameter $\hat{\beta} \leq 1$ by default.

where $(i)$ follows from the update rule of $v_i^t$, $(ii)$ – from the convexity of $\|\cdot\|^2$, $(iii)$ – from the properties of the clipping operator in Lemma C.3. Let $\rho = 2\hat{\beta}\eta \le 1$. Then we have

$$
\begin{aligned}
\|g_i^t - v_i^t\|^2 &\le (1-\rho)\|g_i^{t-1} - v_i^t\|^2 \\
&\stackrel{(i)}{=} (1-\rho)\|g_i^{t-1} - (1-\beta)v_i^{t-1} - \beta\nabla f_i(x^t, \xi_i^t)\|^2 \\
&= (1-\rho)\|g_i^{t-1} - (1-\beta)v_i^{t-1} - \beta\theta_i^t - \beta\nabla f_i(x^t)\|^2 \\
&= (1-\rho)\|g_i^{t-1} - (1-\beta)v_i^{t-1} - \beta\nabla f_i(x^t)\|^2 + (1-\rho)\beta^2\|\theta_i^t\|^2 \\
&\quad - 2(1-\rho)\beta\langle g_i^{t-1} - (1-\beta)v_i^{t-1} - \beta\nabla f_i(x^t), \theta_i^t\rangle \\
&\stackrel{(ii)}{\le} (1-\rho)(1 + {\rho}/{2})\|g_i^{t-1} - v_i^{t-1}\|^2 + (1-\rho)(1 + {2}/{\rho})\beta^2\|v_i^{t-1} - \nabla f_i(x^t)\|^2 + \beta^2 b^2 \\
&\quad - 2(1-\rho)\beta\langle g_i^{t-1} - (1-\beta)v_i^{t-1} - \beta\nabla f_i(x^t), \theta_i^t\rangle \\
&\stackrel{(iii)}{\le} (1 - {\rho}/{2})\|g_i^{t-1} - v_i^{t-1}\|^2 + \frac{4\beta^2}{\rho}\|v_i^{t-1} - \nabla f_i(x^{t-1})\|^2 + \frac{4\beta^2 L^2}{\rho}R^{t-1} + \beta^2 b^2 \\
&\quad - 2(1-\rho)\beta\langle g_i^{t-1} - (1-\beta)v_i^{t-1} - \beta\nabla f_i(x^t), \theta_i^t\rangle,
\end{aligned}
$$

where $(i)$ follows from the update rule of $v_i^t$, $(ii)$ – from the inequality $\|a+b\|^2 \le (1+r)\|a\|^2 + (1 + r^{-1})\|b\|^2$ which holds for any $a, b \in \mathbb{R}^d$ and $r > 0$, and assumption of the lemma, $(iii)$ – from $L$-smoothness, Young's inequality $\|a + b\|^2 \le 2\|a\|^2 + 2\|b\|^2$. $\qquad\square$

**Theorem F.9** (Proof of Theorem 3.3). *Let $B := \max\{3\tau, \max_i\{\|\nabla f_i(x^0)\|\} + b\}$, Assumptions 1.1 and 1.2 hold, probability confidence level $\alpha \in (0,1)$, constants $a, b,$ and $c$ be defined as in (29), and $\Delta \ge \Phi^0$ for $\Phi^0$ defined in (9). Consider the run of Clip21-SGD2M (Algorithm 3) for $T$ iterations with DP noise variance $\sigma_\omega$. Assume the following inequalities hold*

1. **stepsize restrictions:**

    *i)* $12L\gamma \le 1$;
    *ii)*
    $$\frac{1}{3} - \frac{32\beta^2 L^2}{\hat{\beta}^2\eta^2}\gamma^2 - \frac{96L^2}{\hat{\beta}^2\eta^2}\gamma^2 \ge 0; \tag{37}$$

2. **momentum restrictions:**

    *i)* $6L\gamma = \beta$;
    *ii)* $\beta \le \min\{\frac{3\hat{\beta}\tau}{64\sqrt{L\Delta}}, 1\}$;
    *iii)* $\beta \le \min\{\frac{\hat{\beta}\tau}{14(B-\tau)}, 1\}$;
    *iv)* $\beta \le \min\{\frac{\hat{\beta}\tau}{22b}, 1\}$;
    *v)* $\hat{\beta} \le \min\{\frac{\sqrt{L\Delta}}{a}, \sqrt{L\Delta}\left(\frac{4}{\tau a^2 T}\right)^{1/3}, 1\}$;
    *vi)* $\beta, \hat{\beta} \in (0, 1]$;
    *vii)* *and momentum restrictions defined in (40), (41), (42), (43), (44), (46), (45), and (47);*

*Then, with probability $1 - \alpha$, we have $\frac{1}{T}\sum_{t=0}^{T-1}\|\nabla f(x^t)\|^2$ is bounded by*

$$
\widetilde{\mathcal{O}}\left(\left(\frac{L\Delta\sigma d\sigma_\omega^2 B^2}{(nT)^{3/2}\tau^2}\left(\sqrt{L\Delta} + B + \sigma\right)\right)^{1/3} + \sqrt{L\Delta}\left(\frac{\sqrt{d}\sigma_\omega}{\tau\sqrt{nT}} + \left(\frac{\sqrt{d}}{\tau\sqrt{Tn}}\right)^{2/3}\right)\left(\sqrt{L\Delta} + B + \sigma\right)\right),
$$

*where $\widetilde{\mathcal{O}}$ hides constant and polylogarithmic factors and higher order terms decreasing in $T$.*

*Proof.* For convenience, we define $\nabla f_i(x^{-1}, \xi_i^{-1}) = v_i^{-1} = g_i^{-1} = 0, \Phi^{-1} = \Phi^0$. Next, let us define an event $E^t$ for each $t \in \{0, \ldots, T\}$ such that the following inequalities hold for all $k \in \{0, \ldots, t\}$

1. $\|v_i^k - g_i^{k-1}\| \le B$ for $i \in \mathcal{I}_k$;

2. $\|g^k\|\leq \sqrt{64L\Delta} + 3(B-\tau) + 3b + 3\hat{\beta}a$;

3. $\|v_i^k - \nabla f_i(x^k)\|\leq \sqrt{4L\Delta} + \frac{3}{2}(B-\tau) + \frac{3}{2}b + \hat{\beta}a$;

4. $\|\overline{g}^k\|\leq \sqrt{64L\Delta} + 3(B-\tau) + 3b$;

5. $\|\theta_i^k\|\leq b$ for all $i \in [n]$ and $\|\theta^k\|\leq \frac{c}{\sqrt{n}}$;

6. $\left\|\frac{1}{n}\sum_{l=1}^{k+1}\sum_{i=1}^{n}\omega_i^l\right\| \leq a$;

7. $\Phi^k \leq 2\Delta$;

8.

$$
\frac{7}{8}\Delta \geq \frac{4\gamma\beta}{n\hat{\beta}\eta}(1-\eta)^2\sum_{l=0}^{k-1}\sum_{i=1}^{n}\langle(g_i^l - v_i^l) + \beta(v_i^l - \nabla f_i(x^l)) + \beta(\nabla f_i(x^l) - \nabla f_i(x^{l+1})), \theta_i^t\rangle
$$

$$
+ \frac{16\gamma\beta^2}{n\hat{\beta}^2\eta^2}(1-\beta)\sum_{l=0}^{k-1}\sum_{i=1}^{n}\langle v_i^l - \nabla f_i(x^l), \theta_i^{l+1}\rangle + 4\gamma(1-\beta)\sum_{l=0}^{k-1}\langle v^l - \nabla f(x^l), \theta^{l+1}\rangle
$$

$$
+ \frac{15\gamma\beta^2}{n\hat{\beta}^2\eta^2}(1-\beta)\sum_{l=0}^{k-1}\sum_{i=1}^{n}\langle\nabla f_i(x^l) - \nabla f_i(x^{l+1}), \theta_i^{l+1}\rangle
$$

$$
+ 4\gamma(1-\beta)\sum_{l=0}^{k-1}\langle\nabla f(x^l) - \nabla f(x^{l+1}), \theta^{l+1}\rangle.
$$

Then, we will derive the result by induction, i.e., using the induction w.r.t. $t$, we will show that $\Pr(E^t) \geq 1 - \frac{\alpha(t+1)}{T+1}$ for all $t \in \{0, \ldots, T-1\}$.

Before we move on to the induction part of the proof, we need to establish several useful bounds. Denote the events $\Theta_i^t, \Theta^t$ and $N^{t+1}$ as

$$
\Theta_i^t := \{\|\theta_i^t\|\geq b\}, \quad \Theta^t := \left\{\|\theta^t\|\geq \frac{c}{\sqrt{n}}\right\}, \quad \text{and} \quad N^{t+1} := \left\{\left\|\frac{1}{n}\sum_{l=1}^{t}\sum_{i=1}^{n}\omega_i^l\right\| \geq a\right\} \quad (38)
$$

respectively. From Assumption 1.2 we have (see (14))

$$
\Pr(\Theta_i^t) \leq 2\exp\left(-\frac{b^2}{2\sigma^2}\right) = \frac{\alpha}{6(T+1)n}
$$

where the last equality is by definition of $b^2$. Therefore, $\Pr(\overline{\Theta}_i^t) \geq 1 - \frac{\alpha}{6(T+1)n}$. Besides, notice that the constant $c$ in (29) can be viewed as

$$
c = (\sqrt{2} + 2b_3)\sigma \quad \text{where} \quad b_3^2 = 3\log\frac{6(T+1)}{\alpha}.
$$

Now, we can use Lemma C.1 to bound $\Pr(\Theta^t)$. Since all $\theta_i^t$ are independent $\sigma$-sub-Gaussian random vectors, then we have

$$
\Pr\left(\left\|\sum_{i=1}^{n}\theta_i^t\right\| \geq c\sqrt{n}\right) = \Pr\left(\|\theta^t\|\geq \frac{c}{\sqrt{n}}\right) \leq \exp(-b_3^2/3) = \frac{\alpha}{6(T+1)}.
$$

We also use Lemma C.1 to bound $\Pr(N^t)$. Indeed, since all $\omega_i^l$ are independent Gaussian random vectors, then we have

$$
\Pr\left(\left\|\sum_{l=1}^{t}\sum_{i=1}^{n}\omega_i^l\right\| \geq (\sqrt{2} + 2b_2)\sqrt{\sum_{l=1}^{t}\sum_{i=1}^{n}\sigma_\omega^2 d}\right) \leq \exp(-b_2^2/3) = \frac{\alpha}{6(T+1)}.
$$

with $b_2^2 = 3\log\left(\frac{6(T+1)}{\alpha}\right)$. This implies that

$$\Pr\left(\left\|\frac{1}{n}\sum_{l=1}^{t}\sum_{i=1}^{n}\omega_i^l\right\| \geq a\right) \leq \frac{\alpha}{6(T+1)}$$

due to the choice of $a$ from (29):

$$a = (\sqrt{2} + 2b_2)\sigma_\omega\sqrt{d}\sqrt{\frac{T}{n}}, \quad \text{where} \quad b_2^2 = 3\log\frac{6(T+1)}{\alpha}.$$

Note that with this choice of $a$ we have that the above is true for any $t \in \{1, \ldots, T\}$, i.e., $\Pr(N^t) \geq 1 - \frac{\alpha}{6(T+1)}$ for all $t \in \{1, \ldots, T\}$.

Now, we are ready to prove that $\Pr(E^t) \geq 1 - \frac{\alpha(t+1)}{T+1}$ for all $t \in \{0, \ldots, T-1\}$. First, we show that the base of induction holds.

**Base of induction.**

1. $\|v_i^0 - g_i^{-1}\| = \|v_i^0\| = \beta\|\nabla f_i(x^0, \xi_i^0)\| = \beta\|\theta_i^0\| + \beta\|\nabla f_i(x^0)\| \leq \frac{1}{2}b + \frac{1}{2}B \leq \frac{1}{2}B + \frac{1}{2}B = B$ holds with probability $1 - \frac{\alpha}{6(T+1)}$. Indeed, we have

$$\Pr(\Theta_i^0) \leq 2\exp\left(-\frac{b^2}{2\sigma^2}\right) = \frac{\alpha}{6(T+1)n}.$$

Therefore, we have

$$\Pr\left(\cap_{i=1}^{n}\overline{\Theta}_i^0\right) = 1 - \Pr\left(\cup_{i=1}^{n}\Theta_i^0\right) \geq 1 - \sum_{i=1}^{n}\Pr(\Theta_i^0) = 1 - n\frac{\alpha}{6(T+1)n} = 1 - \frac{\alpha}{6(T+1)}.$$

Moreover, we have

$$\Pr(\Theta^0) \leq \frac{\alpha}{6(T+1)}.$$

This means that the probability of the event that each $\left\|\frac{1}{n}\sum_{l=1}^{0}\sum_{i=1}^{n}\omega_i^l\right\| \leq a$, $\|\theta_i^0\| \leq b$, and $\|\theta^0\| \leq \frac{c}{\sqrt{n}}$, and is at least

$$1 - \frac{\alpha}{6(T+1)} - n\frac{\alpha}{6n(T+1)} - \frac{\alpha}{6(T+1)} = 1 - \frac{\alpha}{2(T+1)}.$$

2. We have already shown that

$$\Pr\left(\left\|\frac{1}{n}\sum_{i=1}^{n}\omega_i^1\right\| \geq a\right) \leq \frac{\alpha}{6(T+1)},$$

implying that $\left\|\frac{1}{n}\sum_{i=1}^{n}\omega_i^1\right\| \leq a$ with probability at least $1 - \frac{\alpha}{6(T+1)}$.

3. $g^0 = \frac{1}{n}\sum_{i=1}^{n}(g_i^{-1} + \hat{\beta}\,\mathrm{clip}_\tau(v_i^0 - g_i^{-1})) = \frac{1}{n}\sum_{i=1}^{n}\hat{\beta}\,\mathrm{clip}_\tau(\beta\nabla f_i(x^0, \xi_i^0))$. Therefore, we have

$$\|g^0\| \leq \left\|\frac{1}{n}\sum_{i=1}^{n}\hat{\beta}\beta\nabla f_i(x^0) + \hat{\beta}\beta\theta_i^0 + (\hat{\beta}\,\mathrm{clip}_\tau(\beta\nabla f_i(x^0, \xi_i^0)) - \hat{\beta}\beta\nabla f_i(x^0, \xi_i^0))\right\|$$

$$\leq \hat{\beta}\beta\|\nabla f(x^0)\| + \frac{\hat{\beta}\beta}{n}\sum_{i=1}^{n}\|\theta_i^0\| + \frac{1}{n}\sum_{i=1}^{n}\max\left\{0, \beta\|\nabla f_i(x^0, \xi_i^0)\| - \tau\right\}$$

$$\leq \hat{\beta}\beta\sqrt{2L(f(x^0) - f(x^*))} + \frac{\hat{\beta}\beta}{n}\sum_{i=1}^{n}\|\theta_i^0\| + \frac{\hat{\beta}}{n}\sum_{i=1}^{n}\max\left\{0, \beta\|\nabla f_i(x^0)\| + \beta\|\theta_i^0\| - \tau\right\}$$

$$\leq \frac{1}{2}\sqrt{2L\Phi^0} + \frac{2\hat{\beta}\beta}{n}\sum_{i=1}^{n}\|\theta_i^0\| + \frac{\hat{\beta}\beta}{n}\sum_{i=1}^{n}\|\nabla f_i(x^0)\| - \hat{\beta}\tau$$

$$\leq \sqrt{64L\Delta} + 2\hat{\beta}\beta b + \hat{\beta}\beta B - \hat{\beta}\tau$$

$$\leq \sqrt{64L\Delta} + \frac{3}{2}B - \tau + b \leq \sqrt{64L\Delta} + 3(B - \tau) + \frac{3}{2}b + \hat{\beta}a.$$

The inequalities above again hold in $\cap_{i=1}^{n} \overline{\Theta}_i^0$, i.e., with probability at least $1 - \frac{\alpha}{6(T+1)}$. Note that for the base of induction we have $\overline{g}^0 = \overline{g}$, therefore, the condition 4 holds as well.

4. We have

$$\begin{aligned}
\|v_i^0 - \nabla f_i(x^0)\| &= \|\nabla \beta f_i(x^0, \xi_i^0) - \nabla f_i(x^0)\| \\
&\leq \beta \|\nabla f_i(x^0, \xi_i^0) - \nabla f_i(x^0)\| + (1-\beta) \|\nabla f_i(x^0)\| \\
&\leq \beta b + (1-\beta) B
\end{aligned}$$

This bound holds with probability at least $1 - \frac{\alpha}{6(T+1)}$ because it holds in $\cap_{i=1}^{n} \overline{\Theta}_i^0$.

5. Condition 7 of the induction assumption also hold, as $\Phi^0 \leq 2\Phi^0 \leq 2\Delta$ by the choice of $\Delta$.

6. Finally, condition 8 of the induction assumption holds since the RHS equals 0.

Therefore, we conclude that the conditions 1-8 hold with a probability of at least

$$\begin{aligned}
\Pr\left(\Theta^0 \cap \left(\cap_{i=1}^{n} \overline{\Theta}_i^0\right) \cap \overline{N}^t\right) &\geq 1 - \Pr(\Theta^0) - \sum_{i=1}^{n} \Pr(\Theta_i^0) - \Pr(N^0) \\
&\geq 1 - \frac{\alpha}{6(T+1)} - n \cdot \frac{\alpha}{6n(T+1)} - \frac{\alpha}{6(T+1)} \\
&= 1 - \frac{\alpha}{2(T+1)} > 1 - \frac{\alpha}{T+1},
\end{aligned}$$

i.e., $\Pr(E^0) \geq 1 - \frac{\alpha}{T+1}$ holds. This is the base of the induction.

**Transition step of induction.** **Case** $|\mathcal{I}_{K+1}| > 0$. Assume that all events $\overline{\Theta}^{K+1}, \overline{\Theta}_i^{K+1}$ and $\overline{N}^{K+1}$ take place, i.e., $\|\theta_i^{K+1}\| \leq b, \|\theta^{K+1}\| \leq \frac{c}{\sqrt{n}}$ for all $i \in [n]$ and $\left\|\frac{1}{n} \sum_{l=1}^{K} \sum_{i=1}^{n} \omega_i^l\right\| \leq a$. That is, we assume that event $\overline{\Theta}^{K+1} \cap \left(\cap_{i=1}^{n} \overline{\Theta}_i^{K+1}\right) \cap \overline{N}^{K+1} \cap E^K$ holds. Then, by the assumptions of the induction, from Lemma F.5 we get for all $i \in \mathcal{I}_{K+1}$

$$\|v_i^{K+1} - g_i^K\| \leq \|v_i^K - g_i^{K-1}\| - \frac{\hat{\beta}\tau}{2} \leq B - \frac{\hat{\beta}\tau}{2}.$$

Therefore, from Lemma F.2 we get that

$$\|g^{K+1}\| \leq \sqrt{64L\Delta} + 3(B-\tau) + 3b + 3\hat{\beta}a,$$

from Lemma F.4 we get that

$$\|\overline{g}^{K+1}\| \leq \sqrt{64L\Delta} + 3(B-\tau) + 3b,$$

and from Lemma F.3

$$\|\nabla f_i(x^{K+1}) - v_i^{K+1}\| \leq \sqrt{4L\Delta} + \frac{3}{2}(B-\tau) + \frac{3}{2}b + \hat{\beta}a.$$

This means that conditions 1-6 in the induction assumption are also verified for the step $K+1$. Since for all $t \in \{0, \ldots, K+1\}$ inequalities 1-6 are verified, we can write for each $t \in \{0, \ldots, K\}$ by Lemmas C.2 and F.6 to F.8 the following

$$\Phi^{t+1} = \delta^{t+1} + \frac{2\gamma}{\hat{\beta}\eta}\widetilde{V}^{t+1} + \frac{8\gamma\beta}{\hat{\beta}^2\eta^2}\widetilde{P}^{t+1} + \frac{2\gamma}{\beta}P^{t+1}$$

$$\leq \delta^t - \frac{\gamma}{2}\|\nabla f(x^t)\|^2 - \frac{1}{4\gamma}R^t + 2\gamma\widetilde{V}^t + 2\gamma P^t + \gamma\hat{\beta}^2\|\Omega^t\|^2$$

$$+ \frac{2\gamma}{\hat{\beta}\eta}\left((1-\hat{\beta}\eta)\widetilde{V}^t + \frac{4\beta^2}{\hat{\beta}\eta}\widetilde{P}^t + \frac{4\beta^2 L^2}{\hat{\beta}\eta}R^t + \beta^2 b^2\right.$$

$$+ \left.\frac{2}{n}\beta(1-\hat{\beta}\eta)^2\sum_{i=1}^n\langle(g_i^t - v_i^t) + \beta(v_i^t - \nabla f_i(x^t)) + \beta(\nabla f_i(x^t) - \nabla f_i(x^{t+1})), \theta_i^{t+1}\rangle\right)$$

$$+ \frac{8\gamma\beta}{\hat{\beta}^2\eta^2}\left((1-\beta)\widetilde{P}^t + \frac{3L^2}{\beta}R^t + \beta^2 b^2 + \frac{2}{n}\beta(1-\beta)\sum_{i=1}^n\langle v_i^t - \nabla f_i(x^{t+1}), \theta_i^{t+1}\rangle\right)$$

$$+ \frac{2\gamma}{\beta}\left((1-\beta)P^t + \frac{3L^2}{\beta}R^t + \beta^2\frac{c^2}{n} + 2\beta(1-\beta)\langle v^t - \nabla f(x^{t+1}), \theta^{t+1}\rangle\right)$$

Rearranging terms, we get

$$\Phi^{t+1} \leq \delta^t - \frac{\gamma}{2}\|\nabla f(x^t)\|^2 + \frac{2\gamma}{\hat{\beta}\eta}\widetilde{V}^t\left(\hat{\beta}\eta + 1 - \hat{\beta}\eta\right) + \frac{8\gamma\beta}{\hat{\beta}^2\eta^2}\widetilde{P}^t(\beta + 1 - \beta) + \frac{2\gamma}{\beta}P^t(\beta + 1 - \beta)$$

$$- \frac{1}{4\gamma}R^t\left(1 - \frac{32L^2\beta^2}{\hat{\beta}^2\eta^2}\gamma^2 - \frac{96L^2}{\hat{\beta}^2\eta^2}\gamma^2 - \frac{24L^2}{\beta^2}\gamma^2\right) + b^2\left(\frac{2\beta^2\gamma}{\hat{\beta}\eta} + \frac{8\gamma\beta^3}{\hat{\beta}^2\eta^2}\right) + c^2\frac{2\gamma\beta}{n}$$

$$+ \frac{4\gamma\beta}{n\hat{\beta}\eta}(1-\hat{\beta}\eta)^2\sum_{i=1}^n\langle(g_i^t - v_i^t) + \beta(v_i^t - \nabla f_i(x^t)) + \beta(\nabla f_i(x^t) - \nabla f_i(x^{t+1})), \theta_i^{t+1}\rangle$$

$$+ \frac{16\gamma\beta^2}{n\hat{\beta}^2\eta^2}(1-\beta)\sum_{i=1}^n\langle v_i^t - \nabla f_i(x^t), \theta_i^{t+1}\rangle + 4\gamma(1-\beta)\langle v^t - \nabla f(x^t), \theta^{t+1}\rangle$$

$$+ \frac{16\gamma\beta^2}{n\hat{\beta}^2\eta^2}(1-\beta)\sum_{i=1}^n\langle\nabla f_i(x^t) - \nabla f_i(x^{t+1}), \theta_i^{t+1}\rangle$$

$$+ 4\gamma(1-\beta)\langle\nabla f(x^t) - \nabla f(x^{t+1}), \theta^{t+1}\rangle + \gamma\hat{\beta}^2\|\Omega^t\|^2.$$

Using momentum restriction $(i)$, stepsize restriction, momentum restriction $(i)$, $(ii)$ and assumption of the induction that $\|\Omega^t\| \leq a$, we get rid of the term with $R^t$ and obtain

$$\Phi^{t+1} \leq \Phi^t - \frac{\gamma}{2}\|\nabla f(x^t)\|^2 + b^2\left(\frac{2\beta^2\gamma}{\hat{\beta}\eta} + \frac{8\gamma\beta^3}{\hat{\beta}^2\eta^2}\right) + c^2\frac{2\gamma\beta}{n} + \frac{\beta}{6L}\hat{\beta}^2 a^2$$

$$+ \frac{4\gamma\beta}{n\hat{\beta}\eta}(1-\hat{\beta}\eta)^2\sum_{i=1}^n\langle(g_i^t - v_i^t) + \beta(v_i^t - \nabla f_i(x^t)) + \beta(\nabla f_i(x^t) - \nabla f_i(x^{t+1})), \theta_i^{t+1}\rangle$$

$$+ \frac{16\gamma\beta^2}{n\hat{\beta}^2\eta^2}(1-\beta)\sum_{i=1}^n\langle v_i^t - \nabla f_i(x^t), \theta_i^{t+1}\rangle + 4\gamma(1-\beta)\langle v^t - \nabla f(x^t), \theta^{t+1}\rangle$$

$$+ \frac{16\gamma\beta^2}{n\hat{\beta}^2\eta^2}(1-\beta)\sum_{i=1}^n\langle\nabla f_i(x^t) - \nabla f_i(x^{t+1}), \theta_i^{t+1}\rangle$$

$$+ 4\gamma(1-\beta)\langle\nabla f(x^t) - \nabla f(x^{t+1}), \theta^{t+1}\rangle.$$

Now we sum all the inequalities above using momentum restriction $(ii)$ for $t \in \{0, \dots, K\}$ and get

$$
\Phi^{K+1} \le \Phi^0 - \frac{\gamma}{2} \sum_{t=0}^{K} \|\nabla f(x^t)\|^2 + Kb^2 \left( \frac{2\beta^2 \gamma}{\hat{\beta}\eta} + \frac{8\gamma\beta^3}{\hat{\beta}^2\eta^2} \right) + Kc^2 \frac{2\gamma\beta}{n} + K \frac{\tau}{128L\sqrt{L\Delta}} \hat{\beta}^3 a^2
$$

$$
+ \frac{4\gamma\beta}{n\hat{\beta}\eta}(1 - \hat{\beta}\eta)^2 \sum_{t=0}^{K} \sum_{i=1}^{n} \langle (g_i^t - v_i^t) + \beta(v_i^t - \nabla f_i(x^t)) + \beta(\nabla f_i(x^t) - \nabla f_i(x^{t+1})), \theta_i^{t+1} \rangle
$$

$$
+ \frac{16\gamma\beta^2}{n\hat{\beta}^2\eta^2}(1 - \beta) \sum_{t=0}^{K} \sum_{i=1}^{n} \langle v_i^t - \nabla f_i(x^t), \theta_i^{t+1} \rangle + 4\gamma(1 - \beta) \sum_{t=0}^{K} \langle v^t - \nabla f(x^t), \theta^{t+1} \rangle
$$

$$
+ \frac{16\gamma\beta^2}{n\eta^2}(1 - \beta) \sum_{t=0}^{K} \sum_{i=1}^{n} \langle \nabla f_i(x^t) - \nabla f_i(x^{t+1}), \theta_i^{t+1} \rangle
$$

$$
+ 4\gamma(1 - \beta) \sum_{t=0}^{K} \langle \nabla f(x^t) - \nabla f(x^{t+1}), \theta^{t+1} \rangle. \tag{39}
$$

Rearranging terms, we get

$$
\frac{\gamma}{2} \sum_{t=0}^{K} \|\nabla f(x^t)\|^2 \le \Phi^0 - \Phi^{K+1} + Kb^2 \left( \frac{2\beta^2 \gamma}{\hat{\beta}\eta} + \frac{8\gamma\beta^3}{\hat{\beta}^2\eta^2} \right) + Kc^2 \frac{2\gamma\beta}{n} + \frac{K\tau}{128L\sqrt{L\Delta}} \hat{\beta}^3 a^2
$$

$$
+ \frac{4\gamma\beta}{n\hat{\beta}\eta}(1 - \hat{\beta}\eta)^2 \sum_{t=0}^{K} \sum_{i=1}^{n} \langle (g_i^t - v_i^t) + \beta(v_i^t - \nabla f_i(x^t)) + \beta(\nabla f_i(x^t) - \nabla f_i(x^{t+1})), \theta_i^{t+1} \rangle
$$

$$
+ \frac{16\gamma\beta^2}{n\hat{\beta}^2\eta^2}(1 - \beta) \sum_{t=0}^{K} \sum_{i=1}^{n} \langle v_i^t - \nabla f_i(x^t), \theta_i^{t+1} \rangle + 4\gamma(1 - \beta) \sum_{t=0}^{K} \langle v^t - \nabla f(x^t), \theta^{t+1} \rangle
$$

$$
+ \frac{16\gamma\beta^2}{n\hat{\beta}^2\eta^2}(1 - \beta) \sum_{t=0}^{K} \sum_{i=1}^{n} \langle \nabla f_i(x^t) - \nabla f_i(x^{t+1}), \theta_i^{t+1} \rangle
$$

$$
+ 4\gamma(1 - \beta) \sum_{t=0}^{K} \langle \nabla f(x^t) - \nabla f(x^{t+1}), \theta^{t+1} \rangle.
$$

Taking into account that $\frac{\gamma}{2} \sum_{t=0}^{K} \|\nabla f(x^t)\|^2 \ge 0$, we get that the event $E^K \cap \left( \cap_{i=1}^{n} \overline{\Theta}_i^{K+1} \right) \cap \overline{N}^t \cap \overline{\Theta}^{K+1}$ implies

$$
\Phi^{K+1} \le \Phi^0 + Kb^2 \left( \frac{2\beta^2 \gamma}{\hat{\beta}\eta} + \frac{8\gamma\beta^3}{\hat{\beta}^2\eta^2} \right) + Kc^2 \frac{2\gamma\beta}{n} + \frac{K\tau}{128L\sqrt{L\Delta}} \hat{\beta}^3 a^2
$$

$$
+ \frac{4\gamma\beta}{n\hat{\beta}\eta}(1 - \hat{\beta}\eta)^2 \sum_{t=0}^{K} \sum_{i=1}^{n} \langle (g_i^t - v_i^t) + \beta(v_i^t - \nabla f_i(x^t)) + \beta(\nabla f_i(x^t) - \nabla f_i(x^{t+1})), \theta_i^{t+1} \rangle
$$

$$
+ \frac{16\gamma\beta^2}{n\hat{\beta}^2\eta^2}(1 - \beta) \sum_{t=0}^{K} \sum_{i=1}^{n} \langle v_i^t - \nabla f_i(x^t), \theta_i^{t+1} \rangle + \frac{4\gamma(1 - \beta)}{n} \sum_{t=0}^{K} \sum_{i=1}^{n} \langle v^t - \nabla f(x^t), \theta_i^{t+1} \rangle
$$

$$
+ \frac{16\gamma\beta^2}{n\hat{\beta}^2\eta^2}(1 - \beta) \sum_{t=0}^{K} \sum_{i=1}^{n} \langle \nabla f_i(x^t) - \nabla f_i(x^{t+1}), \theta_i^{t+1} \rangle
$$

$$
+ \frac{4\gamma(1 - \beta)}{n} \sum_{t=0}^{K} \sum_{i=1}^{n} \langle \nabla f(x^t) - \nabla f(x^{t+1}), \theta_i^{t+1} \rangle.
$$

Next, we define the following random vectors:

$$\zeta_{1,i}^t := \begin{cases} g_i^t - v_i^t, & \text{if } \|g_i^t - v_i^t\| \le B \\ 0, & \text{otherwise} \end{cases},$$

$$\zeta_{2,i}^t := \begin{cases} v_i^t - \nabla f_i(x^t), & \text{if } \|v_i^t - \nabla f_i(x^t)\| \le \sqrt{4L\Delta} + \frac{3}{2}(B-\tau) + \frac{3}{2}b + \hat{\beta}a \\ 0, & \text{otherwise} \end{cases},$$

$$\zeta_{3,i}^t := \begin{cases} \nabla f_i(x^t) - \nabla f_i(x^{t+1}), & \text{if } \|\nabla f_i(x^t) - \nabla f_i(x^{t+1})\| \le L\gamma\left(\sqrt{64L\Delta} + 3(B-\tau) + 3b + 3\hat{\beta}a\right) \\ 0, & \text{otherwise} \end{cases},$$

$$\zeta_4^t := \begin{cases} v^t - \nabla f(x^t), & \text{if } \|v^t - \nabla f(x^t)\| \le \sqrt{4L\Delta} + \frac{3}{2}(B-\tau) + \frac{3}{2}b + \hat{\beta}a \\ 0, & \text{otherwise} \end{cases},$$

$$\zeta_5^t := \begin{cases} \nabla f(x^t) - \nabla f(x^{t+1}), & \text{if } \|\nabla f(x^t) - \nabla f(x^{t+1})\| \le L\gamma\left(\sqrt{64L\Delta} + 3(B-\tau) + 3b + 3\hat{\beta}a\right) \\ 0, & \text{otherwise} \end{cases}.$$

By definition, all introduced random vectors $\zeta_{l,i}^t, l \in [3], i \in [n], \zeta_{4,5}^t$ are bounded with probability 1. Moreover, by the definition of $E^t$ we get that the event $E^K \cap \overline{\Theta}^{K+1} \cap \left(\cap_{i=1}^n \overline{\Theta}_i^{K+1}\right) \cap \overline{N}^{K+1}$ implies

$$\zeta_{1,i}^t = g_i^t - v_i^t, \quad \zeta_{2,i}^t = v_i^t - \nabla f_i(x^t), \quad \zeta_{3,i}^t = \nabla f_i(x^t) - \nabla f_i(x^{t+1}),$$
$$\zeta_4^t = v^t - \nabla f(x^t), \quad \zeta_5^t = \nabla f(x^t) - \nabla f(x^{t+1}).$$

Therefore, the event $E^K \cap \overline{\Theta}^{K+1} \cap \left(\cap_{i=1}^n \overline{\Theta}_i^{K+1}\right) \cap \overline{N}^{K+1}$ implies

$$\Phi^{K+1} \le \Phi^0 + \underbrace{Kb^2\left(\frac{2\beta^2\gamma}{\hat{\beta}\eta} + \frac{8\gamma\beta^3}{\hat{\beta}^2\eta^2}\right) + Kc^2\frac{2\gamma\beta}{n} + K\gamma L\Delta\mathbb{1}_{a>0}}_{①} + \underbrace{\frac{4\gamma\beta}{n\hat{\beta}\eta}(1-\eta)^2\sum_{t=0}^K\sum_{i=1}^n\langle\zeta_{1,i}^t, \theta_i^{t+1}\rangle}_{②}$$

$$+ \underbrace{\frac{4\gamma\beta^2}{n\hat{\beta}\eta}(1-\hat{\beta}\eta)^2\sum_{t=0}^K\sum_{i=1}^n\langle\zeta_{2,i}^t, \theta_i^{t+1}\rangle + \frac{4\gamma\beta^2}{n\hat{\beta}\eta}(1-\hat{\beta}\eta)^2\sum_{t=0}^K\sum_{i=1}^n\langle\zeta_{3,i}^t, \theta_i^{t+1}\rangle}_{③ \qquad\qquad\qquad\qquad ④}$$

$$+ \underbrace{\frac{16\gamma\beta^2}{n\hat{\beta}^2\eta^2}(1-\beta)\sum_{t=0}^K\sum_{i=1}^n\langle\zeta_{2,i}^t, \theta_i^{t+1}\rangle}_{⑤} + \underbrace{\frac{4\gamma(1-\beta)}{n}\sum_{t=0}^K\sum_{i=1}^n\langle\zeta_4^t, \theta_i^{t+1}\rangle}_{⑥}$$

$$+ \underbrace{\frac{16\gamma\beta^2}{n\hat{\beta}^2\eta^2}(1-\beta)\sum_{t=0}^K\sum_{i=1}^n\langle\zeta_{3,i}^t, \theta_i^{t+1}\rangle}_{⑦} + \underbrace{\frac{4\gamma(1-\beta)}{n}\sum_{t=0}^K\sum_{i=1}^n\langle\zeta_5^t, \theta_i^{t+1}\rangle}_{⑧}.$$

BOUND OF THE TERM ①. Since $6L\gamma \le \beta$, for the term ① we have

$$Kb^2\left(\frac{2\beta^2\gamma}{\hat{\beta}\eta} + \frac{8\gamma\beta^3}{\hat{\beta}\eta^2}\right) + Kc^2\frac{2\gamma\beta}{n} + \frac{K\tau}{128L\sqrt{L\Delta}}\hat{\beta}^3a^2 \le Kb^2\left(\frac{\beta^3}{3L\hat{\beta}\eta} + \frac{4\beta^4}{3L\hat{\beta}^2\eta^2}\right) + Kc^2\frac{\beta^2}{3Ln}$$
$$+ \frac{K\tau}{128L\sqrt{L\Delta}}\hat{\beta}^3a^2.$$

By choosing $\beta$ such that

$$\beta \le \min\left\{\left(\frac{3L\Delta\hat{\beta}\eta}{32Tb^2}\right)^{1/3}, \left(\frac{3L\Delta\hat{\beta}^2\eta^2}{128Tb^2}\right)^{1/4}, \left(\frac{3L\Delta n}{32Tc^2}\right)^{1/2}\right\}, \qquad (40)$$

and $\hat{\beta}$ satisfying momentum restriction $(v)$ we get that

$$Kb^2\left(\frac{2\beta^2\gamma}{\hat{\beta}\eta} + \frac{8\gamma\beta^3}{\hat{\beta}^2\eta^2}\right) + Kc^2\frac{2\gamma\beta}{n} + \frac{K\tau}{128L\sqrt{L\Delta}}\hat{\beta}^3a^2 \le 4\cdot\frac{\Delta}{32} = \frac{\Delta}{8}.$$

Note that the worst dependency in the restriction on $\beta$ in $T$ is $\mathcal{O}(1/T)$ but it is present only in the case $a > 0$. The second worst on $\beta$ is $\mathcal{O}(1/T^{3/4})$ since $\hat{\beta} \sim \frac{1}{a} \sim \frac{1}{T}$ that comes from the second term in (40).

BOUND OF THE TERM ②.    For term ②, let us enumerate random variables as

$$\langle \zeta_{1,1}^0, \theta_1^1 \rangle, \ldots, \langle \zeta_{1,n}^0, \theta_n^1 \rangle, \langle \zeta_{1,1}^1, \theta_1^2 \rangle, \ldots, \langle \zeta_{1,n}^1, \theta_n^2 \rangle \ldots \langle \zeta_{1,1}^K, \theta_1^{K+1} \rangle, \ldots, \langle \zeta_{1,n}^K, \theta_n^{K+1} \rangle,$$

i.e., first by index $i$, then by index $t$. Then we have that the event $E^K \cap \left( \cap_{i=1}^n \overline{\Theta}_i^{K+1} \right)$ implies

$$\mathbb{E}\left[ \frac{4\gamma\beta}{n\hat{\beta}\eta}(1-\eta)^2 \langle \zeta_{1,i}^l, \theta_i^{l+1} \rangle \mid \langle \zeta_{1,i-1}^l, \theta_{i-1}^{l+1} \rangle, \ldots, \langle \zeta_{1,1}^l, \theta_1^{l+1} \rangle, \ldots, \langle \zeta_{1,1}^0, \theta_1^1 \rangle \right] = 0,$$

because $\{\theta_i^{l+1}\}_{i=1}^n$ are independent. Let

$$\sigma_2^2 := \frac{16\gamma^2\beta^2}{n^2\hat{\beta}^2\eta^2} \cdot B^2 \cdot \sigma^2.$$

Since $\theta_i^{l+1}$ is $\sigma$-sub-Gaussian random vector, for

$$\mathbb{E}\left[ \cdot \mid l, i-1 \right] := \mathbb{E}\left[ \cdot \mid \langle \zeta_{1,i-1}^l, \theta_{i-1}^{l+1} \rangle, \ldots, \langle \zeta_{1,1}^l, \theta_1^{l+1} \rangle, \ldots, \langle \zeta_{1,1}^0, \theta_1^1 \rangle \right]$$

we have

$$\mathbb{E}\left[ \exp\left( \left| \frac{1}{\sigma_2^2} \frac{16\gamma^2\beta^2}{n^2\hat{\beta}^2\eta^2}(1-\eta)^4 \langle \zeta_{1,i}^l, \theta_i^{l+1} \rangle^2 \right| \right) \mid l, i-1 \right]$$

$$\leq \mathbb{E}\left[ \exp\left( \frac{1}{\sigma_1^2} \frac{16\gamma^2\beta^2}{n^2\hat{\beta}^2\eta^2} \|\zeta_{1,i}^l\|^2 \cdot \|\theta_i^{l+1}\|^2 \right) \mid l, i-1 \right]$$

$$\leq \mathbb{E}\left[ \exp\left( \frac{1}{\sigma_2^2} \frac{16\gamma^2\beta^2}{n^2\hat{\beta}^2\eta^2} \cdot B^2 \|\theta_i^{l+1}\|^2 \right) \mid l, i-1 \right]$$

$$\leq \mathbb{E}\left[ \exp\left( \frac{n^2\hat{\beta}^2\eta^2}{16\gamma^2\beta^2 \cdot B^2 \cdot \sigma^2} \frac{16\gamma^2\beta^2}{n^2\hat{\beta}^2\eta^2} \cdot B^2 \|\theta_i^{l+1}\|^2 \right) \mid l, i-1 \right]$$

$$= \mathbb{E}\left[ \exp\left( \frac{\|\theta_i^{l+1}\|^2}{\sigma^2} \mid l, i-1 \right) \right] \leq \exp(1).$$

Therefore, we have by Lemma C.1 with $\sigma_k^2 \equiv \sigma_2^2$ that

$$\Pr\left( \frac{4\gamma\beta}{n\hat{\beta}\eta}(1-\hat{\beta}\eta)^2 \left\| \sum_{t=0}^K \sum_{i=1}^n \langle \zeta_{1,i}^t, \theta_i^{t+1} \rangle \right\| \geq (\sqrt{2} + \sqrt{2}b_1)\sqrt{\sum_{t=0}^K \sum_{i=1}^n \frac{16B^2\gamma^2\beta^2\sigma^2}{n^2\hat{\beta}^2\eta^2}} \right)$$

$$\leq \exp(-b_1^2/3)$$

$$= \frac{\alpha}{14(T+1)}$$

with $b_1^2 = 3\log\left( \frac{14(T+1)}{\alpha} \right)$. Note that since $6L\gamma \leq \beta$

$$(\sqrt{2} + \sqrt{2}b_1)\sqrt{\sum_{t=0}^K \sum_{i=1}^n \frac{16B^2\gamma^2\beta^2\sigma^2}{n^2\hat{\beta}^2\eta^2}} \leq (\sqrt{2} + \sqrt{2}b_1)\sqrt{\sum_{t=0}^K \sum_{i=1}^n \frac{4B^2\beta^4\sigma^2}{9L^2n^2\hat{\beta}^2\eta^2}}$$

$$= (\sqrt{2} + \sqrt{2}b_1)\frac{2B\beta^2\sigma}{3Ln\hat{\beta}\eta}\sqrt{(K+1)n}$$

$$\leq \frac{\Delta}{8},$$

because we choose $\beta$ such that

$$\beta \leq \left( \frac{3L\Delta\sqrt{n}\hat{\beta}\eta}{16\sqrt{2}(1+b_1)B\sigma\sqrt{T}} \right)^{1/2}, \quad \text{and} \quad K+1 \leq T. \tag{41}$$

This implies that

$$\Pr\left(\frac{4\gamma\beta}{n\hat{\beta}\eta}(1-\hat{\beta}\eta)^2\left\|\sum_{t=0}^{K}\sum_{i=1}^{n}\langle\zeta_{1,i}^t,\theta_i^{t+1}\rangle\right\|\geq\frac{\Delta}{8}\right)\leq\frac{\alpha}{14(T+1)}$$

with this choice of momentum parameter. The dependency of (41) on $T$ is $\widetilde{\mathcal{O}}(1/T^{3/4})$ since $\hat{\beta}\sim\frac{1}{T}$.

BOUND OF THE TERM ③.    The bound in this case is similar to the previous one. Let

$$\sigma_3^2:=\frac{16\gamma^2\beta^4}{n^2\hat{\beta}^2\eta^2}\cdot\left(\sqrt{4L\Delta}+\frac{3}{2}(B-\tau)+\frac{3}{2}b+\hat{\beta}a\right)^2\cdot\sigma^2.$$

Then,

$$\mathbb{E}\left[\exp\left(\left|\frac{1}{\sigma_3^2}\frac{16\gamma^2\beta^4}{n^2\hat{\beta}^2\eta^2}(1-\hat{\beta}\eta)^4\langle\zeta_{2,i}^l,\theta_i^{l+1}\rangle^2\right|\right)\mid l,i-1\right]$$

$$\leq\mathbb{E}\left[\exp\left(\frac{1}{\sigma_3^2}\frac{16\gamma^2\beta^4}{n^2\hat{\beta}^2\eta^2}\|\zeta_{2,i}^l\|^2\cdot\|\theta_i^{l+1}\|^2\right)\right]$$

$$\leq\mathbb{E}\left[\exp\left(\frac{1}{\sigma_3^2}\frac{16\gamma^2\beta^4}{n^2\hat{\beta}^2\eta^2}\cdot\left(\sqrt{4L\Delta}+\frac{3}{2}(B-\tau)+\frac{3}{2}b+\hat{\beta}a\right)^2\cdot\|\theta_i^{l+1}\|^2\right)\mid l,i-1\right]$$

$$\leq\mathbb{E}\left[\exp\left(\left[\frac{16\gamma^2\beta^4}{n^2\hat{\beta}^2\eta^2}\cdot\left(\sqrt{4L\Delta}+\frac{3}{2}(B-\tau)+\frac{3}{2}b+\hat{\beta}a\right)^2\cdot\sigma^2\right]^{-1}\cdot\right.\right.$$

$$\left.\left.\frac{16\gamma^2\beta^4}{n^2\hat{\beta}^2\eta^2}\cdot\left(\sqrt{4L\Delta}+\frac{3}{2}(B-\tau)+\frac{3}{2}b+\hat{\beta}a\right)^2\cdot\|\theta_i^{l+1}\|^2\right)\mid l,i-1\right]$$

$$=\mathbb{E}\left[\exp\left(\frac{\|\theta_i^{l+1}\|^2}{\sigma^2}\right)\mid l,i-1\right]\leq\exp(1).$$

Therefore, we have by Lemma C.1 that

$$\Pr\left[\frac{4\gamma\beta^2}{n\hat{\beta}\eta}(1-\hat{\beta}\eta)^2\left\|\sum_{t=0}^{K}\sum_{i=1}^{n}\langle\zeta_{2,i}^t,\theta_i^{t+1}\rangle\right\|\right.$$

$$\left.\geq(\sqrt{2}+\sqrt{2}b_1)\sqrt{\sum_{t=0}^{K}\sum_{i=1}^{n}\frac{16\gamma^2\beta^4\sigma^2}{n^2\hat{\beta}^2\eta^2}\cdot\left(\sqrt{4L\Delta}+\frac{3}{2}(B-\tau)+\frac{3}{2}b+\hat{\beta}a\right)^2}\right]$$

$$\leq\exp(-b_1^2/3)=\frac{\alpha}{14(T+1)}.$$

Note that by using the restrictions $\hat{\beta}\leq\frac{\sqrt{L\Delta}}{a}$ and $6L\gamma\leq\beta$ we get

$$(\sqrt{2}+\sqrt{2}b_1)\sqrt{(K+1)n}\frac{4\gamma\beta^2\sigma}{\hat{\beta}\eta n}\left(\sqrt{4L\Delta}+\frac{3}{2}(B-\tau)+\frac{3}{2}b+\hat{\beta}a\right)$$

$$\leq(\sqrt{2}+\sqrt{2}b_1)\sqrt{(K+1)n}\frac{2\beta^3\sigma}{3L\hat{\beta}\eta n}\left(\sqrt{4L\Delta}+\frac{3}{2}(B-\tau)+\frac{3}{2}b+\sqrt{L\Delta}\right)$$

$$\leq\frac{\Delta}{8}$$

holds because we choose

$$\beta\leq\left(\frac{3L\Delta\hat{\beta}\eta\sqrt{n}}{16\sqrt{2}(1+b_1)\sigma\sqrt{T}\left(\sqrt{9L\Delta}+\frac{3}{2}(B-\tau)+\frac{3}{2}b\right)}\right)^{1/3},\qquad\text{and}\quad K+1\leq T.\quad(42)$$

This implies

$$\Pr\left(\frac{4\gamma\beta^2}{n\hat{\beta}\eta}(1-\hat{\beta}\eta)^2\left\|\sum_{t=0}^{K}\sum_{i=1}^{n}\langle\zeta_{2,i}^t,\theta_i^{t+1}\rangle\right\| \geq \frac{\Delta}{8}\right) \leq \frac{\alpha}{14(T+1)}.$$

Note that the worst dependency in the choice of $\beta$ w.r.t. $T$ is $\widetilde{\mathcal{O}}(1/T^{1/2})$ since $\hat{\beta}\sim\frac{1}{T}$.

BOUND OF THE TERM ④. The bound in this case is similar to the previous one. Let

$$\sigma_4^2 := \frac{16L^2\gamma^4\beta^4}{n^2\hat{\beta}^2\eta^2}\left(\sqrt{64L\Delta}+3(B-\tau)+3b+3\hat{\beta}a\right)^2\cdot\sigma^2.$$

Then we have

$$\mathbb{E}\left[\exp\left(\left|\frac{1}{\sigma_4^2}\frac{16\gamma^2\beta^4}{n^2\hat{\beta}^2\eta^2}(1-\hat{\beta}\eta)^4\langle\zeta_{3,i}^l,\theta_i^{l+1}\rangle^2\right|\right) \mid l,i-1\right]$$

$$\leq \mathbb{E}\left[\exp\left(\frac{1}{\sigma_4^2}\frac{16\gamma^2\beta^4}{n^2\hat{\beta}^2\eta^2}\|\zeta_{3,i}^l\|^2\cdot\|\theta_i^{l+1}\|^2\right) \mid l,i-1\right]$$

$$\leq \mathbb{E}\left[\exp\left(\frac{1}{\sigma_4^2}\frac{16\gamma^2\beta^4}{n^2\hat{\beta}^2\eta^2}\cdot L^2\gamma^2\left(\sqrt{64L\Delta}+3(B-\tau)+3b+3a\right)^2\cdot\|\theta_i^{l+1}\|^2\right) \mid l,i-1\right]$$

$$\leq \mathbb{E}\left[\exp\left(\left[\frac{16L^2\gamma^4\beta^4}{n^2\hat{\beta}^2\eta^2}\left(\sqrt{64L\Delta}+3(B-\tau)+3b+3\hat{\beta}a\right)^2\cdot\sigma^2\right]^{-1}\right.\right.$$

$$\left.\left.\frac{16L^2\gamma^4\beta^4}{n^2\hat{\beta}^2\eta^2}\left(\sqrt{64L\Delta}+3(B-\tau)+3b+3\hat{\beta}a\right)^2\cdot\|\theta_i^{l+1}\|^2\right) \mid l,i-1\right]$$

$$= \mathbb{E}\left[\exp\left(\frac{\|\theta_i^{l+1}\|^2}{\sigma^2}\right)\right] \leq \exp(1).$$

Therefore, we have by Lemma C.1 that

$$\Pr\left(\frac{4\gamma\beta^2}{n\hat{\beta}\eta}(1-\hat{\beta}\eta)^2\left\|\sum_{t=0}^{K}\sum_{i=1}^{n}\langle\zeta_{3,i}^t,\theta_i^{t+1}\rangle\right\|\right.$$

$$\left.\geq (\sqrt{2}+\sqrt{2}b_1)\sqrt{\sum_{t=0}^{K}\sum_{i=1}^{n}\frac{16L^2\gamma^4\beta^4\sigma^2}{n^2\hat{\beta}^2\eta^2}\cdot\left(\sqrt{64L\Delta}+3(B-\tau+b)+3\hat{\beta}a\right)^2}\right)$$

$$\leq \exp(-b_1^2/3) = \frac{\alpha}{14(T+1)}.$$

Using the restrictions $\hat{\beta}\leq\frac{\sqrt{L\Delta}}{a}$ and $6L\gamma\leq\beta$ we get

$$(\sqrt{2}+\sqrt{2}b_1)\sqrt{(K+1)n}\frac{4L\gamma^2\beta^2\sigma}{\hat{\beta}\eta n}\left(\sqrt{64L\Delta}+3(B-\tau+b)+3\hat{\beta}a\right)$$

$$\leq\sqrt{2}(1+b_1)\sqrt{(K+1)n}\frac{\beta^4\sigma}{9L\hat{\beta}\eta n}\left(\sqrt{64L\Delta}+3(B-\tau+b)+3\sqrt{L\Delta}\right)$$

$$\leq\frac{\Delta}{8},$$

because we choose $\beta$ such that

$$\beta \leq \left(\frac{9L\Delta\hat{\beta}\eta\sqrt{n}}{8\sqrt{2}(1+b_1)\sigma\sqrt{T}\left(11\sqrt{L\Delta}+3(B-\tau+b)\right)}\right)^{1/4}, \qquad \text{and} \quad K+1\leq T. \quad (43)$$

This implies

$$\Pr\left(\frac{4\gamma\beta^2}{n\hat{\beta}\eta}(1-\hat{\beta}\eta)^2\left\|\sum_{t=0}^{K}\sum_{i=1}^{n}\langle\zeta_{2,i}^t,\theta_i^{t+1}\rangle\right\| \geq \frac{\Delta}{8}\right) \leq \frac{\alpha}{14(T+1)},$$

Note that the worst dependency in the choice of $\beta$ w.r.t. $T$ is $\widetilde{\mathcal{O}}(1/T^{3/8})$ since $\hat{\beta}\sim\frac{1}{T}$.

BOUND OF THE TERM ⑤. The bound in this case is similar to the previous one. Let

$$\sigma_5^2 := \frac{256\gamma^2\beta^4}{n^2\hat{\beta}^4\eta^4} \cdot \left(\sqrt{4L\Delta} + \frac{3}{2}(B-\tau) + \frac{3}{2}b + \hat{\beta}a\right)^2 \cdot \sigma^2.$$

Then we have

$$\mathbb{E}\left[\exp\left(\left|\frac{1}{\sigma_5^2}\frac{256\gamma^2\beta^4}{n^2\hat{\beta}^4\eta^4}(1-\beta)^2\langle\zeta_{2,i}^l, \theta_i^{l+1}\rangle^2\right|\right) \mid l, i-1\right]$$

$$\leq \mathbb{E}\left[\exp\left(\frac{1}{\sigma_5^2}\frac{256\gamma^2\beta^4}{n^2\hat{\beta}^4\eta^4}\|\zeta_{2,i}^l\|^2\cdot\|\theta_i^{l+1}\|^2\right) \mid l, i-1\right]$$

$$\leq \mathbb{E}\left[\exp\left(\frac{1}{\sigma_5^2}\frac{256\gamma^2\beta^4}{n^2\hat{\beta}^4\eta^4} \cdot \left(\sqrt{4L\Delta} + \frac{3}{2}(B-\tau) + \frac{3}{2}b + \hat{\beta}a\right)^2 \cdot \|\theta_i^{l+1}\|^2\right) \mid l, i-1\right]$$

$$= \mathbb{E}\left[\exp\left(\left[\frac{256\gamma^2\beta^4}{L^2n^2\hat{\beta}^4\eta^4} \cdot \left(\sqrt{4L\Delta} + \frac{3}{2}(B-\tau) + \frac{3}{2}b + \hat{\beta}a\right)^2 \cdot \sigma^2\right]^{-1}\right.\right.$$

$$\left.\left.\frac{256\gamma^2\beta^4}{n^2\hat{\beta}^4\eta^4} \cdot \left(\sqrt{4L\Delta} + \frac{3}{2}(B-\tau) + \frac{3}{2}b + \hat{\beta}a\right)^2 \cdot \|\theta_i^{l+1}\|^2\right) \mid l, i-1\right]$$

$$= \mathbb{E}\left[\exp\left(\frac{\|\theta_i^{l+1}\|^2}{\sigma^2}\right) \mid l, i-1\right] \leq \exp(1).$$

Therefore, we have by Lemma C.1 that

$$\Pr\left[\frac{16\gamma\beta^2}{n\hat{\beta}^2\eta^2}(1-\beta)\left\|\sum_{t=0}^{K}\sum_{i=1}^{n}\langle\zeta_{2,i}^t, \theta_i^{t+1}\rangle\right\|\right.$$

$$\geq (\sqrt{2} + \sqrt{2}b_1)\sqrt{\sum_{t=0}^{K}\sum_{i=1}^{n}\frac{256\gamma^2\beta^4\sigma^2}{n^2\hat{\beta}^4\eta^4}\left(\sqrt{4L\Delta} + \frac{3}{2}(B-\tau) + \frac{3}{2}b + \hat{\beta}a\right)^2}\right]$$

$$\leq \exp(-b_1^2/3) = \frac{\alpha}{14(T+1)}.$$

Using the restrictions $6L\gamma \leq \beta$ and $\hat{\beta} \leq \frac{\sqrt{L\Delta}}{a}$ we get

$$(\sqrt{2} + \sqrt{2}b_1)\sqrt{(K+1)n}\frac{16\gamma\beta^2\sigma}{n\hat{\beta}^2\eta^2}\left(\sqrt{4L\Delta} + \frac{3}{2}(B-\tau) + \frac{3}{2}b + \hat{\beta}a\right)$$

$$\leq (\sqrt{2} + \sqrt{2}b_1)\sqrt{(K+1)n}\frac{8\beta^3\sigma}{3Ln\hat{\beta}^2\eta^2}\left(\sqrt{4L\Delta} + \frac{3}{2}(B-\tau) + \frac{3}{2}b + \sqrt{L\Delta}\right)$$

$$\leq \frac{\Delta}{8}$$

because we choose $\beta$ such that

$$\beta \leq \left(\frac{3L\Delta\hat{\beta}^2\eta^2\sqrt{n}}{64\sqrt{2}(1+b_1)\sigma\sqrt{T}\left(3\sqrt{L\Delta} + \frac{3}{2}(B-\tau) + \frac{3}{2}b\right)}\right)^{1/3}, \qquad \text{and } K+1 \leq T. \qquad (44)$$

This implies

$$\Pr\left(\frac{16\gamma\beta^2}{n\hat{\beta}^2\eta^2}(1-\hat{\beta}\beta)\left\|\sum_{t=0}^{K}\sum_{i=1}^{n}\langle\zeta_{2,i}^t, \theta_i^{t+1}\rangle\right\| \geq \frac{\Delta}{8}\right) \leq \frac{\alpha}{14(T+1)}.$$

Note that the worst dependency in the choice of $\beta$ w.r.t. $T$ is $\widetilde{\mathcal{O}}(1/T^{5/6})$ since $\hat{\beta} \sim \frac{1}{T}$.

BOUND OF THE TERM ⑦. The bound in this case is similar to the previous one. Let

$$\sigma_7^2 := \frac{256L^2\gamma^4\beta^4}{n^2\hat{\beta}^4\eta^4}\left(\sqrt{64L\Delta} + 3(B - \tau + b) + 3\hat{\beta}a\right)^2 \cdot \sigma^2.$$

Then we have

$$\mathbb{E}\left[\exp\left(\left|\frac{1}{\sigma_7^2}\frac{256L^2\gamma^4\beta^4}{n^2\hat{\beta}^4\eta^4}(1-\beta)^2\langle\zeta_{3,i}^l, \theta_i^{l+1}\rangle^2\right|\right) \mid l, i-1\right]$$

$$\leq \mathbb{E}\left[\exp\left(\frac{1}{\sigma_7^2}\frac{256\gamma^2\beta^4}{n^2\hat{\beta}^4\eta^4}\|\zeta_{3,i}^l\|^2 \cdot \|\theta_i^{l+1}\|^2\right) \mid l, i-1\right]$$

$$\leq \mathbb{E}\left[\exp\left(\frac{256\gamma^2\beta^4}{n^2\hat{\beta}^4\eta^4} \cdot L^2\gamma^2\left(\sqrt{64L\Delta} + 3(B-\tau+b) + 3\hat{\beta}a\right)^2 \cdot \|\theta_i^{l+1}\|^2\right) \mid l, i-1\right]$$

$$\leq \mathbb{E}\left[\exp\left(\left[\frac{256L^2\gamma^4\beta^4}{n^2\hat{\beta}^4\eta^4}\left(\sqrt{64L\Delta} + 3(B-\tau+b) + 3\hat{\beta}a\right)^2 \cdot \sigma^2\right]^{-1}\right.\right.$$

$$\left.\left.\frac{256L^2\gamma^4\beta^4}{n^2\hat{\beta}^4\eta^4}\left(\sqrt{64L\Delta} + 3(B-\tau+b) + 3\hat{\beta}a\right)^2 \cdot \|\theta_i^{l+1}\|^2\right) \mid l, i-1\right]$$

$$= \mathbb{E}\left[\exp\left(\frac{\|\theta_i^{l+1}\|^2}{\sigma^2}\right) \mid l, i-1\right] \leq \exp(1).$$

Therefore, we have by Lemma C.1 that

$$\Pr\left[\frac{16\gamma\beta^2}{n\hat{\beta}^2\eta^2}(1-\beta)\left\|\sum_{t=0}^{K}\sum_{i=1}^{n}\langle\zeta_{3,i}^t, \theta_i^{t+1}\rangle\right\| \geq\right.$$

$$\left.(\sqrt{2} + \sqrt{2}b_1)\sqrt{\sum_{t=0}^{K}\sum_{i=1}^{n}\frac{256L^2\gamma^4\beta^4\sigma^2}{n^2\hat{\beta}^4\eta^4}\cdot\left(\sqrt{64L\Delta} + 3(B-\tau+b) + 3\hat{\beta}a\right)^2}\right]$$

$$\leq \exp(-b_1^2/3) = \frac{\alpha}{14(T+1)}.$$

Using the restrictions $6L\gamma \leq \beta$ and $\hat{\beta} \leq \frac{\sqrt{L\Delta}}{a}$ we get

$$(\sqrt{2} + \sqrt{2}b_1)\sqrt{(K+1)n}\frac{16L\gamma^2\beta^2\sigma}{\hat{\beta}^2\eta^2 n}\left(\sqrt{64L\Delta} + 3(B-\tau+b) + 3\hat{\beta}a\right)$$

$$\leq(\sqrt{2} + \sqrt{2}b_1)\sqrt{(K+1)n}\frac{4\beta^4\sigma}{9L\hat{\beta}^2\eta^2 n}\left(8\sqrt{L\Delta} + 3(B-\tau+b) + 3\sqrt{L\Delta}\right)$$

$$\leq\frac{\Delta}{8}$$

because we choose

$$\beta \leq \left(\frac{9L\Delta\hat{\beta}^2\eta^2\sqrt{n}}{32\sqrt{2}(1+b_1)\sigma\sqrt{T}\left(11\sqrt{L\Delta} + 3(B-\tau+B)\right)}\right)^{1/4}, \quad \text{and} \quad K+1 \leq T. \quad (45)$$

This implies

$$\Pr\left(\frac{8\gamma\beta^2}{n\eta^2}(1-\beta)\left\|\sum_{t=0}^{K}\sum_{i=1}^{n}\langle\zeta_{3,i}^t, \theta_i^{t+1}\rangle\right\| \geq \frac{\Delta}{8}\right) \leq \frac{\alpha}{14(T+1)}.$$

Note that the worst dependency in the choice of $\beta$ w.r.t. $T$ is $\widetilde{\mathcal{O}}(1/T^{5/8})$ since $\hat{\beta} \sim \frac{1}{T}$.

BOUND OF THE TERM ⑥.    The bound in this case is similar to the previous one. Let

$$\sigma_6^2 := \frac{16\gamma^2}{n^2}\left(\sqrt{4L\Delta} + \frac{3}{2}(B-\tau) + \frac{3}{2}b + \hat{\beta}a\right)^2 \cdot \sigma^2.$$

Then we have

$$\mathbb{E}\left[\exp\left(\left|\frac{1}{\sigma_6^2}\frac{16\gamma^2}{n^2}(1-\beta)^2\langle\zeta_4^l, \theta_i^{l+1}\rangle^2\right|\right) \mid l, i-1\right]$$

$$\leq \mathbb{E}\left[\exp\left(\frac{1}{\sigma_6^2}\frac{16\gamma^2}{n^2}\|\zeta_4^l\|^2\cdot\|\theta_i^{l+1}\|^2\right) \mid l, i-1\right]$$

$$\leq \mathbb{E}\left[\exp\left(\frac{1}{\sigma_6^2}\frac{16\gamma^2}{n^2}\left(\sqrt{4L\Delta} + \frac{3}{2}(B-\tau) + \frac{3}{2}b + \hat{\beta}a\right)^2 \cdot \|\theta_i^{l+1}\|^2\right) \mid l, i-1\right]$$

$$\leq \mathbb{E}\left[\exp\left(\left[\frac{16\gamma^2}{n^2}\left(\sqrt{4L\Delta} + \frac{3}{2}(B-\tau) + \frac{3}{2}b + \hat{\beta}a\right)^2 \cdot \sigma^2\right]^{-1}\right.\right.$$

$$\left.\left.\frac{16\gamma^2}{n^2}\left(\sqrt{4L\Delta} + \frac{3}{2}(B-\tau) + \frac{3}{2}b + \hat{\beta}a\right)^2 \cdot \|\theta_i^{l+1}\|^2\right) \mid l, i-1\right]$$

$$= \mathbb{E}\left[\exp\left(\frac{\|\theta_i^{t+1}\|^2}{\sigma^2}\right) \mid l, i-1\right] \leq \exp(1).$$

Therefore, we have by Lemma C.1 that

$$\Pr\left[\frac{\gamma(1-\beta)}{n}\left\|\sum_{t=0}^{K}\sum_{i=1}^{n}\langle\zeta_{4,i}^t, \theta_i^{t+1}\rangle\right\|\right.$$

$$\left.\geq (\sqrt{2} + \sqrt{2}b_1)\sqrt{\sum_{t=0}^{K}\sum_{i=1}^{n}\frac{16\gamma^2}{n^2}\sigma^2\cdot\left(\sqrt{4L\Delta} + \frac{3}{2}(B-\tau) + \frac{3}{2}b + \hat{\beta}a\right)^2}\right]$$

$$\leq \exp(-b_1^2/3) = \frac{\alpha}{14(T+1)},$$

Using the restrictions $6L\gamma \leq \beta$ and $\hat{\beta} \leq \frac{\sqrt{L\Delta}}{a}$ we get

$$(\sqrt{2} + \sqrt{2}b_1)\sqrt{(K+1)n} \cdot \frac{4\gamma}{n}\sigma\left(\sqrt{4L\Delta} + \frac{3}{2}(B-\tau) + \frac{3}{2}b + \hat{\beta}a\right)$$

$$\leq(\sqrt{2} + \sqrt{2}b_1)\sqrt{(K+1)n} \cdot \frac{2\beta}{3Ln}\sigma\left(\sqrt{4L\Delta} + \frac{3}{2}(B-\tau) + \frac{3}{2}b + \sqrt{L\Delta}\right)$$

$$\leq\frac{\Delta}{8}$$

because we choose $\beta$ such that

$$\beta \leq \left(\frac{3L\Delta\sqrt{n}}{16\sqrt{2}(1+b_1)\sigma\sqrt{T}\left(3\sqrt{L\Delta} + \frac{3}{2}(B-\tau) + \frac{3}{2}b\right)}\right), \qquad \text{and} \quad K+1 \leq T. \tag{46}$$

This implies

$$\Pr\left(\frac{4\gamma(1-\beta)}{n}\left\|\sum_{t=0}^{K}\sum_{i=1}^{n}\langle\zeta_{4,i}^t, \theta_i^{t+1}\rangle\right\| \geq \frac{\Delta}{8}\right) \leq \frac{\alpha}{14(T+1)}.$$

Note that the worst dependency in the choice of $\beta$ w.r.t. $T$ is $\widetilde{\mathcal{O}}(1/T^{1/2})$.

BOUND OF THE TERM ⑧.    The bound in this case is similar to the previous one. Let

$$\sigma_8^2 := \frac{16L^2\gamma^4}{n^2} \cdot \left(\sqrt{64L\Delta} + 3(B - \tau + b) + 3\hat{\beta}a\right)^2 \cdot \sigma^2.$$

Then we have

$$\mathbb{E}\left[\exp\left(\left|\frac{1}{\sigma_8^2}\frac{16\gamma^2}{n^2}(1-\beta)^2\langle\zeta_5^l, \theta_i^{l+1}\rangle^2\right|\right) \mid l, i-1\right]$$

$$\leq \mathbb{E}\left[\exp\left(\frac{1}{\sigma_8^2}\frac{16\gamma^2}{n^2}\|\zeta_5^l\|^2 \cdot \|\theta_i^{l+1}\|^2\right) \mid l, i-1\right]$$

$$\leq \mathbb{E}\left[\exp\left(\frac{1}{\sigma_8^2}\frac{16\gamma^2}{n^2}L^2\gamma^2\left(\sqrt{64L\Delta} + 3(B - \tau + b) + 3\hat{\beta}a\right)^2 \cdot \|\theta_i^{l+1}\|^2\right) \mid l, i-1\right].$$

Since $\theta_i^{l+1}$ is sub-Gaussian with parameter $\sigma^2$, then we can continue the chain of inequalities above using the definition of $\sigma_8^2$

$$\mathbb{E}\left[\exp\left(\left[\frac{16L^2\gamma^4}{n^2} \cdot \left(\sqrt{64L\Delta} + 3(B - \tau + b) + 3\hat{\beta}a\right)^2 \cdot \sigma^2\right]^{-1}\right.\right.$$

$$\left.\left.\frac{4L^2\gamma^4}{n^2} \cdot \left(\sqrt{64L\Delta} + 3(B - \tau + b) + 3\hat{\beta}a\right)^2 \cdot \|\theta_i^{l+1}\|^2\right) \mid l, i-1\right]$$

$$= \mathbb{E}\left[\exp\left(\frac{\|\theta_i^{l+1}\|^2}{\sigma^2}\right)\right] \leq \exp(1).$$

Therefore, we have by Lemma C.1 that

$$\Pr\left[\frac{4\gamma(1-\beta)}{n}\left\|\sum_{t=0}^{K}\sum_{i=1}^{n}\langle\zeta_{5,i}^t, \theta^{t+1}\rangle\right\|\right.$$

$$\geq (\sqrt{2} + \sqrt{2}b_1)\sqrt{\sum_{t=0}^{K}\sum_{i=1}^{n}\frac{16L^2\gamma^4}{n^2}\sigma^2 \cdot \left(\sqrt{64L\Delta} + 3(B - \tau + b) + 3\hat{\beta}a\right)^2}\right]$$

$$\leq \exp(-b_1^2/3) = \frac{\alpha}{14(T+1)}.$$

Using the restrictions $6L\gamma \leq \beta$ and $\hat{\beta} \leq \frac{\sqrt{L\Delta}}{a}$ we get

$$(\sqrt{2} + \sqrt{2}b_1)\sqrt{(K+1)n} \cdot \frac{4L\gamma^2}{n}\sigma\left(\sqrt{64L\Delta} + 3(B - \tau + b) + 3\hat{\beta}a\right)$$

$$\leq (\sqrt{2} + \sqrt{2}b_1)\sqrt{(K+1)n} \cdot \frac{\beta^2\sigma}{9Ln}\left(8\sqrt{L\Delta} + 3(B - \tau) + 3b + 3\sqrt{L\Delta}\right)$$

$$\leq \frac{\Delta}{8}$$

because we choose $\beta$ such that

$$\beta \leq \left(\frac{9L\Delta\sqrt{n}}{\sqrt{2}(1+b_1)\sigma\sqrt{T}\left(11\sqrt{L\Delta} + 3(B - \tau + b)\right)}\right)^{1/2} \quad \text{and} \quad K + 1 \leq T. \quad (47)$$

This implies

$$\Pr\left(4\gamma(1-\beta)\left\|\sum_{t=0}^{K}\sum_{i=1}^{n}\langle\zeta_{5,i}^t, \theta^{t+1}\rangle\right\| \geq \frac{\Delta}{8}\right) \leq \frac{\alpha}{14(T+1)}.$$

Note that the worst dependency w.r.t $T$ is $\widetilde{\mathcal{O}}(1/T^{1/4})$.

**Final probability.** Therefore, the probability event

$$\Omega := E^K \cap \overline{\Theta}^{K+1} \cap \left( \cap_{i=1}^n \overline{\Theta}_i^{K+1} \right) \cap \overline{N}^{K+1} \cap E_① \cap E_② \cap E_③ \cap E_④ \cap E_⑤ \cap E_⑥ \cap E_⑦ \cap E_⑧,$$

where each $E_①$-$E_⑧$ denotes that each of 1-8-th terms is smaller than $\frac{\Delta}{8}$, implies that

$$① + ② + ③ + ④ + ⑤ + ⑥ + ⑦ + ⑧ \le 8 \cdot \frac{\Delta}{8} = \Delta,$$

i.e., condition 7 in the induction assumption holds. Moreover, this also implies that

$$\Phi^{K+1} \le \Phi^0 + \Delta \le \Delta + \Delta = 2\Delta,$$

i.e., condition 6 in the induction assumption holds. The probability $\Pr(E_{K+1})$ can be lower bounded as follows

$$\Pr(E_{K+1}) \ge \Pr(\Omega)$$
$$= \Pr \left( E_K \cap \overline{\Theta}^{K+1} \cap \left( \cap_{i=1}^n \overline{\Theta}_i^{K+1} \right) \cap \overline{N}^{K+1} \cap E_① \cap E_② \cap E_③ \cap E_④ \cap E_⑤ \cap E_⑥ \right.$$
$$\left. \cap E_⑦ \cap E_⑧ \right)$$
$$= 1 - \Pr \left( \overline{E}_K \cup \Theta^{K+1} \cup \left( \cup_{i=1}^n \Theta_i^{K+1} \right) \cup N^{K+1} \cup \overline{E}_① \cup \overline{E}_② \cup \overline{E}_③ \cup \overline{E}_④ \cup \overline{E}_⑤ \cup \overline{E}_⑥ \right.$$
$$\left. \cup \overline{E}_⑦ \cup \overline{E}_⑧ \right)$$
$$\ge 1 - \Pr(\overline{E}_K) - \Pr(\Theta^{K+1}) - \sum_{i=1}^n \Pr(\Theta_i^{K+1}) - \Pr(N^{K+1}) - \Pr(\overline{E}_①) - \Pr(\overline{E}_②)$$
$$- \Pr(\overline{E}_③) - \Pr(\overline{E}_④) - \Pr(\overline{E}_⑤) - \Pr(\overline{E}_⑥) - \Pr(\overline{E}_⑦) - \Pr(\overline{E}_⑧)$$
$$\ge 1 - \frac{\alpha(K+1)}{T+1} - \frac{\alpha}{6(T+1)} - \sum_{i=1}^n \frac{\alpha}{6n(T+1)} - \frac{\alpha}{6(T+1)} - 0 - 7 \cdot \frac{\alpha}{14(T+1)}$$
$$= 1 - \frac{\alpha(K+2)}{T+1}.$$

This finalizes the transition step of induction. The result of the theorem follows by setting $K = T - 1$. Indeed, from (39) we obtain

$$\frac{\gamma}{2} \sum_{t=0}^{K} \|\nabla f(x^t)\|^2 \le \Phi^0 - \Phi^{K+1} + \Delta \le 2\Delta \Rightarrow \frac{1}{T} \sum_{t=0}^{T-1} \|\nabla f(x^t)\|^2 \le \frac{4\Delta}{\gamma T}. \tag{48}$$

**Final rate.** Translating momentum restrictions (40), (41), (42), (43), (44), (46), (45), and (47) to the stepsize restriction using $6L\gamma = \beta$ equality we get that the stepsize should satisfy

$$\gamma \le \frac{1}{L} \widetilde{\mathcal{O}} \left( \min \left\{ \underbrace{\left( \frac{L\Delta n}{T\sigma^2} \right)^{1/2}}_{\text{from term 1}}, \underbrace{\left( \frac{L\Delta \hat{\beta}^2 \eta^2}{T\sigma^2} \right)^{1/4}}_{}, \underbrace{\left( \frac{L\Delta \sqrt{n} \hat{\beta} \eta}{B\sigma\sqrt{T}} \right)^{1/2}}_{\text{from term 2}}, \underbrace{\left( \frac{L\Delta \sqrt{n} \hat{\beta} \eta}{\sigma(\sqrt{L\Delta} + B + \sigma)\sqrt{T}} \right)^{\frac{1}{3}}}_{\text{from term 3}}, \right.$$

$$\underbrace{\left( \frac{L\Delta \hat{\beta} \eta \sqrt{n}}{\sigma(\sqrt{L\Delta} + B + \sigma)\sqrt{T}} \right)^{1/4}}_{\text{from term 4}}, \underbrace{\left( \frac{L\Delta \hat{\beta}^2 \eta^2 \sqrt{n}}{\sigma(\sqrt{L\Delta} + B + \sigma)\sqrt{T}} \right)^{1/3}}_{\text{from term 5}}, \underbrace{\left( \frac{L\Delta \hat{\beta}^2 \eta^2 \sqrt{n}}{\sigma(\sqrt{L\Delta} + B + \sigma)\sqrt{T}} \right)^{1/4}}_{\text{from term 7}},$$

$$\left. \underbrace{\left( \frac{L\Delta \sqrt{n}}{\sigma(\sqrt{L\Delta} + B + \sigma)\sqrt{T}} \right)}_{\text{from term 6}}, \underbrace{\left( \frac{L\Delta \sqrt{n}}{\sigma(\sqrt{L\Delta} + B + \sigma)\sqrt{T}} \right)^{\frac{1}{2}}}_{\text{from term 8}} \right\} \right). \tag{49}$$

The worst power of $T$ comes from the term ⑤ and equals $\frac{1}{T^{5/6}}$. The second worst comes from terms ①, ②, and ④, and equals to $\gamma \leq \frac{1}{T^{3/4}}$ in the case $\hat{\beta} \sim \frac{1}{T}$. These terms give the rate of the form

$$
\widetilde{\mathcal{O}}\left( \frac{L\Delta}{T}\left( \frac{T\sigma^2}{L\Delta\hat{\beta}^2\eta^2} \right)^{1/4} + \frac{L\Delta}{T}\left( \frac{\sigma(\sqrt{L\Delta}+B+\sigma)\sqrt{T}}{L\Delta\hat{\beta}\eta\sqrt{n}} \right)^{1/3} \right.
$$
$$
\left. + \frac{L\Delta}{T}\left( \frac{\sigma(\sqrt{L\Delta}+B+\sigma)\sqrt{T}}{L\Delta\hat{\beta}^2\eta^2\sqrt{n}} \right)^{1/3} + \frac{L\Delta}{T}\left( \frac{B\sigma\sqrt{T}}{L\Delta\sqrt{n}\hat{\beta}\eta} \right)^{1/2} \right). \tag{50}
$$

In the case, when $\hat{\beta} = 1$ the worst dependency in (49) w.r.t. $T$ comes from the terms ① and ⑥. We also have restriction $\gamma \leq \mathcal{O}(1/L)$. All of those restrictions give the rate of the form

$$
\frac{L\Delta}{T}\widetilde{\mathcal{O}}\left( 1 + \frac{T^{1/2}\sigma}{L^{1/2}\Delta^{1/2}n^{1/2}} + \frac{\sigma(\sqrt{L\Delta}+B+\sigma)\sqrt{T}}{L\Delta\sqrt{n}} \right)
$$
$$
= \widetilde{\mathcal{O}}\left( \frac{L\Delta}{T} + \frac{\sqrt{L\Delta}\sigma}{\sqrt{nT}} + \frac{\sigma(\sqrt{L\Delta}+B+\sigma)}{\sqrt{nT}} \right)
$$
$$
= \widetilde{\mathcal{O}}\left( \frac{L\Delta}{T} + \frac{\sigma(\sqrt{L\Delta}+B+\sigma)}{\sqrt{nT}} \right). \tag{51}
$$

Choosing $\hat{\beta} \leq \sqrt{L\Delta}/a$ in (50), where $a$ is defined in (29), and setting $\eta = \frac{\tau}{B}$ we get

$$
\frac{L\Delta}{T} \cdot \widetilde{\mathcal{O}}\left( \left( \frac{T\sigma^2 B^2 a^2}{L^2\Delta^2\tau^2} \right)^{1/4} + \left( \frac{\sigma a B(\sqrt{L\Delta}+B+\sigma)\sqrt{T}}{L^{3/2}\Delta^{3/2}\tau\sqrt{n}} \right)^{1/3} + \left( \frac{\sigma a^2(\sqrt{L\Delta}+B+\sigma)B^2\sqrt{T}}{L^2\Delta^2\tau^2\sqrt{n}} \right)^{1/3} \right.
$$
$$
\left. + \left( \frac{aB^2\sigma\sqrt{T}}{L^{3/2}\Delta^{3/2}\sqrt{n}\tau} \right)^{1/2} \right)
$$
$$
= \frac{L\Delta}{T} \cdot \widetilde{\mathcal{O}}\left( \left( \frac{T\sigma^2 B^2 a^2}{L^2\Delta^2\tau^2} \right)^{1/4} + \left( \frac{\sigma a B\sqrt{T}}{L\Delta\tau\sqrt{n}} \right)^{1/3} + \left( \frac{\sigma a B^2\sqrt{T}}{L^{3/2}\Delta^{3/2}\tau\sqrt{n}} \right)^{1/3} + \left( \frac{\sigma^2 a B\sqrt{T}}{L^{3/2}\Delta^{3/2}\tau\sqrt{n}} \right)^{1/3} \right.
$$
$$
+ \left( \frac{\sigma a^2 B^2\sqrt{T}}{L^{3/2}\Delta^{3/2}\tau^2\sqrt{n}} \right)^{1/3} + \left( \frac{\sigma a^2 B^3\sqrt{T}}{L^2\Delta^2\tau^2\sqrt{n}} \right)^{1/3} + \left( \frac{\sigma^2 a^2 B^2\sqrt{T}}{L^2\Delta^2\tau^2\sqrt{n}} \right)^{1/3}
$$
$$
\left. + \left( \frac{aB^2\sigma\sqrt{T}}{L^{3/2}\Delta^{3/2}\sqrt{n}\tau} \right)^{1/2} \right).
$$

Now we use the exact value for $a$ to derive

$$\widetilde{\mathcal{O}}\left(\left(\frac{L^4\Delta^4 T\sigma^2 B^2 d\sigma_\omega^2 \frac{T}{n}}{T^4 L^2\Delta^2\tau^2}\right)^{1/4} + \left(\frac{L^3\Delta^3\sigma d^{1/2}\sigma_\omega \frac{T^{1/2}}{n^{1/2}}B\sqrt{T}}{T^3 L\Delta\tau\sqrt{n}}\right)^{1/3} + \left(\frac{L^3\Delta^3\sigma d^{1/2}\sigma_\omega \frac{T^{1/2}}{n^{1/2}}B^2\sqrt{T}}{T^3 L^{3/2}\Delta^{3/2}\tau\sqrt{n}}\right)^{1/3}\right.$$

$$+ \left(\frac{L^3\Delta^3\sigma^2 d^{1/2}\sigma_\omega \frac{T^{1/2}}{n^{1/2}}B\sqrt{T}}{T^3 L^{3/2}\Delta^{3/2}\tau\sqrt{n}}\right)^{1/3} + \left(\frac{L^3\Delta^3\sigma d\sigma_\omega^2 \frac{T}{n}B^2\sqrt{T}}{T^3 L^{3/2}\Delta^{3/2}\tau^2\sqrt{n}}\right)^{1/3} + \left(\frac{L^3\Delta^3\sigma d\sigma_\omega^2 \frac{T}{n}B^3\sqrt{T}}{T^3 L^2\Delta^2\tau^2\sqrt{n}}\right)^{1/3}$$

$$+ \left(\frac{L^3\Delta^3\sigma^2 d\sigma_\omega^2 \frac{T}{n}B^2\sqrt{T}}{T^3 L^2\Delta^2\tau^2\sqrt{n}}\right)^{1/3} + \left.\left(\frac{L^2\Delta^2 d^{1/2}\sigma_\omega \frac{T^{1/2}}{n^{1/2}}B^2\sigma\sqrt{T}}{T^2 L^{3/2}\Delta^{3/2}\sqrt{n}\tau}\right)^{1/2}\right)$$

$$= \widetilde{\mathcal{O}}\left(\left(\frac{L^2\Delta^2\sigma^2 B^2 d\sigma_\omega^2}{T^2 n\tau^2}\right)^{1/4} + \left(\frac{L^2\Delta^2\sigma d^{1/2}\sigma_\omega B}{nT^2\tau}\right)^{1/3} + \left(\frac{L^{3/2}\Delta^{3/2}\sigma d^{1/2}\sigma_\omega B^2}{nT^2\tau}\right)^{1/3}\right.$$

$$+ \left(\frac{L^{3/2}\Delta^{3/2}\sigma^2 d^{1/2}\sigma_\omega B}{nT^2\tau}\right)^{1/3} + \left(\frac{L^{3/2}\Delta^{3/2}\sigma d\sigma_\omega^2 B^2}{T^{3/2}n^{3/2}\tau^2}\right)^{1/3} + \left(\frac{L\Delta\sigma d\sigma_\omega^2 B^3}{n^{3/2}T^{3/2}\tau^2}\right)^{1/3}$$

$$+ \left.\left(\frac{L\Delta\sigma^2 d\sigma_\omega^2 B^2}{T^{3/2}n^{3/2}\tau^2}\right)^{1/3} + \left(\frac{L^{1/2}\Delta^{1/2}d^{1/2}\sigma_\omega B^2\sigma}{Tn\tau}\right)^{1/2}\right). \tag{52}$$

As we can see, the worst dependency on $T$ and $\sigma_\omega$ comes from terms $5 - 7$. Therefore, we omit the rest of the terms. Hence, the worst term w.r.t. $T$ in the presence of DP noise gives the rate

$$\widetilde{\mathcal{O}}\left(\left(\frac{L^{3/2}\Delta^{3/2}\sigma d\sigma_\omega^2 B^2}{T^{3/2}n^{3/2}\tau^2}\right)^{1/3} + \left(\frac{L\Delta\sigma d\sigma_\omega^2 B^3}{n^{3/2}T^{3/2}\tau^2}\right)^{1/3} + \left(\frac{L\Delta\sigma^2 d\sigma_\omega^2 B^2}{T^{3/2}n^{3/2}\tau^2}\right)^{1/3}\right)$$

$$= \widetilde{\mathcal{O}}\left(\frac{L^{1/2}\Delta^{1/2}\sigma^{1/3}d^{1/3}\sigma_\omega^{2/3}B^{2/3}}{T^{1/2}n^{1/2}\tau^{2/3}} + \frac{L^{1/3}\Delta^{1/3}\sigma^{1/3}d^{1/3}\sigma_\omega^{2/3}B}{n^{1/2}T^{1/2}\tau^{2/3}} + \frac{L^{1/3}\Delta^{1/3}\sigma^{2/3}d^{1/3}\sigma_\omega^{2/3}B^{2/3}}{T^{3/2}n^{3/2}\tau}\right)$$

$$= \widetilde{\mathcal{O}}\left(\frac{L^{1/3}\Delta^{1/3}\sigma^{1/3}d^{1/3}\sigma_\omega^{2/3}B^{2/3}}{T^{1/2}n^{1/2}\tau^{2/3}}\left((L\Delta)^{1/6} + B^{1/3} + \sigma^{1/3}\right)\right)$$

$$= \widetilde{\mathcal{O}}\left(\left(\frac{L\Delta\sigma d\sigma_\omega^2 B^2}{(nT)^{3/2}\tau^2}\left(\sqrt{L\Delta} + B + \sigma\right)\right)^{1/3}\right). \tag{53}$$

Besides, the momentum restrictions $\hat{\beta} \leq \frac{\sqrt{L\Delta}}{a}$ and $6L\gamma = \beta$ give us the following restrictions on the stepsize

$$\gamma \leq \frac{1}{L}\widetilde{\mathcal{O}}\left(\min\left\{\frac{\tau}{a}, \frac{\tau\sqrt{L\Delta}}{BaT}, \frac{\sqrt{L\Delta}\tau}{\sigma a}\right\}\right)$$

that translate to the following rate

$$\frac{L\Delta}{T}\widetilde{\mathcal{O}}\left(\frac{a}{\tau} + \frac{Ba}{\tau\sqrt{L\Delta}} + \frac{\sigma a}{\tau\sqrt{L\Delta}}\right)$$

$$= \widetilde{\mathcal{O}}\left(\frac{L\Delta}{T}\frac{d^{1/2}\sigma_\omega \frac{T^{1/2}}{n^{1/2}}}{\tau} + \frac{\sqrt{L\Delta}}{T}\frac{Bd^{1/2}\sigma_\omega \frac{T^{1/2}}{n^{1/2}}}{\tau} + \frac{L\Delta}{T}\frac{\sigma d^{1/2}\sigma_\omega \frac{T^{1/2}}{n^{1/2}}}{\tau\sqrt{L\Delta}}\right)$$

$$= \widetilde{\mathcal{O}}\left(\frac{\sqrt{L\Delta}d\sigma_\omega}{\tau\sqrt{nT}}\left(\sqrt{L\Delta} + B + \sigma\right)\right). \tag{54}$$

Besides, the momentum restrictions $\hat{\beta} \leq \sqrt{L\Delta}\left(\frac{4}{a^2\tau T}\right)^{1/3}$ and $6L\gamma = \beta$ give us the following restrictions on the stepsize

$$\gamma \leq \frac{1}{L}\widetilde{\mathcal{O}}\left(\min\left\{\frac{\tau^{2/3}}{a^{2/3}T^{1/3}}, \frac{\tau^{2/3}\sqrt{L\Delta}}{Ba^{2/3}T^{1/3}}, \frac{\sqrt{L\Delta}\tau^{2/3}}{\sigma a^{2/3}T^{1/3}}\right\}\right)$$

that translate to the following rate

$$\frac{L\Delta}{T}\widetilde{\mathcal{O}}\left(\frac{a^{2/3}T^{1/3}}{\tau^{2/3}} + \frac{Ba^{2/3}T^{1/3}}{\tau^{2/3}\sqrt{L\Delta}} + \frac{\sigma a^{2/3}T^{1/3}}{\tau^{2/3}\sqrt{L\Delta}}\right)$$

$$= \widetilde{\mathcal{O}}\left(\frac{L\Delta}{T^{2/3}}\frac{d^{1/3}\sigma_\omega^{2/3}\frac{T^{1/3}}{n^{1/3}}}{\tau^{2/3}} + \frac{\sqrt{L\Delta}}{T^{2/3}}\frac{Bd^{1/3}\sigma_\omega^{2/3}\frac{T^{1/3}}{n^{1/3}}}{\tau^{2/3}} + \frac{\sqrt{L\Delta}}{T^{2/3}}\frac{\sigma d^{1/3}\sigma_\omega^{2/3}\frac{T^{1/3}}{n^{1/3}}}{\tau^{2/3}\sqrt{L\Delta}}\right)$$

$$= \widetilde{\mathcal{O}}\left(\frac{L\Delta}{T^{1/3}}\frac{d^{1/3}\sigma_\omega^{2/3}}{\tau^{2/3}n^{1/3}} + \frac{\sqrt{L\Delta}}{T^{1/3}}\frac{Bd^{1/3}\sigma_\omega^{2/3}}{\tau^{2/3}n^{1/3}} + \frac{\sqrt{L\Delta}}{T^{1/3}}\frac{\sigma d^{1/3}\sigma_\omega^{2/3}}{\tau^{2/3}n^{1/3}}\right)$$

$$= \widetilde{\mathcal{O}}\left(\frac{\sqrt{L\Delta}d^{1/3}\sigma_\omega^{2/3}}{\tau^{2/3}(Tn)^{1/3}}\left(\sqrt{L\Delta} + B + \sigma\right)\right). \tag{55}$$

The restriction in (37) translates to

$$\gamma \leq \widetilde{\mathcal{O}}\left(\min\left\{\frac{\hat{\beta}\eta}{L}, \frac{\sqrt{\hat{\beta}\eta}}{L}\right\}\right),$$

that translates to the following rate of convergence

$$\frac{L\Delta}{T}\widetilde{\mathcal{O}}\left(\frac{Bd^{1/2}\sigma_\omega\frac{T^{1/2}}{n^{1/2}}}{\tau\sqrt{L\Delta}} + \frac{B^{1/2}d^{1/4}\sigma_\omega^{1/2}\frac{T^{1/4}}{n^{1/4}}}{\tau^{1/2}}\right)$$

$$= \widetilde{\mathcal{O}}\left(\frac{\sqrt{L\Delta}Bd^{1/2}\sigma_\omega}{\sqrt{T}n\tau} + \frac{L^{3/4}\Delta^{3/4}B^{1/2}d^{1/4}\sigma_\omega^{1/2}}{T^{3/4}n^{1/4}\tau^{1/2}}\right). \tag{56}$$

Combining (53), (54), (55), and (56), we derive the final bound

$$\widetilde{\mathcal{O}}\left(\left(\frac{L\Delta\sigma d\sigma_\omega^2 B^2}{(nT)^{3/2}\tau^2}\left(\sqrt{L\Delta} + B + \sigma\right)\right)^{1/3} + \frac{\sqrt{L\Delta}d\sigma_\omega}{\tau\sqrt{nT}}\left(\sqrt{L\Delta} + B + \sigma\right)\right. \tag{57}$$

$$\left. + \frac{\sqrt{L\Delta}d^{1/3}\sigma_\omega^{2/3}}{\tau^{2/3}(Tn)^{1/3}}\left(\sqrt{L\Delta} + B + \sigma\right)\right), \tag{58}$$

where we hide the terms that decrease faster in $T$ than the two in (57).

CASE $\mathcal{I}_{K+1} = 0$. This case is even easier. The only change will be with the term next to $R^t$. We will get

$$1 - \frac{96L^2}{\hat{\beta}^2\eta^2}\gamma^2 - \frac{24L^2}{\beta^2}\gamma^2 \geq \frac{1}{3} - \frac{96L^2}{\hat{\beta}^2\eta^2}\gamma^2 \geq 0$$

instead of

$$1 - \frac{32\beta^2L^2}{\hat{\beta}^2\eta^2}\gamma^2 - \frac{96L^2}{\hat{\beta}^2\eta^2}\gamma^2 - \frac{24L^2}{\beta^2}\gamma^2 \geq 0$$

as in the previous case. This difference comes from Lemma F.8 because $\widetilde{V}^{K+1} = 0$. The rest is a repetition of the previous derivations.

$\square$

# G PROOF OF COROLLARY 3.4 (PRIVACY ANALYSIS OF Clip21-SGD2M)

**Corollary 3.4.** *Let Assumptions 1.1 and 1.2 hold and $\alpha \in (0, 1)$. Let $\Delta \geq \Phi^0$ and $\sigma_\omega$ be chosen as $\sigma_\omega = \Theta\left(\frac{\tau}{\varepsilon}\sqrt{T\log\left(\frac{T}{\delta}\right)\log\left(\frac{1}{\delta}\right)}\right)$ for some $\varepsilon, \delta \in (0, 1)$. Then there exists a stepsize $\gamma$ and momentum parameters $\beta, \hat{\beta}$ such that the iterates of Clip21-SGD2M (Algorithm 3) with probability at least $1 - \alpha$ satisfy local $(\varepsilon, \delta)$-DP and*

$$\frac{1}{T}\sum_{t=0}^{T-1}\|\nabla f(x^t)\|^2 \leq \widetilde{\mathcal{O}}\left(\sqrt{L\Delta}\left(\frac{\sqrt{d}}{\sqrt{n}\varepsilon} + \left(\frac{\sqrt{d}}{\sqrt{n}\varepsilon}\right)^{2/3}\right)\left(\sqrt{L\Delta} + \widetilde{B} + \sigma\right)\right), \tag{12}$$

*where $\widetilde{\mathcal{O}}$ hides constant and polylogarithmic factors, and terms decreasing in $T$.*

*Proof.* We need to plug in the value of $\sigma_\omega$ inside (11). Indeed, we have that

$$\widetilde{\mathcal{O}}\left(\left(\frac{\sqrt{L\Delta d}\sqrt{T}\frac{\tau}{\varepsilon}}{\sqrt{nT}\tau} + \frac{\sqrt{L\Delta}d^{1/3}\frac{\tau^{2/3}}{\varepsilon^{2/3}}T^{1/3}}{\tau^{2/3}(Tn)^{1/3}}\right)(\sqrt{L\Delta} + B + \sigma)\right.$$

$$\left. + \left(\frac{L\Delta\sigma B^2\frac{\tau^2}{\varepsilon^2}T}{(nT)^{3/2}\tau^2}(\sqrt{L\Delta} + B + \sigma)\right)^{1/3}\right)$$

$$= \widetilde{\mathcal{O}}\left(\sqrt{L\Delta}\left(\frac{\sqrt{d}}{\sqrt{n}\varepsilon} + \left(\frac{\sqrt{d}}{\sqrt{n}\varepsilon}\right)^{2/3}\right)(\sqrt{L\Delta} + B + \sigma) + \left(\frac{L\Delta\sigma B^2}{n^{3/2}T^{1/2}\varepsilon^2}(\sqrt{L\Delta} + B + \sigma)\right)^{1/3}\right)$$

Leaving only the terms that do not improve with $T$ we get the result, i.e., the utility bound.

It remains to formally show that for chosen $\sigma_\omega$, `Clip21-SGD2M` satisfies local $(\varepsilon, \delta)$-DP. First, we notice that for $\sigma_\omega = \frac{8\tau}{\varepsilon}\sqrt{T\log\left(\frac{5T}{4\delta}\right)\log\left(\frac{1}{\delta}\right)}$ each step of `Clip21-SGD2M` satisfies $(\tilde{\varepsilon}, \tilde{\delta})$-DP (Dwork et al., 2014, Theorem 3.22) with

$$\tilde{\varepsilon} = \frac{\varepsilon}{2\sqrt{2T\log(\frac{1}{\delta})}} \quad \text{and} \quad \tilde{\delta} = \frac{\delta}{T}.$$

Then, applying advanced composition theorem (Dwork et al., 2014, Theorem 3.20 and Corollary 3.21 with $\delta' = \delta$), we get that $T$ steps of `Clip21-SGD2M` satisfy $(\varepsilon, \delta)$-DP, which concludes the proof. $\square$

# H  PROOF OF THEOREM 3.2 (CONVERGENCE OF `CLIP21-SGD2M` IN THE STOCHASTIC SETTING WITHOUT DP NOISE

We highlight that the proof of Theorem 3.2 mainly follows that of Theorem 3.3. The main difference comes from the fact that stepsize and momentum restrictions become less demanding as in a purely stochastic setting (without DP noise) $a = 0$. This, in particular, means that the restriction $\hat{\beta} \leq \frac{\sqrt{L\Delta}}{a}$ disappears and we can set $\hat{\beta} = 1$.

**Theorem H.1** (Full statement of Theorem 3.2). *Let Assumptions 1.1 and 1.2 hold,*

$$B := \max\{3\tau, \max_i\{\|\nabla f_i(x^0)\|\} + b\} > \tau,$$

*probability confidence level $\alpha \in (0, 1)$, constants $b$ and $c$ be defined as in (29), and $\Delta \geq \Phi^0$ for $\Phi^0$ defined in (9). Let us run Algorithm 3 for $T$ iterations with DP noise variance $\sigma_\omega = 0$. Assume the following inequalities hold*

1. **stepsize restrictions:**

       *i)* $12L\gamma \leq 1$;

       *ii)*

   $$\frac{1}{3} - \frac{32\beta^2 L^2}{\eta^2}\gamma^2 - \frac{96L^2}{\eta^2}\gamma^2 \geq 0;$$

2. **momentum restrictions:**

       *i)* $6L\gamma = \beta$;

       *ii)* $\beta \leq \frac{3\tau}{64\sqrt{L\Delta}}$;

       *iii)* $\beta \leq \frac{\tau}{14(B-\tau)}$;

       *iv)* $\beta \leq \frac{\tau}{22b}$;

       *v)* *and momentum restrictions defined in (40), (41), (42), (43), (44), (46), (45), and (47), where $\hat{\beta} = 1$.*

*Then with probability $1 - \alpha$ we have*

$$\frac{1}{T} \sum_{t=0}^{T-1} \|\nabla f(x^t)\|^2 \leq \widetilde{\mathcal{O}} \left( \frac{\sigma(\sqrt{L\Delta} + B + \sigma)}{\sqrt{Tn}} \right),$$

*where $\widetilde{\mathcal{O}}$ hides constant and polylogarithmic factors, and higher order terms decrease in $T$.*

*Proof.* The proof mainly follows that of Theorem 3.3. Since in this case, we can set $\hat{\beta} = 1$ and $a = 0$, the worst stepsize restrictions that we have in this case lead to the rate (51), which concludes the proof.

$\square$

## I  EXPERIMENTS: ADDITIONAL DETAILS AND RESULTS

### I.1  EXPERIMENTS WITH LOGISTIC REGRESSION

We evaluate our methods on non-convex logistic regression with regularization $\lambda = 10^{-3}$ over $10^4$ iterations—a setup standard in recent studies (Gao et al., 2024; Islamov et al., 2024b; Makarenko et al., 2022). Using the Duke and Leukemia datasets from LIBSVM (Chang & Lin, 2011), we split each into $n = 4$ equal shards and normalize each feature vector. To emulate stochastic gradients, we either add zero-mean Gaussian noise (variance $\sigma = 0.05$ for Duke, $\sigma = 0.1$ for Leukemia) or sample mini-batches of size $1/3$ of each local dataset for Duke and $1/4$ for Leukemia. For `Clip-SGD` and `Clip21-SGD`, we sweep the stepsize $\gamma \in \{2^{-5}, \dots, 2^5\}$ and select the value minimizing the final gradient norm (averaged over three random seeds). `Clip21-SGD2M` is tuned over the same $\gamma$ grid plus momentum $\beta \in \{0.1, 0.5, 0.9\}$, choosing the best $(\gamma, \beta)$ pair similarly. Figure I.1 shows the resulting convergence curves. We observe that `Clip21-SGD2M` remains stable across a wide range of clipping thresholds $\tau$, whereas `Clip-SGD` requires sufficiently large $\tau$ to converge, and `Clip21-SGD` often fails altogether—consistent with our theoretical non-convergence result in Theorem 2.2.

### I.2  EXPERIMENTS WITH NEURAL NETWORKS

The experiments of this section are conducted on a single Nvidia GTX 3090 GPU with 24 Gb RAM.

#### I.2.1  VARYING CLIPPING RADIUS $\tau$

We then turn to training ResNet-20 and VGG-16 on CIFAR-10, deliberately avoiding any learning-rate schedules, warm-up schemes, or weight-decay regularization across all methods. For `Clip-SGD` and `Clip21-SGD`, we sweep the stepsize $\gamma \in \{10^{-3}, \dots, 10^0\}$ and select the value that maximizes test accuracy. For `Clip21-SGD2M`, we search over the same $\gamma$ grid and momentum $\beta \in \{0.1, 0.5, 0.9\}$ (with $\hat{\beta} = 1$), picking the $(\gamma, \beta)$ pair that yields the highest test performance. All experiments use a batch size of 32, and we evaluate both global and layer-wise clipping.

Figure I.2 reports that `Clip21-SGD2M` enjoys more robustness to the choice of the clipping parameter $\tau$ when clipping is applied layer-wise. As shown in Figures I.5–I.4, `Clip-SGD`'s accuracy and loss deteriorate sharply once the clipping radius $\tau$ becomes small. In contrast, `Clip21-SGD2M` remains robust to the choice of $\tau$, consistently achieving lower training loss and higher test accuracy even under aggressive clipping.

#### I.2.2  RESULTS WITH ADDITIVE DP NOISE

We evaluate private training on MNIST using two architectures—a one-hidden-layer MLP (256 units, Tanh activation) and a CNN with two convolutional layers (16 filters, kernel size 5), one max-pooling layer, and Tanh activations—under privacy budgets $\varepsilon \in \{3, 5.2, 9, 15.6, 27\}$ (with $\delta = 10^{-3}$). For each $\varepsilon$, we conduct a thorough grid search over the stepsize $\gamma \in \{10^{-3}, \dots, 10^0\}$, clipping thresholds $\tau \in \{10^{-5}, 10^{-4}, 10^{-3}, \dots, 10^{-2}\}$ for `Clip21-SGD2M` and `Clip21-SGD` and $\tau \in \{10^{-4}, 10^{-3}, 10^{-2}, \dots, 10^0\}$ for `Clip-SGD`, and algorithm-specific parameters: $\alpha \in \{10^{-2}, \dots, 10^1\}$ for $\alpha$-`NormEC-SGD`, $\beta \in \{0.1, 0.5, 0.9\}$ for `Clip21-SGD2M` client momentum, and

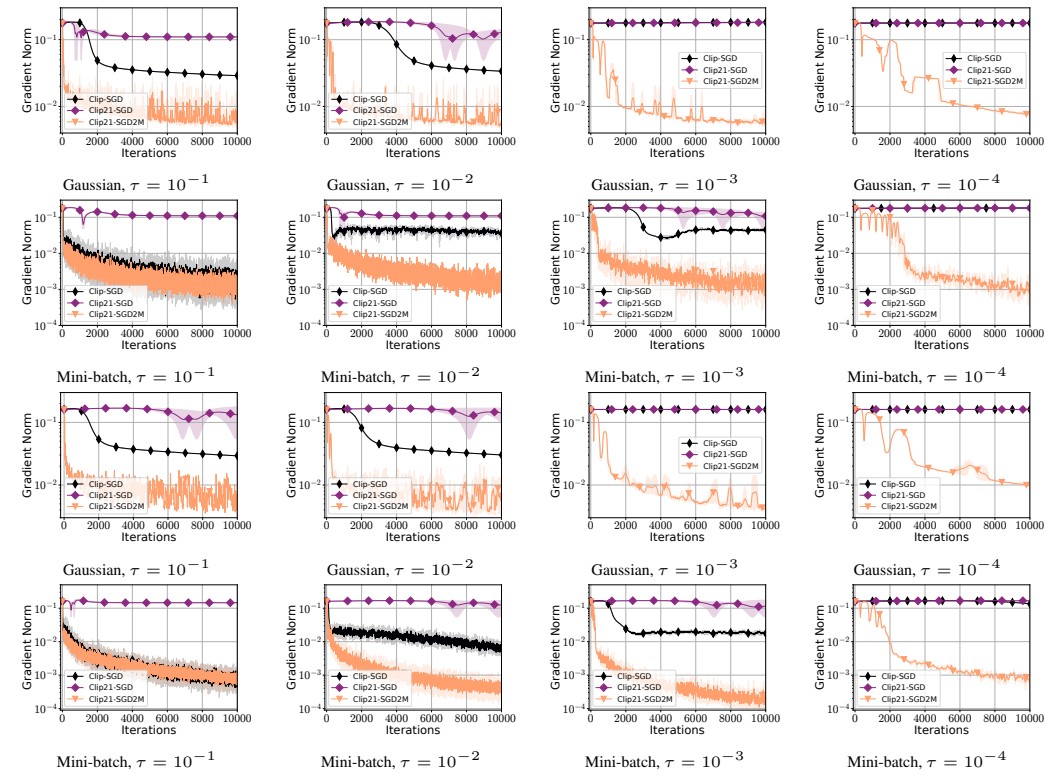

Figure I.1: Comparison of `Clip-SGD`, `Clip21-SGD`, and `Clip21-SGD2M` ($\hat{\beta} = 1$) on logistic regression with non-convex regularization for various the clipping radii $\tau$ with mini-batch and Gaussian-added stochastic gradients on Duke (**two first rows**) and Leukemia (**two last rows**).

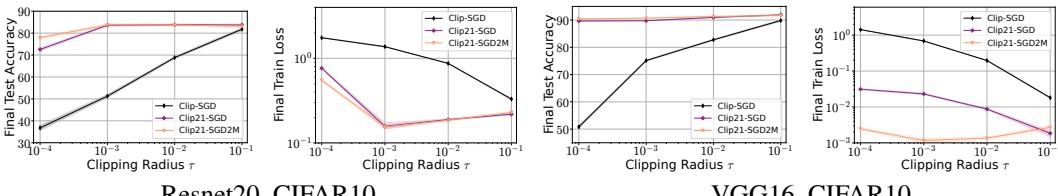

Figure I.2: Comparison of `Clip-SGD`, `Clip21-SGD`, and `Clip21-SGD2M` when training Resnet20 (**two left**) and VGG16 (**two right**) models on CIFAR10 dataset where the clipping is applied layer-wise. The training loss and test accuracy dynamics are presented in Figure I.4 and Figure I.6.

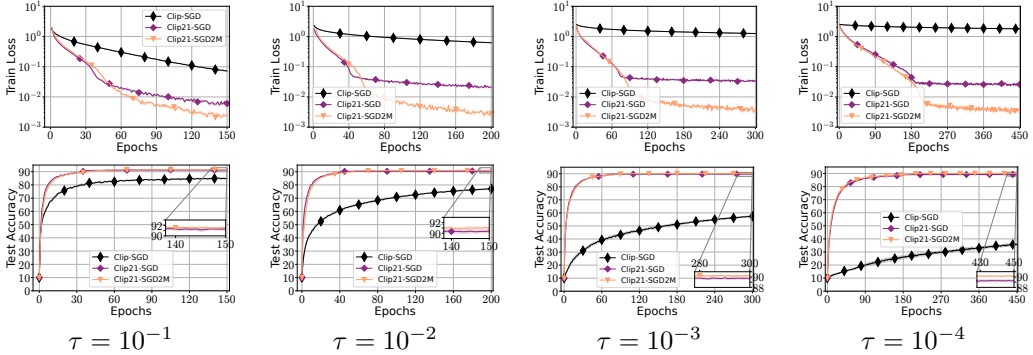

Figure I.3: Comparison of `Clip-SGD`, `Clip21-SGD`, and `Clip21-SGD2M` ($\hat{\beta} = 1$) on training VGG16 model on CIFAR10 dataset where the clipping is applied globally.

$\hat{\beta} \in \{0.01, 0.1, 0.5, 0.9\}$ for both `Clip21-SGD2M` and $\alpha$-`NormEC-SGD`. No learning-rate schedules or weight decay are used, and all methods train with batch size 64.

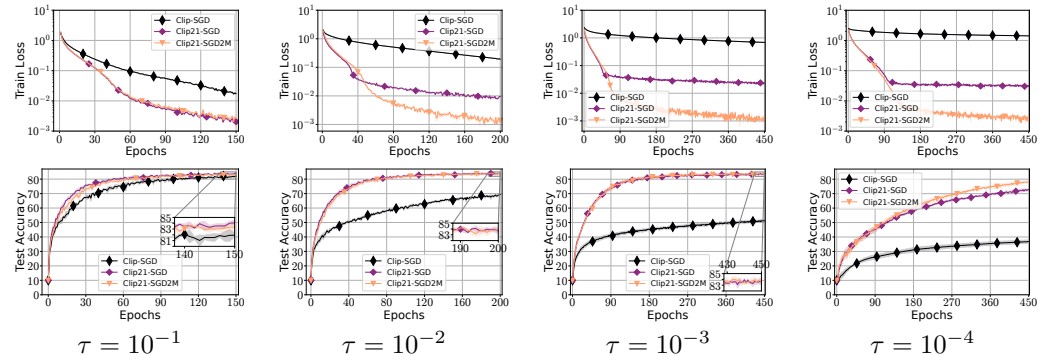

Figure I.4: Comparison of `Clip-SGD`, `Clip21-SGD`, and `Clip21-SGD2M` ($\hat{\beta} = 1$) on training VGG16 model on CIFAR10 dataset the clipping is applied layer-wise.

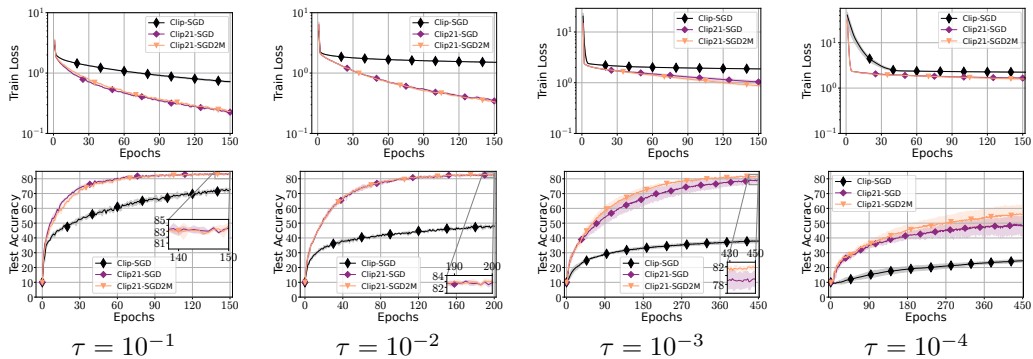

Figure I.5: Comparison of `Clip-SGD`, `Clip21-SGD`, and `Clip21-SGD2M` ($\hat{\beta} = 1$) on training Resnet20 model on CIFAR10 dataset where the clipping is applied globally.

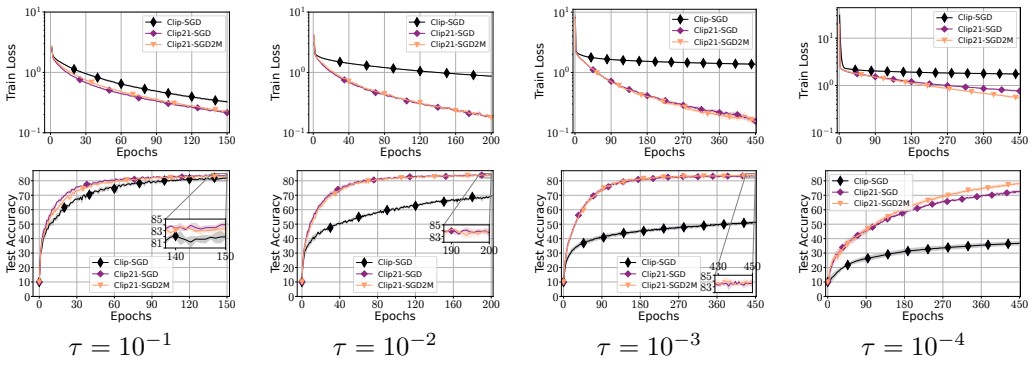

Figure I.6: Comparison of `Clip-SGD`, `Clip21-SGD`, and `Clip21-SGD2M` ($\hat{\beta} = 1$) on training Resnet20 model on CIFAR10 dataset where the clipping is applied layer-wise.

As shown in Figures I.7–I.10, both `Clip-SGD` and `Clip21-SGD2M` consistently surpass `Clip21-SGD` and $\alpha$-`NormEC-SGD` across privacy budgets. `Clip-SGD` achieves marginally higher accuracy on the CNN, while `Clip21-SGD2M` leads on the MLP. These results demonstrate that `Clip21-SGD2M` matches the state-of-the-art performance of `Clip-SGD` under differential privacy, but does so with stronger theoretical optimization guarantees and without assuming bounded data heterogeneity or gradient norms. Final test accuracy is reported in Table 1.

Table 1: Test accuracy when training MLP and CNN models with additive Gaussian noise for $(\varepsilon, \delta)$-DP. We vary the privacy budget $\varepsilon$ and fix $\delta = 10^{-3}$. These results demonstrate that `Clip21-SGD2M` achieves competitive performance to the state-of-the-art `Clip-SGD` method without relying on the bounded heterogeneity assumptions.

| Model | Dataset | Method | Hyperparameters | Final Test Accuracy | | | | |
|---|---|---|---|---|---|---|---|---|
| | | | | $\varepsilon = 3$ | $\varepsilon = 5.2$ | $\varepsilon = 9$ | $\varepsilon = 15.6$ | $\varepsilon = 27$ |
| MLP | MNIST | `Clip-SGD` `Clip21-SGD` $\alpha$-`NormEC` `Clip21-SGD2M` | batch size 64, # epochs 150, $n = 25$ | $59.5_{\pm2.6}$ $49.2_{\pm4.0}$ $9.0_{\pm2.0}$ $62.6_{\pm2.8}$ | $74.5_{\pm1.3}$ $68.1_{\pm1.9}$ $28.7_{\pm6.7}$ $75.9_{\pm0.9}$ | $79.5_{\pm0.4}$ $79.0_{\pm0.7}$ $42.2_{\pm5.6}$ $83.0_{\pm0.9}$ | $81.2_{\pm0.3}$ $77.9_{\pm0.6}$ $53.4_{\pm3.8}$ $87.7_{\pm0.6}$ | $88.5_{\pm0.1}$ $86.7_{\pm0.5}$ $64.1_{\pm3.5}$ $89.2_{\pm0.3}$ |
| CNN | MNIST | `Clip-SGD` `Clip21-SGD` $\alpha$-`NormEC` `Clip21-SGD2M` | batch size 64, # epochs 150, $n = 25$ | $58.9_{\pm2.4}$ $46.1_{\pm2.4}$ $10.4_{\pm2.4}$ $61.2_{\pm2.4}$ | $78.7_{\pm1.4}$ $67.9_{\pm1.4}$ $23.0_{\pm1.4}$ $76.0_{\pm1.4}$ | $82.8_{\pm1.6}$ $76.4_{\pm1.6}$ $56.4_{\pm1.6}$ $80.9_{\pm1.6}$ | $83.9_{\pm1.4}$ $79.3_{\pm1.4}$ $56.4_{\pm1.4}$ $87.6_{\pm1.4}$ | $91.0_{\pm0.4}$ $86.7_{\pm0.4}$ $57.1_{\pm0.4}$ $89.6_{\pm0.4}$ |

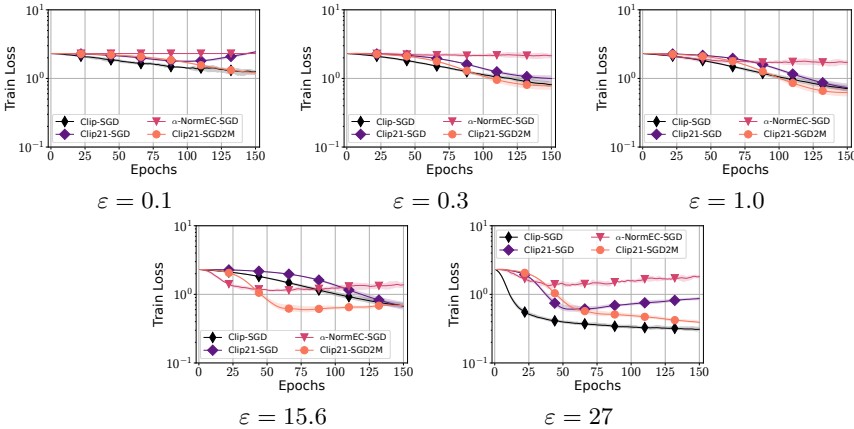

Figure I.7: Comparison of `Clip-SGD`, `Clip21-SGD`, and `Clip21-SGD2M` when training the CNN model on the MNIST dataset, varying the privacy budget $\varepsilon$.

Figure I.8: Comparison of `Clip-SGD`, `Clip21-SGD`, and `Clip21-SGD2M` when training the CNN model on the MNIST dataset, varying the privacy budget $\varepsilon$.

### I.3 LEARNING RATE TUNING FOR CNN

In this section, we provide the learning rate and clipping sweep details used in Figure 4 when training the CNN model on the MNIST dataset. We select the best hyperparameters based on a single run. Afterwards, we run the algorithms with the selected hyperparameters 3 times, which corresponds to the results in Figure 4.

The results are presented in Tables 2, 3, 4, 5, 6, 7. We observe that in most cases, the optimal learning rate lies strictly inside the tested range.

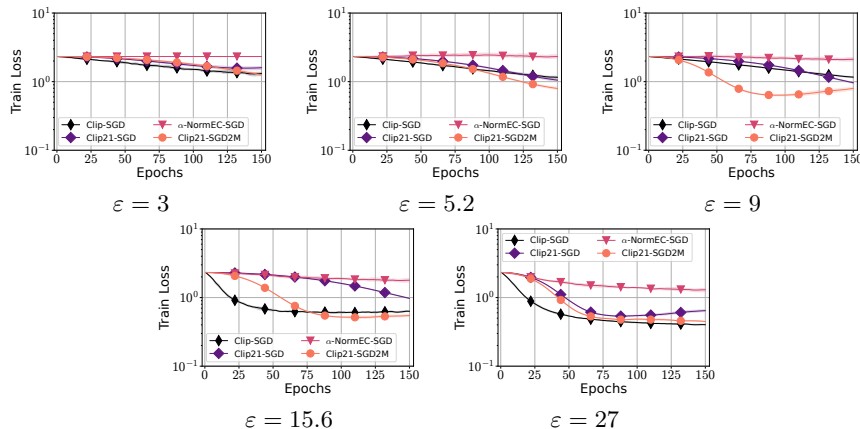

Figure I.9: Comparison of `Clip-SGD`, `Clip21-SGD`, $\alpha$-`NormEC`, and `Clip21-SGD2M` when training the MLP model on the MNIST dataset, varying the privacy budget $\varepsilon$.

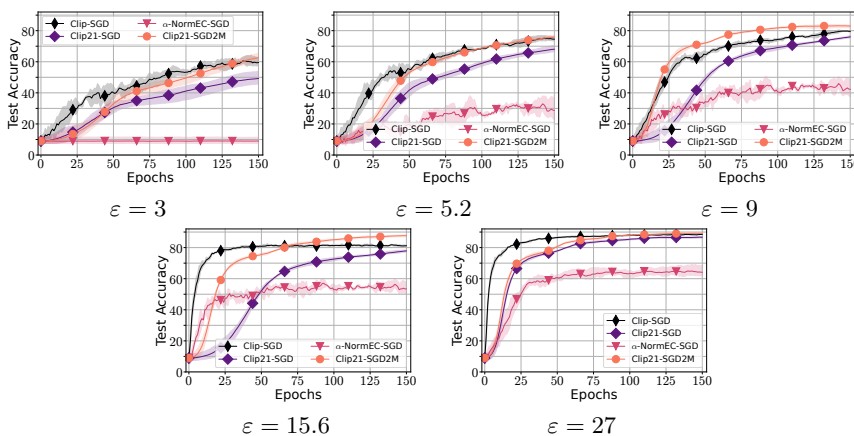

Figure I.10: Comparison of `Clip-SGD`, `Clip21-SGD`, $\alpha$-`NormEC`, and `Clip21-SGD2M` when training the MLP model on the MNIST dataset, varying the noise-clipping ratio.

Table 2: Performance (test accuracy) of `Clip21-SGD2M` when training the CNN model on the MNIST dataset, varying the clipping radius $\tau$ and learning rate.

| | | Learning rate | | | | | | | | | | | | | | | | | | | |
|---|---|---|---|---|---|---|---|---|---|---|---|---|---|---|---|---|---|---|---|---|---|
| | | $\varepsilon = 3$ | | | | $\varepsilon = 5.2$ | | | | $\varepsilon = 9$ | | | | $\varepsilon = 15.6$ | | | | $\varepsilon = 27$ | | | |
| | | 1e-3 | 1e-2 | 1e-1 | 1e0 | 1e-3 | 1e-2 | 1e-1 | 1e0 | 1e-3 | 1e-2 | 1e-1 | 1e0 | 1e-3 | 1e-2 | 1e-1 | 1e0 | 1e-3 | 1e-2 | 1e-1 | 1e0 |
| Clipping radius | 1e-5 | 18.4 | 38.0 | 63.4 | 29.8 | 16.2 | 21.0 | 56.8 | 78.7 | 16.2 | 20.7 | 63.6 | 82.9 | 18.0 | 47.6 | 78.6 | 88.0 | 18.0 | 47.9 | 79.0 | 89.2 |
| | 1e-4 | 37.9 | **63.5** | 29.8 | 13.4 | 21.0 | 56.8 | 78.8 | 53.0 | 20.7 | 63.5 | 82.9 | 75.7 | 47.4 | 78.9 | **87.9** | 75.8 | 47.8 | 79.4 | 89.3 | 84.8 |
| | 1e-3 | 58.4 | 27.6 | 13.2 | 7.3 | 56.6 | **78.9** | 52.33 | 28.0 | 63.3 | **83.2** | 74.8 | 49.8 | 81.6 | 85.5 | 73.0 | 45.6 | 82.1 | **89.6** | 83.1 | 70.0 |
| | 1e-2 | 22.4 | 14.5 | 6.2 | 5.3 | 75.0 | 44.6 | 25.8 | 8.1 | 82.8 | 66.6 | 46.3 | 16.7 | 69.9 | 58.6 | 36.6 | 14.4 | 81.1 | 68.4 | 55 | 26.5 |

## I.4 LEARNING RATE TUNING FOR MLP

In this section, we provide the learning rate and clipping sweep details used in Figure 4 when training the MLP model on the MNIST dataset. We select the best hyperparameters based on a single run. Afterwards, we run the algorithms with the selected hyperparameters 3 times, which corresponds to the results in Figure 4. We refer to Tables 2 to 7 for the results of the sweeps. We observe that in most cases, the optimal learning rate lies strictly inside the tested range.

Table 3: Performance (test accuracy) of `Clip21-SGD` when training the CNN model on the MNIST dataset, varying the clipping radius $\tau$ and learning rate.

| | | Learning rate | | | | | | | | | | | | | | | | | | | |
| | | $\varepsilon = 3$ | | | | $\varepsilon = 5.2$ | | | | $\varepsilon = 9$ | | | | $\varepsilon = 15.6$ | | | | $\varepsilon = 27$ | | | |
| | | 1e-3 | 1e-2 | 1e-1 | 1e0 | 1e-3 | 1e-2 | 1e-1 | 1e0 | 1e-3 | 1e-2 | 1e-1 | 1e0 | 1e-3 | 1e-2 | 1e-1 | 1e0 | 1e-3 | 1e-2 | 1e-1 | 1e0 |
|---|---|---|---|---|---|---|---|---|---|---|---|---|---|---|---|---|---|---|---|---|---|
| Clipping radius | 1e-5 | 19.6 | 34.5 | **46.1** | 16.9 | 16.4 | 45.1 | **71.6** | 30.7 | 19.3 | 53.8 | **79.4** | 60.6 | 19.3 | 57.7 | 81.8 | 79.0 | 19.2 | 59.2 | 82.8 | **87.0** |
| | 1e-4 | 33.2 | 45.3 | 16.9 | 8.2 | 43.5 | 71.2 | 30.4 | 9.0 | 52.4 | 79.2 | 60.0 | 23.9 | 56.1 | **81.9** | 78.6 | 44.9 | 57.6 | 82.9 | 86.8 | 68.3 |
| | 1e-3 | 32.2 | 15.9 | 7.8 | 7.0 | 61.5 | 29.4 | 10.6 | 7.4 | 74.2 | 52.4 | 21.6 | 7.7 | 79.5 | 71.3 | 41.5 | 14.5 | 80.8 | 83.4 | 65.1 | 24.4 |
| | 1e-2 | 12.1 | 8.1 | 7.0 | 6.6 | 20.5 | 7.1 | 6.8 | 6.7 | 31.7 | 17.1 | 7.6 | 7.3 | 48.8 | 31.8 | 14.1 | 5.6 | 63.8 | 49.3 | 26.0 | 7.0 |
| | 1e-1 | 7.0 | 6.9 | 6.6 | 6.8 | 9.7 | 7.4 | 6.5 | 6.9 | 11.1 | 6.6 | 7.2 | 7.2 | 13.0 | 8.0 | 5.9 | 7.1 | 20.2 | 17.0 | 6.8 | 5.5 |
| | 1e0 | 6.9 | 7.1 | 6.7 | 6.6 | 6.5 | 6.6 | 6.7 | 6.6 | 6.9 | 6.7 | 6.5 | 6.7 | 8.5 | 6.9 | 6.6 | 6.7 | 9.4 | 7.9 | 7.3 | 7.1 |

Table 4: Performance (test accuracy) of `Clip-SGD` when training the CNN model on the MNIST dataset, varying the clipping radius $\tau$ and learning rate.

| | | Learning rate | | | | | | | | | | | | | | | | | | | |
| | | $\varepsilon = 3$ | | | | $\varepsilon = 5.2$ | | | | $\varepsilon = 9$ | | | | $\varepsilon = 15.6$ | | | | $\varepsilon = 27$ | | | |
| | | 1e-3 | 1e-2 | 1e-1 | 1e0 | 1e-3 | 1e-2 | 1e-1 | 1e0 | 1e-3 | 1e-2 | 1e-1 | 1e0 | 1e-3 | 1e-2 | 1e-1 | 1e0 | 1e-3 | 1e-2 | 1e-1 | 1e0 |
|---|---|---|---|---|---|---|---|---|---|---|---|---|---|---|---|---|---|---|---|---|---|
| Clipping radius | 1e-5 | 15.8 | 15.8 | 16.0 | 18.4 | 15.8 | 15.8 | 16.0 | 18.4 | 15.8 | 15.8 | 15.9 | 18.2 | 15.8 | 15.8 | 15.9 | 18.1 | 15.8 | 15.8 | 15.9 | 18.0 |
| | 1e-4 | 15.8 | 16.0 | 18.4 | 37.1 | 15.8 | 16.0 | 18.4 | 42.9 | 15.8 | 15.9 | 18.2 | 46.4 | 15.8 | 15.9 | 18.1 | 47.6 | 15.8 | 15.9 | 18.0 | 47.9 |
| | 1e-3 | 16.0 | 18.4 | 37.1 | **57.4** | 16.0 | 18.4 | 42.9 | **79.9** | 15.9 | 18.2 | 46.4 | **84.3** | 15.9 | 18.1 | 47.6 | **85.2** | 15.9 | 18.0 | 47.9 | 85.5 |
| | 1e-2 | 18.4 | 37.1 | **57.4** | 13.5 | 18.4 | 42.9 | **79.9** | 9.2 | 18.2 | 46.4 | **84.3** | 59.3 | 18.1 | 47.6 | **85.2** | 82.0 | 18.0 | 47.9 | 85.5 | **91.4** |
| | 1e-1 | 37.1 | **57.4** | 13.5 | 7.8 | 42.9 | **79.9** | 9.2 | 15.7 | 46.7 | **84.3** | 59.3 | 17.7 | 47.6 | **85.2** | 82.0 | 10.6 | 47.9 | 85.5 | **91.4** | 62.0 |
| | 1e0 | **57.4** | 13.5 | 7.6 | 6.1 | **79.9** | 9.2 | 15.6 | 6.4 | **84.3** | 59.3 | 17.5 | 7.7 | **85.2** | 82.1 | 10.6 | 14.1 | 85.4 | **91.4** | 68.2 | 11.0 |

Table 5: Performance (test accuracy) of `Clip21-SGD2M` when training the MLP model on the MNIST dataset, varying the clipping radius $\tau$ and learning rate.

| | | Learning rate | | | | | | | | | | | | | | | | | | | |
| | | $\varepsilon = 3$ | | | | $\varepsilon = 5.2$ | | | | $\varepsilon = 9$ | | | | $\varepsilon = 15.6$ | | | | $\varepsilon = 27$ | | | |
| | | 1e-3 | 1e-2 | 1e-1 | 1e0 | 1e-3 | 1e-2 | 1e-1 | 1e0 | 1e-3 | 1e-2 | 1e-1 | 1e0 | 1e-3 | 1e-2 | 1e-1 | 1e0 | 1e-3 | 1e-2 | 1e-1 | 1e0 |
|---|---|---|---|---|---|---|---|---|---|---|---|---|---|---|---|---|---|---|---|---|---|
| Clipping radius | 1e-5 | 14.0 | 39.6 | **65.4** | 53.4 | 11.9 | 16.5 | 58.9 | 76.8 | 13.7 | 41.0 | 74.5 | 83.4 | 13.7 | 41.3 | 75.5 | 87.7 | 15.4 | 59.9 | 80.8 | 89.5 |
| | 1e-4 | 38.1 | 64.7 | 52.9 | 38.1 | 16.5 | 59.0 | **76.8** | 66.3 | 40.8 | 74.8 | **83.9** | 68.4 | 41.3 | 75.8 | **87.8** | 76.2 | 60.0 | 81.4 | **89.6** | 80.4 |
| | 1e-3 | 34.5 | 39.5 | 32.9 | 23.4 | 56.9 | 76.6 | 64.8 | 49.3 | 72.9 | 76.5 | 63.7 | 49.7 | 75.5 | 84.9 | 72.6 | 64.0 | 77.8 | 85.3 | 75.3 | 68.6 |
| | 1e-2 | 14.7 | 14.6 | 14.1 | 13.1 | 56.9 | 50.8 | 41.1 | 29.9 | 45.7 | 40.8 | 35.3 | 27.2 | 61.6 | 50.8 | 46.7 | 38.9 | 60.9 | 50.9 | 48.4 | 43.8 |

Table 6: Performance (test accuracy) of `Clip21-SGD` when training the MLP model on the MNIST dataset, varying the clipping radius $\tau$ and learning rate.

| | | Learning rate | | | | | | | | | | | | | | | | | | | |
| | | $\varepsilon = 3$ | | | | $\varepsilon = 5.2$ | | | | $\varepsilon = 9$ | | | | $\varepsilon = 15.6$ | | | | $\varepsilon = 27$ | | | |
| | | 1e-3 | 1e-2 | 1e-1 | 1e0 | 1e-3 | 1e-2 | 1e-1 | 1e0 | 1e-3 | 1e-2 | 1e-1 | 1e0 | 1e-3 | 1e-2 | 1e-1 | 1e0 | 1e-3 | 1e-2 | 1e-1 | 1e0 |
|---|---|---|---|---|---|---|---|---|---|---|---|---|---|---|---|---|---|---|---|---|---|
| Clipping radius | 1e-5 | 15.7 | 42.7 | **53.7** | 32.1 | 15.1 | 46.0 | **70.4** | 49.5 | 14.8 | 50.8 | **76.8** | 67.1 | 14.7 | 52.8 | **78.1** | 81.0 | 14.6 | 53.6 | 78.3 | **87.1** |
| | 1e-4 | 40.1 | 51.7 | 31.5 | 18.7 | 45.0 | 69.4 | 48.6 | 29.0 | 48.7 | 76.4 | 66.0 | 43.2 | 51.3 | 78.0 | 80.3 | 58.2 | 52.3 | 78.2 | 86.8 | 69.8 |
| | 1e-3 | 26.0 | 24.8 | 17.4 | 12.3 | 48.5 | 40.6 | 27.2 | 16.9 | 66.7 | 57.5 | 39.8 | 25.2 | 72.2 | 72.2 | 53.9 | 137.1 | 74.4 | 82.5 | 65.4 | 51.9 |
| | 1e-2 | 12.6 | 12.1 | 11.6 | 10.8 | 18.1 | 16.2 | 13.8 | 12.2 | 30.0 | 25.0 | 19.4 | 14.5 | 42.7 | 35.9 | 28.4 | 19.7 | 57.4 | 46.5 | 39.6 | 28.4 |
| | 1e-1 | 9.9 | 9.8 | 9.7 | 9.7 | 10.7 | 10.7 | 10.4 | 10.1 | 12.1 | 12.0 | 11.5 | 10.8 | 14.1 | 13.7 | 12.9 | 12.0 | 17.7 | 17.2 | 15.9 | 13.7 |
| | 1e0 | 9.4 | 9.4 | 9.5 | 9.4 | 9.7 | 9.7 | 9.7 | 9.6 | 9.9 | 9.9 | 9.8 | 9.8 | 10.3 | 10.2 | 10.0 | 10.0 | 10.7 | 10.5 | 10.1 | 10.1 |

# J DISCUSSION ON PRIVACY AMPLIFICATION BY SUBSAMPLING

We acknowledge that enabling amplification through data subsampling is an important aspect of algorithm design. However, example-wise clipping – required to incorporate such a modification – necessitates a substantially more involved theoretical analysis and more advanced proof techniques. Moreover, it remains an open question whether `Clip-SGD` can provably achieve privacy amplification through subsampling under standard assumptions. We therefore leave this direction to future work.

Table 7: Performance (test accuracy) of `Clip-SGD` when training the MLP model on the MNIST dataset, varying the clipping radius $\tau$ and learning rate.

| | | Learning rate | | | | | | | | | | | | | | | | | | | |
| | | $\varepsilon = 3$ | | | | $\varepsilon = 5.2$ | | | | $\varepsilon = 9$ | | | | $\varepsilon = 15.6$ | | | | $\varepsilon = 27$ | | | |
| | | 1e-3 | 1e-2 | 1e-1 | 1e0 | 1e-3 | 1e-2 | 1e-1 | 1e0 | 1e-3 | 1e-2 | 1e-1 | 1e0 | 1e-3 | 1e-2 | 1e-1 | 1e0 | 1e-3 | 1e-2 | 1e-1 | 1e0 |
|---|---|---|---|---|---|---|---|---|---|---|---|---|---|---|---|---|---|---|---|---|---|
| Clipping radius | 1e-5 | 15.8 | 15.8 | 16.0 | 18.4 | 15.8 | 15.8 | 16.0 | 18.4 | 15.8 | 15.8 | 15.9 | 18.2 | 15.8 | 15.8 | 15.9 | 18.1 | 15.8 | 15.8 | 15.9 | 18.0 |
| | 1e-4 | 15.8 | 16.0 | 18.4 | 37.1 | 15.8 | 16.0 | 18.4 | 42.9 | 15.8 | 15.9 | 18.2 | 46.4 | 15.8 | 15.9 | 18.1 | 47.6 | 15.8 | 15.9 | 18.0 | 47.9 |
| | 1e-3 | 16.0 | 18.4 | 37.1 | **57.4** | 16.0 | 18.4 | 42.9 | **79.9** | 15.9 | 18.2 | 46.4 | **84.3** | 15.9 | 12.1 | 47.6 | **85.2** | 15.9 | 18.0 | 47.9 | 85.5 |
| | 1e-2 | 18.4 | 37.1 | **57.4** | 13.5 | 18.4 | 42.9 | **79.9** | 9.2 | 18.2 | 46.4 | **84.3** | 59.3 | 18.1 | 47.6 | **85.2** | 82.0 | 18.0 | 47.9 | 85.5 | **91.4** |
| | 1e-1 | 37.1 | **57.4** | 13.5 | 7.8 | 42.9 | **79.9** | 9.2 | 15.7 | 46.4 | **84.3** | 59.3 | 17.7 | 47.6 | **85.2** | 82.0 | 10.6 | 47.9 | 85.5 | **91.4** | 62.0 |
| | 1e0 | **57.3** | 13.5 | 7.6 | 6.1 | **79.9** | 9.2 | 15.6 | 6.4 | **84.3** | 59.3 | 17.5 | 7.7 | **85.2** | 82.0 | 10.6 | 14.1 | 85.4 | **91.4** | 68.2 | 11.0 |

Nonetheless, we study this question in practice. In this setting, we assume that local functions $f_i$ have a finite-sum structure, namely, $f_i(x) := \frac{1}{m} \sum_{j=1}^{m} f_{ij}(x)$. To enable privacy amplification by data subsampling, each client $i \in [n]$ at iteration $t$ samples a batch $\mathcal{S}_i^t$ of size $b$, and each example-wise gradient is clipped. In this case, DP-noise variance can be significantly reduced by a factor $\frac{b}{m}$, which allows for achieving better practical performance. We call a modification of `Clip21-SGD2M` with example-wise clipping as `Clip21-SGD2M+` for clarity.

## J.1 ON THE THEORETICAL ANALYSIS OF CLIP21-SGD2M+

The key difficulty in the theoretical convergence analysis of `Clip21-SGD2M+` comes from per-sample gradient clipping (see Line 7 in Algorithm 5), which introduces bias in the local momentum vector $v_i^{t+1}$. Therefore, for an arbitrary clipping level $\tau_{\text{in}}$, we expect that the method will provably converge to some irreducible neighborhood even when $\sigma_\omega = 0$, similarly to the case of `Clip-SGD` (Koloskova et al., 2023). One may address this issue by taking $\tau_{\text{in}}$ sufficiently large such that the introduced bias is controlled, similarly to the analysis of `DProx-clipped-SGD-shift` in the convex case (Gorbunov et al., 2024, Theorem 2.5). The clipping level in this case will depend on some notion of gradient heterogeneity at some reference point. Nevertheless, for large enough $\tau_{\text{in}}$ our analysis of `Clip21-SGD2M` will require just minor modifications to be extended to `Clip21-SGD2M+`. The main idea behind this analysis is to show that $\|\nabla f_{ij}(x^{t+1})\|$ is bounded with high probability throughout the trajectory of the method. More precisely, taking $\tau_{\text{in}} \sim \max_{ij} \|\nabla f_{ij}(x^0)\| + CLR$ with $R = \sup\{\|x^0 - x^*\| \mid \nabla f(x^*) = 0\}$ and showing by induction that $\|x^0 - x^t\| \leq CR$ for some $C > 0$ with high probability, one can prove that $\|\nabla f_{ij}(x^{t+1})\| \leq \|\nabla f_{ij}(x^0)\| + \|\nabla f_{ij}(x^{t+1}) - \nabla f_{ij}(x^0)\| \leq \max_{ij} \|\nabla f_{ij}(x^0)\| + CLR = \tau_{\text{in}}$. That is, the inner clipping in this case is turned off with high probability, and the proof should closely follow the current analysis of `Clip21-SGD2M`, where only one clipping is used. Such an analysis still avoids using unrealistic assumptions like bounded gradients.

We leave the formal theoretical convergence analysis of `Clip21-SGD2M+` for future work.

## J.2 EMPIRICAL PERFORMANCE OF CLIP21-SGD2M+

Now we test the performance of `Clip21-SGD2M+` when training the same CNN and MLP models on the MNIST dataset. In this setting, we rescale the DP-noise variance $\sigma_\omega$ by a factor $\frac{b}{m}$. We test the performance of `Clip21-SGD2M+` against `Clip-SGD`, where we similarly use example-wise clipping to enable privacy amplification by data subsampling. Since `Clip21-SGD2M+` has two clipping parameters, we fix $\tau_{\text{in}} = 0.1$ and tune $\tau_{\text{out}}$. In this experiment, we tune the learning rate $\gamma \in \{10^{-2}, 10^{-1}, 10^0, 10^1\}$, clipping radius in $\{0.01, 0.03, 0.1, 0.3, 1\}$, while fixing $\beta = 0.1$, $\hat{\beta} = 0.01$. For both algorithms, we use the batch size 32, while the data partitioning is the same as before.

We present the results in fig. J.1. We observe that `Clip21-SGD2M+` achieves competitive performance to `Clip-SGD`, even in the setting when privacy amplification by data subsampling is used.

---

**Algorithm 5** `Clip21-SGD2M+`

---

**Require:** $x^0, g^0, v^0 \in \mathbb{R}^d$ (by default $g^0 = v^0 = 0$), momentum parameters $\beta, \hat{\beta} \in (0, 1]$, stepsize $\gamma > 0$, clipping parameters $\tau_{\text{in}}, \tau_{\text{out}} > 0$, batch size $b$, DP-variance parameter $\sigma_\omega^2 \geq 0$

1: Set $g_i^0 = g^0$ and $v_i^0 = v^0$ for all $i \in [n]$
2: **for** $t = 0, \ldots, T - 1$ **do**
3:    $x^{t+1} = x^t - \gamma g^t$
4:    **for** $i = 1, \ldots, n$ **do**
5:       Sample DP-noise $\omega_i^{t+1} \sim \mathcal{N}(0, \sigma_\omega^2 \mathbf{I})$                 only for DP version
6:       Sample batch $\mathcal{S}_i^t$
7:       $v_i^{t+1} = (1 - \beta)v_i^t + \beta \left( \frac{1}{b} \sum_{j \in \mathcal{S}_i^t} \text{clip}_{\tau_{\text{in}}}(\nabla f_{ij}(x^{t+1})) + \omega_i^{t+1} \right)$
8:       $c_i^{t+1} = \text{clip}_{\tau_{\text{out}}}(v_i^{t+1} - g_i^t)$
9:       $g_i^{t+1} = g_i^t + \hat{\beta}\,\text{clip}_{\tau_{\text{out}}}(v_i^{t+1} - g_i^t)$
10:   **end for**
11:   $g^{t+1} = g^t + \frac{\hat{\beta}}{n} \sum_{i=1}^n c_i^{t+1}$
12: **end for**

---

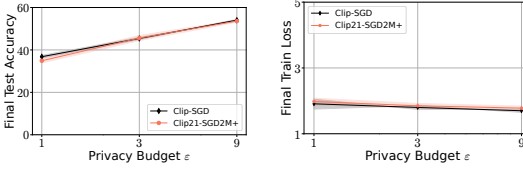

CNN, MNIST

Figure J.1: Comparison of `Clip-SGD` and `Clip21-SGD2M+` when training CNN on CIFAR10 dataset.

