# OpenReview forum: "Double Momentum and Error Feedback for Clipping with Fast Rates and Differential Privacy"
_ICLR.cc/2026/Conference — Submitted to ICLR 2026_

### Official Review · Reviewer_xbHq · 2025-10-21

**Soundness:** 3
**Presentation:** 3
**Contribution:** 3
**Rating:** 4
**Confidence:** 3

**Summary:**

The paper proposes Clip21-SGD2M as a federated learning (FL) algorithm that simultaneously delivers strong differential privacy (DP) guarantees and optimal optimization rates. The method integrates heavy-ball momentum with gradient-clipping updates, corrected via error feedback (EF) to mitigate clipping bias. Although the algorithmic novelty is limited, the paper offers several theoretical contributions. Without assuming bounded gradients or bounded client heterogeneity, it proves: (i) an optimal $O(1/T)$ rate for the squared gradient norm under non-convex loss functions with full gradients; (ii) an optimal $O(1/\sqrt{nT})$ rate under stochastic (sub-Gaussian) gradients; and (iii) under local DP and certain conditions, a privacy–utility trade-off that matches the best known non-convex utility bound. The effectiveness of proposed method is investigated through several numerical experiments.

**Strengths:**

**S1: Clarifying the limitations of related works via counterexamples (Sec. 2)**

Using example and theorem, the paper shows that existing methods (Clip-SGD, Clip21-SGD) may fail to converge.




**S2: Sufficient theoretical contributions (Sec. 3)**

S2a: Weak assumptions. The analysis does not assume bounded gradients or bounded data heterogeneity, restricting assumptions to $L$-smoothness, lower boundedness, and sub-Gaussianity regarding stochastic gradients.

S2b: Full-gradient case. Achieves the optimal rate $O(1/T)$ for the squared gradient norm.

S2c: Stochastic-gradient case. Achieves the optimal rate $O(1/\sqrt{nT})$.

S2d: With local DP. Presents a privacy–utility trade-off for locally client updates wity DP, and shows consistency with the best-known non-convex utility bound when the model dimension $d$ is sufficiently large.

**Weaknesses:**

**W1: Concerns about the tightness of the convergence rates (Sec. 3)**

In FL, data heterogeneity is empirically known to significantly affect convergence rates. While the paper gains generality by not assuming bounded data heterogeneity, the reported rates do not depend on the data heterogeneity parameter, which raises the possibility that the rates may NOT be tight. Discussion appears limited regarding whether (and how) data heterogeneity parameters could or should appear in the bounds.


**W2: Limited experimental coverage (Sec. 4)**

While I understand the paper’s primary contribution is theoretical, there are several critical experimental gaps:


**W2a (Lack of heterogeneity evaluation).**
The paper does not report performance changes under controlled data heterogeneity, which is standard in FL (e.g., partitioning data to clients using a Dirichlet concentration parameter to artificially control heterogeneity).


**W2b (Insufficient SOTA comparisons).**
Comparisons to state-of-the-art methods seem incomplete. In the non-DP setting, could you compare with EF21-SGDM to isolate the pure effect of clipping? In the DP setting, beyond the three baselines already included, could you add DP-SCAFFOLD, PORTER, and DIFF2?

**W2c (Limited model cale).**
It appears there are no evaluations with larger models. In the DP experiments, could you test beyond CNN-MNIST, for example, CIFAR-10/100 classification with ResNet-20/VGG16?

**W2d (Result consistency).**
In several settings, the proposed method does NOT appear consistently superior.

**Questions:**

Q1: In the DP experiments with multiple $\epsilon$ values, how did you compute the Gaussian noise size?

Q2: Could you report the learning-rate tuning results? (e.g., a plot with learning rates on the x-axis and test accuracy on the y-axis, to make clear that the chosen learning rate is not at endpoints.)

---

> ### Author Response · Authors · 2025-11-20
> **Rebuttals**
>
> We sincerely thank the reviewer for the constructive and encouraging feedback. In particular, we greatly appreciate the acknowledgement of our work’s motivation and presentation, as well as the emphasis on the limitations of existing algorithms and the strength of our theoretical analysis.
>
> We hope that the following clarifications will help re-evaluate our submission and potentially adjust the score.
>
> **W1: Concerns about the tightness of the convergence rates**
>
> ***Answer:*** We thank the reviewer for this comment. Let us clarify our rates in more detail. We claim that the analysis does not require any boundedness assumption on data heterogeneity (e.g., bounded gradients), and that Clip21-SGD2M is provably valid in arbitrarily heterogeneous environments. However, as the reviewer correctly noted, the convergence rate can deteriorate as data heterogeneity increases. In particular, the rate of Clip21-SGD2M depends on the term $B := \max_i \\|\nabla f_i(x^0)\\|$. Greater heterogeneity leads to a larger value of $B$, which in turn results in a slower convergence rate.
>
> Furthermore, as we explain in lines 314-315 and 339-340, the rates derived in Theorems 3.1 and 3.2 are indeed tight since they match the existing lower bounds for non-DP settings.
>
> **W2: Experimental gaps.**
>
> ***Answer:***
>
> - **a.** We would like to clarify our experimental setup. To simulate data heterogeneity, we partition the dataset as follows. Each client $i\in[n]$ is first assigned 1000 samples from a single class. We then shuffle the remaining data and distribute it randomly across all clients. This procedure creates a heterogeneous data distribution among clients.
>
> - **b.** We refer the reviewer to Figure 3. In the non-private setting, smaller clipping thresholds $\tau$ lead to worse empirical performance for all tested algorithms. This degradation for Clip21-SGD2M is expected, as its convergence rate depends on the ratio $B/\tau$, which increases when $\tau$ decreases. Nevertheless, Clip21-SGD2M performs slightly better than Clip-SGD for small values of $\tau$, and it significantly outperforms Clip-SGD in the non-private regime.
>
> - **c.** The convergence guarantees for all of the mentioned algorithms rely on the bounded-gradient assumption, which our analysis removes. Our choice of baselines is guided by the desire to compare against methods whose theoretical guarantees ***do not depend on such strong and unrealistic assumptions***. For this reason, we omit comparisons with those algorithms. Instead, we compare against Clip-SGD, which achieves state-of-the-art performance in practice, but still requires a bounded gradient assumption in theory.
>
>
> - **d.** The goal of our experimental section is to validate our theoretical claims and to demonstrate the practical competitiveness of Clip21-SGD2M. We do not aim to achieve state-of-the-art performance, and a broad, larger experimental study is beyond the scope of this work.
>
> **Q1: Gaussian noise size in experiments.**
>
> ***Answer:*** We choose the DP variance $\sigma_{\omega}$ to ensoure $(\varepsilon,\delta)$-privacy. To do so, we choose $\sigma_{\omega} = \frac{\tau\sqrt{T\frac{\text{local dataset size}}{\text{batch size}}\log\frac{1}{\delta}}}{\varepsilon}$, where $T$ is the number of epochs. Here, we take into account the total number of steps that each client performs.
>
> **Q2: Learning rate sweeps.**
> ***Answer:*** We include these results in Tables 2-6 that were used in the experiments of Figure 4, following the reviewer's request.

---

> > ### Author Response · Authors · 2025-11-27
> > **Reminder**
> >
> > Dear Reviewer xbHq,
> >
> > We would like to respectfully follow up regarding our rebuttal. We understand the significant workload during the review period and appreciate the time you devote to evaluating submissions. If you have an opportunity, we would be grateful for any further comments or clarifications. Your feedback is highly valued.
> >
> > Thank you for your time and consideration.
> >
> > Best regards,
> >
> > Authors

---

### Official Review · Reviewer_n6J7 · 2025-10-30

**Soundness:** 4
**Presentation:** 4
**Contribution:** 3
**Rating:** 8
**Confidence:** 3

**Summary:**

The paper introduces a novel algorithm Clip21-SGD2M for federated learning with differential privacy. The algorithm combines 1) gradient clipping, 2) error feedback, and 3) double momentum (momentum on client and server). It achieves optimal convergence rates in the challenging but realistic setting of arbitrary data heterogeneity and stochastic gradients without any bounded-gradient assumption as in prior work. Experiments demonstrate a) that momentum is indeed essential to stabilize Clip21-SGD, and b) the proposed algorithm has empirically competitive (and in some cases superior) privacy/utility tradeoff to baselines that do not employ all three techniques.

**Strengths:**

This is a high-quality, technically dense paper that identifies a clear failure mode in existing methods, proposes a novel and well-motivated algorithm to fix it, and provides good theoretical and empirical validation. Theorem 2.2 proves that a recent and related algorithm, Clip21-SGD fails in the stochastic setting. The proposed algorithm combines three existing ideas to achieve optimal convergence rates in the challenging but realistic setting of arbitrary data heterogeneity and stochastic gradients without any bounded-gradient assumption as in prior work (Theorems 3.1/3.2). The paper also proves that the full algorithm achieves a competitive utility-privacy trade-off that matches state-of-the-art bounds (Theorem 3.3, Corollary 3.4). The experiments are very clear and demonstrate a) that the proposed algorithm is robust to the clipping threshold, while the baselines are not, and b) that controlling for the privacy budget $\epsilon$, the proposed algorithm has competitive-to-superior performance. The work is very clearly written and highly significant for the FL and private optimization communities.

**Weaknesses:**

The primary weakness of the paper is that the entire theoretical analysis assumes a full-participation model, where all clients participate at every round. This weakness is transparently acknowledged by the authors. The experiment in Fig 5 partially alleviates this by providing at least some preliminary empirical support for the effectiveness of the algorithm under subsampling.

Also, the experiments with DP were run only on MNIST -- even rerunning these experiments on CIFAR would strengthen the conclusions.

**Questions:**

Could you give the equivalent of Figure 4 using CIFAR? Or better yet, a truly federated dataset (and a different architecture) for example an LSTM trained on the Stack overflow dataset? This would strengthen the empirical claims of the paper, although I would not insist that it is necessary for a conference publication.

Typo: extra parens in algs 1/2

---

> ### Author Response · Authors · 2025-11-20
> **Rebuttals**
>
> We sincerely thank the reviewer for the constructive and encouraging feedback. We greatly appreciate the positive assessment of our work, emphasizing its significance for the FL and privacy communities, the clarity of its motivation and presentation, and the strength of its theoretical and empirical validation.
>
> **W1: Full client participation.**
>
> **Answer:** Our primary goal is to relax the assumptions on the objective and the gradient noise used in the DP literature, and to carry out the analysis under conditions that are closer to those observed in practice. In general, conducting a high-probability analysis with any form of subsampling is challenging, even for vanilla SGD. Advancing this line of work, therefore, requires first developing more refined proof techniques in simpler settings.
>
> **W2: Limited experiments.**
>
> **Answer:** Our contributions are primarily theoretical, and the experimental setup is designed to support these theoretical claims, and it follows the standard practice in theoretical studies. To address the reviewer’s concern, we additionally include results for training a CNN model on the CIFAR-10 dataset (section J). In this setting, Clip21-SGD2M+ (a version of Clip21-SGD2M with per-sample clipping; Algorithm 5 in the revision) attains competitive performance and matches the accuracy of tuned Clip-SGD.
>
>
> **Q1: More experiments.**
>
> **Answer:** We acknowledge that additional empirical evidence on larger models and datasets would further strengthen our work. Nevertheless, we believe that the experiments presented already provide sufficient empirical support for our theoretical claims. We also refer to the response to **W2**.
>
> ***Q2:*** Typo: extra parens in algs 1/2
>
> ***Answer:*** We thank the reviewer for pointing out the typo.

---

### Official Review · Reviewer_7Gp8 · 2025-10-31

**Soundness:** 2
**Presentation:** 2
**Contribution:** 2
**Rating:** 2
**Confidence:** 4

**Summary:**

This manuscript addresses the challenge of designing algorithms that provide both strong Differential Privacy (DP) guarantees and efficient optimization rates for Federated Learning (FL). The authors propose a framework, called, Clip21-SGD2M, to improve optimization performance while maintaining strong privacy guarantees. Clip21-SGD2M integrates gradient clipping, heavy-ball momentum, and error feedback to as improvement. The authors prove that the proposed method achieves optimal convergence rates for non-convex smooth distributed problems with heterogeneous client data, without requiring restrictive boundedness assumptions. They also provide a privacy analysis of Clip21-SGD2M, demonstrating competitive local DP guarantees.

**Strengths:**

The manuscript provides extensive theoretical analysis for smooth non-convex distributed objectives under arbitrary data heterogeneity, and Clip21-SGD2M achieves the optimal O(1/T) in the full-batch regime while maintaining a competitive local DP guarantee.

**Weaknesses:**

1. There are several prior works that has explored the application of error feedback 21 (EF21) in the DP setup. For example, I noted that the structure proposed in this paper is almost identical to that shown in "Smoothed Normalization for Efficient Distributed Private Optimization" while the latter paper seems to include some further improvement through normalization. Given the different assumptions on gradient (this paper assumes stronger sub-Gaussian), I cannot easily compare the results and the contribution of this paper is unclear.

2. Clip21-SGD2M requires storing all historical updates for each client’s data, which introduces significant memory overhead. Moreover, the authors do not employ client or data sub-sampling in Clip21-SGD2M, which limits its scalability.

3. The paper did not present a proper and consistent comparison with sota DP-SGD works. The experiment results are organized in a confusing way. The majority of the experiments ignore the noise but only focus on Clip21-SGD2M. For the experiments with DP guarantees, the results seem much worse than the best-known results even for simple dataset like MNIST. For example, from "DIFFERENTIALLY PRIVATE LEARNING NEEDS BETTER FEATURES (OR MUCH MORE DATA)", even for $\epsilon=1$, one can achieve accuracy over 98%. Also, the results for DP-Clip21-SGD2M on CIFAR10 is missing.

3. A relatively minor issue here is that the theoretical analysis of Clip21-SGD2M is conducted under the assumptions of local smoothness and sub-Gaussian stochastic gradients. The latter is a stronger assumption than bounded variance, although the authors note that it is common to use sub-Gaussian local gradients for the type of high-probability analysis while only expectation-based results are presented in the paper.

**Questions:**

1. The experiments use privacy budgets ε∈{3,5.2,9,15.6,27} with δ=10^(-3), which covers moderate to weak privacy levels. Could the authors justify this choice and, if feasible, include results for smaller privacy budgets (e.g., ε≤1.5, δ>10^(-5)) to better illustrate performance under more practical privacy constraints? Also, please include concrete comparison with sota DP-SGD results on both MNIST and CIFAR10.

2. The manuscript claims robustness under arbitrary data heterogeneity, but the experimental setup lacks details. Could the authors specify the exact data partitioning schemes (e.g., type and degree of non-IIDness) used in each experiment?

3. Could the author detail and explain the key contribution of this paper compared to the prior work using EF21 in DP-SGD?

---

> ### Author Response · Authors · 2025-11-20
> **Rebuttals (part I)**
>
> We sincerely thank the reviewer for the constructive feedback, and in particular for highlighting that our paper provides extensive theoretical analysis of the proposed algorithm under arbitrary data heterogeneity, and competitive local DP guarantees.
>
> We hope that the following clarifications will help re-evaluate our submission and potentially adjust the score.
>
> **W1: Comparison with $\alpha$-NormEC.**
>
> ***Answer:*** We thank the reviewer for this comment. We would like to clarify the differences between these two works:
>
> - **a.** The algorithmic design is indeed similar, as both methods follow the EF21 framework. The key difference is that in $\alpha$-NormEC the transmitted vectors are bounded through normalization, whereas in our approach they are bounded through clipping.
>
> - **b.** The analysis of $\alpha$-NormEC ***relies on an even stronger assumption***, namely that full-batch gradients are used and the only source of randomness comes from the DP mechanism. In contrast, our work assumes a sub-Gaussian approximation of the gradient noise distribution, which has been observed in several ***practical settings*** [1]. Moreover, our results can be extended beyond the sub-Gaussian regime to heavier sub-Weibull tails with only minor modifications to the proofs, at the cost of slightly worse logarithmic factors in the final rates.
>
> - **c.** The experimental setup in [Shulgin et al., 2024] uses the central DP framework, where the aim is to prevent the server from distinguishing among clients, rather than to protect each client’s individual data. In contrast, our experiments employ the local DP framework, which is designed to safeguard each client’s data separately. Under this setting, we find that $\alpha$-NormEC performs poorly in practice, which aligns with the limitations highlighted in our earlier comments.
>
> - **d.** Finally, our non-convergence results for Clip21-SGD also extend to $\alpha$-NormEC, suggesting that the reliance on full-batch gradients is fundamental and cannot be eliminated at the theoretical level. In fact, $\alpha$-NormEC fails to converge even on simple quadratic objectives, which highlights its limitations in noisy environments.
>
>     Taken together, these points show that our analysis is carried out under ***substantially more relaxed*** assumptions, while $\alpha$-NormEC is limited to the full-batch regime.
>
>
>
> **W2: Necessity for momentum buffer storage and no data subsampling.**
>
> ***Answer:*** We would like to address each of the raised concerns.
>
> - **a.** We respectfully disagree that using momentum on the client side introduces significant memory overhead. In most modern applications, models are mostly trained with Adam or its modifications, which already maintain two momentum buffers. Our algorithmic design is therefore ***fully aligned with standard practice***.
>
> - **b.** We explicitly acknowledge that our theory does not cover subsampling in the text. At the same time, it remains an open question whether Clip-SGD can provably benefit from amplification through subsampling without ***imposing impractical assumptions*** like bounded gradients.
>
> - **c.** To partially address this limitation, we include experiments for client subsampling in the experiments. Our results demonstrate that our method is competitive in the client-subsampling regime.
>
> - **d.** To address the reviewer's concern regarding data subsampling, we provide new experimental results in the revision (section J). We demonstrate that Clip21-SGD2M+ (a variant of Clip21-SGD2M with per-sample clipping; Algorithm 5) achieves competitive performance in this case as well.
>
>
> **W3: Criticism of the experimental part.**
>
> ***Answer:*** For clarity, we break down our answer below.
>
> - **a.** First, we would like the reviewer to clarify their criticism. What "noise" does the reviewer refer to? In all our experiments, we have either mini-batch or additive Gaussian noise. In experiments, we test both sources of stochasticity, while in deep learning experiments, we use mini-batches.
>
> - **b.** Our experiments do not use amplification by subsampling, while in the work that the reviewer mentioned, they do use it. It allows for a significant reduction in the amount of DP noise added in the training, and leads to small drops in the performance in comparison to non-private training. Since in our theoretical framework we assume that gradients come from sub-Gaussian distributions, amplification by subsampling does not apply to our settings.
>
> - **c.** To address the reviewer's concern regarding the lack of experiments on CIFAR10, we include experiments of Clip21-SGD2M vs Clip-SGD when training the CNN model on CIFAR10 (section J). Our results demonstrate a competitive performance in this setup as well.
>
> - **d.** We do not aim to achieve SOTA performance of the algorithms in the experiments, but rather demonstrate the competitiveness of the proposed method to state-of-the-art Clip-SGD and support our theoretical analysis.

---

> > ### Author Response · Authors · 2025-11-20
> > **Rebuttals (part II)**
> >
> > **W4: Strong assumptions.**
> >
> > ***Answer:*** We kindly disagree with this statement for the following reasons.
> >
> > - **a.** The smoothness assumption is ***standard*** and widely used in the optimization literature, especially in the context of high-probability analysis. It is also ***far less restrictive*** than the bounded-gradient assumption, which is commonly imposed in the analysis of DP algorithms.
> >
> > - **b.** Sub-Gaussian noise does not encompass the bounded-variance assumption, but it still provides a good approximation of the gradient noise observed in practice [1] (see also our response to **W1**). It is also strictly less restrictive than the ***full-batch gradient*** or ***bounded-gradient*** assumptions used in prior work.
> >
> > - **c.** Finally, the reviewer’s comment that we present only an in-expectation analysis ***raises concerns about whether our work was read carefully***. All of our results are stated ***in high probability***, and this is explicitly indicated in each theorem. Such a misunderstanding may also suggest an ***improper reliance on LLM-based reviewing tools***.
> >
> > **Q1: Additional experiments.**
> >
> > ***Answer:***
> > - **a.** Such a choice of the privacy budget was made, since the models cannot be trained using smaller values of $\epsilon$ without amplification by subsampling.
> >
> > - **b.** To address the reviewer's concern, we include the results when training the CNN model on the CIFAR-10 dataset. We include them in the revision (section J), where we show the results for $\epsilon\in\{1,3,9\}$. We observe that our algorithm is competitive with the state-of-the-art performance of Clip-SGD, for which both LR and clipping radii were tuned.
> >
> > **Q2: Data partitioning.**
> >
> > ***Answer:*** We thank the reviewer for this comment. To simulate data heterogeneity, we partition the dataset as follows. Each client $i\in[n]$ is first assigned 1000 samples from a single class. We then shuffle the remaining data and distribute it randomly across all clients. This procedure creates a heterogeneous data distribution among clients.
> >
> > **Q3: Key contributions.**
> >
> > ***Answer:*** Our key contributions include
> >
> > - **a.** A novel algorithm for private training, whose design uniquely incorporates several important components: the EF21 mechanism to address data heterogeneity, a double-momentum scheme to smooth both gradient and DP noise, and a clipping procedure that enables the use of the Gaussian mechanism for protecting local data.
> >
> > - **b.** A high-probability analysis of the proposed algorithm. Our theoretical guarantees match known lower bounds for non-convex objectives: an $O(1/T)$ rate in the full-batch setting and an $O(1/\sqrt{T})$ rate without requiring strong assumptions such as bounded gradients. In the private training regime, our privacy–utility tradeoff aligns with the lower bounds in the regime $\sqrt{d}/(\sqrt{n}\varepsilon) \gg 1$, which is common in modern applications where the model dimension $d$ is large relative to the number of clients $n$.
> >
> > - **c.** Our analysis does not rely on the data heterogeneity bounds, such as bounded gradients. Clip21-SGD2M provably converges in arbitrarily heterogeneous environments, which is a standard situation for FL applications.
> >
> > - **d.** We demonstrate that prior methods based on the EF21 mechanism fail to converge in the presence of gradient noise. This limitation is fundamental and motivates the development of more advanced algorithms such as Clip21-SGD2M.
> >
> > [1] Wu, Yixin, et al. Revisiting the characteristics of stochastic gradient noise and dynamics. arXiv preprint arXiv:2109.09833, 2021.

---

> > > ### Author Response · Authors · 2025-11-27
> > > **Reminder**
> > >
> > > Dear Reviewer 7Gp8,
> > >
> > > We would like to respectfully follow up regarding our rebuttal. We understand the significant workload during the review period and appreciate the time you devote to evaluating submissions. If you have an opportunity, we would be grateful for any further comments or clarifications. Your feedback is highly valued.
> > >
> > > Thank you for your time and consideration.
> > >
> > > Best regards,
> > >
> > > Authors

---

### Official Review · Reviewer_JhTW · 2025-11-02

**Soundness:** 3
**Presentation:** 3
**Contribution:** 2
**Rating:** 4
**Confidence:** 3

**Summary:**

The paper proposes a new DP-FedAvg style algorithm that additionally integrates error feedback into design. The proposed algorithm proven to converge without bounded gradient assumption on heterogeneous data.

**Strengths:**

1. Comprehensive analysis are provided. The algorithm mixes a few mechanism including gradient clipping, error feedback, and momentum under arbitrary client heterogeneity, and convergence is proven without boundedness assumptions which is somehow novel.
2. Experiments are provided to show validity and superiority of the proposed algorithm.

**Weaknesses:**

1. The main weakness is, as acknowledged in the paper, the algorithm can not benefit from privacy amplification by subsampling. This makes the privacy-utility trade-off way worse.
2. Gradient distribution is assumed to be sub-gaussian, which is still a relatively restrictive assumption.
2. Experiments are in limited scale and limited variants.

**Questions:**

Can authors comment more on privacy-utility trade-offs of the proposed algorithm given it cannot benefit from privacy amplification by subsampling?

---

> ### Author Response · Authors · 2025-11-20
> **Rebuttals**
>
> We sincerely thank the reviewer for the constructive feedback and for recognizing the clarity and soundness of our work, as well as the novelty of our comprehensive analysis.
>
> We hope that the following clarifications will help re-evaluate our submission and potentially adjust the score.
>
> **W1: The algorithm can not benefit from privacy amplification.**
>
>
> ***Answer:*** For clarity, we break down our answer below.
> - **a.** We acknowledge this limitation of our theoretical analysis. At the same time, it remains an open question whether Clip-SGD can provably benefit from amplification through data subsampling ***without relying on impractical assumptions such as bounded gradients***.
>
> - **b.** To further address the reviewer’s concern, we compare Clip-SGD against a modified version of Clip21-SGD2M+ (a variant of Clip21-SGD2M with per-sample clipping; Algorithm 5 in the revision) when training a CNN model on the CIFAR10 dataset for $\varepsilon\in\{1,3,9\}$ and $\delta=10^{-4}$. We refer the reviewer to Appendix J for a detailed algorithm description, discussion on the theoretical analysis, and empirical results. We show that when privacy amplification by data subsampling is allowed, Clip21-SGD2M is still competitive in practice, matching the performance of the state-of-the-art Clip-SGD (see Figure J.1).
>
>
>
> **W2: Sub-Gaussian assumption is still restrictive.**
>
> ***Answer:*** First, we emphasize that this assumption is ***standard in high-probability analyses***, as noted in our work. Second, it provides a ***good approximation*** of the gradient noise distribution in several ***practical settings*** [1]. Finally, our results ***can be extended beyond the sub-Gaussian case*** to heavier sub-Weibull tails with only minor adjustments to the proofs, although this leads to slightly worse logarithmic factors in the final rates.
>
> Overall, we believe that assuming sub-Gaussian gradient noise offers a realistic approximation of empirical behavior while being ***far less restrictive than the bounded-gradient assumptions*** used in many previous studies. Relaxing this assumption remains an important direction for future work.
>
> [1] Wu, Yixin, et al. Revisiting the characteristics of stochastic gradient noise and dynamics. arXiv preprint arXiv:2109.09833, 2021.
>
>
>
> **W3: Limited experiments.**
>
> ***Answer:*** We acknowledge that increasing the size of models and datasets would provide more empirical evidence of the efficacy of Clip21-SGD2M. However, the ***main contributions of our work are theoretical***, and the primary goal of the experimental part of our work is to support our theoretical results by demonstrating competitive performance against baselines.
>
> Nonetheless, we add an additional experiment when training the CNN model on the CIFAR-10 dataset (see section J), summarized in our response to **W1**.
>
>
> **Q1: Comments on the privacy-utility tradeoff.**
>
>
> ***Answer:*** Corollary 3.4 describes the privacy-utility trade-off of Clip21-SGD2M. Moreover, in the paragraph following the corollary, we discuss the dependencies on $\varepsilon, d, n$. Under realistic assumptions on these parameters, the established dependency on $\varepsilon, d, n$ ***matches the best-known utility bounds*** for the non-convex case. We note that the amplification by subsampling does not affect the dependency on $\varepsilon, d, n$, though it improves the utility bound through the multiplicative factor of $q$ equal to the sampling probability (see lines 388-391). Therefore, the established privacy-utility trade-offs are standard up to the factor of $q$.

---

> > ### Author Response · Authors · 2025-11-27
> > **Reminder**
> >
> > Dear Reviewer JhTW,
> >
> > We would like to respectfully follow up regarding our rebuttal. We understand the significant workload during the review period and appreciate the time you devote to evaluating submissions. If you have an opportunity, we would be grateful for any further comments or clarifications. Your feedback is highly valued.
> >
> > Thank you for your time and consideration.
> >
> > Best regards,
> >
> > Authors

---

### Meta-Review · Area_Chair_xyN6 · 2026-01-06

**Summary:**

The paper proposes Clip21-SGD2M, a federated learning optimization algorithm combining clipping, error feedback, and double momentum, with convergence guarantees under differential privacy and arbitrary data heterogeneity. The main disagreement among reviewers is about whether the theoretical contribution (removing bounded gradient/heterogeneity assumptions with high-probability rates) is strong enough to outweigh limitations in assumptions, experimental scope, and lack of subsampling-based privacy amplification.

**Reviewer Concerns:**

1. Assumptions and theoretical novelty (sub-Gaussian noise, full participation): Partially addressed. The rebuttal clarifies that sub-Gaussianity is standard for high-probability analysis and weaker than bounded gradients, and explicitly corrects a misunderstanding about expectation vs high-probability results. However, concerns about full participation and realism remain acknowledged but unresolved.
2. Experimental coverage and comparisons to SOTA DP methods: Partially addressed. Authors add CIFAR-10 experiments and clarify goals are theoretical validation, not SOTA. This helps, but comparisons to stronger DP baselines and broader heterogeneity sweeps remain limited.
3. Privacy amplification and practical DP trade-offs: Not fully addressed. The rebuttal explains why subsampling amplification is not covered theoretically and provides partial empirical evidence via variants. Still, inability to leverage amplification remains a real downside for practical DP utility.

**Reviewer Scores:**

Reviewer JhTW (4 > 4): Clarifications help, but core concerns (privacy amplification, scale) still stand.
Reviewer 7Gp8: (2 > 2/3): Rebuttal addresses misunderstandings and adds experiments, but skepticism about novelty and empirical strength likely remains.
Reviewer n6J7: (8 > 8): Already positive; rebuttal aligns with reviewer’s framing and adds requested CIFAR results.
Reviewer xbHq: (4 > 4): Tightness and heterogeneity dependence clarified; experimental gaps partially acknowledged but not fully closed.

---

### Decision · Program_Chairs · 2026-01-26

Reject